# Neural Collapse: A Review on Modelling Principles and Generalization

**Vignesh Kothapalli**                                                                                          *vk2115@nyu.edu*
*Courant Institute of Mathematical Sciences*
*New York University*

**Reviewed on OpenReview:** *https://openreview.net/forum?id=QTXocpAP9p*

## Abstract

Deep classifier neural networks enter the terminal phase of training (TPT) when training error reaches zero and tend to exhibit intriguing Neural Collapse (NC) properties. Neural collapse essentially represents a state at which the within-class variability of final hidden layer outputs is infinitesimally small and their class means form a simplex equiangular tight frame. This simplifies the last layer behaviour to that of a nearest-class center decision rule. Despite the simplicity of this state, the dynamics and implications of reaching it are yet to be fully understood. In this work, we review the principles which aid in modelling neural collapse, followed by the implications of this state on generalization and transfer learning capabilities of neural networks. Finally, we conclude by discussing potential avenues and directions for future research.

## 1 Introduction

With unprecedented growth in the size of neural networks to billions and trillions of parameters, their capabilities seem to be limitless in the modern era (Liu et al., 2019; Brown et al., 2020; Thoppilan et al., 2022; Chowdhery et al., 2022; Yu et al., 2022; Zhai et al., 2022). Yet, their generalization capabilities continue to evade our understanding of deep learning techniques based on model complexity (Hu et al., 2021). One can aim to reason about these overparameterized networks by tracking and analysing the feature learning process over time, or by understanding simplified theoretical models. Although the theoretical foundations (Goodfellow et al., 2016; He & Tao, 2020) are being steadily improved, the role of novel empirical analysis is of paramount importance to speed up the process. In our work, we take a principled approach to review and analyse one such intriguing empirical phenomenon called "Neural Collapse" (Papyan et al., 2020). NC essentially defines four inter-related characteristics of the final and penultimate layers in deep classifier neural networks when trained beyond zero classification error (see figure 1):

**NC1:** *Collapse of variability:* For data samples belonging to the same class, their final hidden layer (i.e the penultimate layer) features concentrate around their class mean. Thus, the variability of intra-class features during training is lost as they collapse to a point.

**NC2:** *Preference towards a simplex equiangular tight frame:* The class means of the penultimate layer features tend to form a simplex equiangular tight frame (simplex ETF). A simplex ETF is a symmetric structure whose vertices lie on a hyper-sphere, are linearly separable and are placed at the maximum possible distance from each other.

**NC3:** *Self-dual alignment:* The columns of the last layer linear classifier matrix also form a simplex ETF in their dual vector space and converge to the simplex ETF (up to rescaling) of the penultimate layer features.

**NC4:** *Choose the nearest class mean:* When a test point is to be classified, the last-layer classifier essentially acts as a nearest (train)-class mean decision rule w.r.t penultimate layer features.

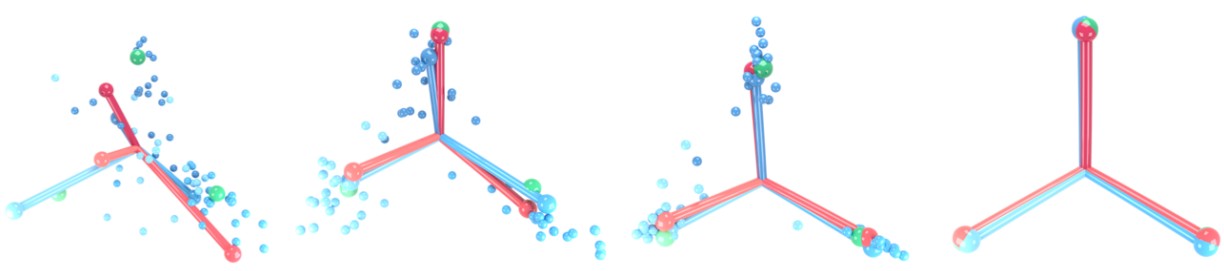

Figure 1: Evolution of penultimate layer outputs of a VGG13 neural network when trained on the CIFAR10 dataset with 3 randomly selected classes. The green balls represent the coordinates of a simplex ETF, red ball-and-sticks represent the final layer classifier, blue ball-and-sticks represent the class means and the small blue balls represent the last hidden layer (penultimate layer) activations. Image credit: Papyan et al. (2020)

Intuitively, these properties portray the tendency of the network to maximally separate class features while minimizing the separation within them. The benefits of such properties extend not only to generalization but to transfer learning and adversarial robustness as well (Liu et al., 2016; Wen et al., 2016; Sokolić et al., 2017; Liu et al., 2017; Cisse et al., 2017; Snell et al., 2017; Elsayed et al., 2018; Wang et al., 2018; Soudry et al., 2018; Jiang et al., 2018; Deng et al., 2019; Sun et al., 2020). On a related note, prior work by Giryes et al. (2016) showed that even in a shallow neural network with Gaussian random weights, the angle between intra-class features shrinks faster than the inter-class features. The variability collapse (NC1) property is an extreme version of this shrinking process when the network is deep enough. From a theoretical standpoint, by leveraging scattering transform based convolution operators as an alternative to trainable filters, the fundamental results of group invariant scattering by Mallat (2012); Bruna & Mallat (2013); Sifre & Mallat (2013) show the tendency of the network to reduce the scattering variance as the number of layers increases.

Furthermore, constraining the network weights to tight-frames has been shown to improve the adversarial robustness and training efficiency of the networks. For instance, Cisse et al. (2017) enforce the hidden layer weights of wide residual networks (Wide ResNet) (Zagoruyko & Komodakis, 2016) to be parseval tight frames (Kovačević et al., 2008) and show improved robustness to adversarial examples on CIFAR10 and SVHN data sets. Unlike the parseval networks which enforce structural constraints on all the hidden layers, the line of work by Pernici et al. (2019); Pernici et al.; 2021) fixes the final layer classifier as a regular polytope i.e, either a simplex, cube or an orthoplex, (Coxeter, 1973) and observe similar performance to learnable baselines on image classification tasks. Additionally, there seems to be an interesting, yet unexplored connection between the low-rank nature of these symmetric structures (such as a simplex ETF) and the 'rank-collapse' phase of training. Martin & Mahoney (2021) describe the 'rank-collapse' phase as a state of over-regularization during training, where the empirical spectral density of the weight matrices is dominated by a few large eigenvalues. On a similar note, the spectrum of the hessian of training loss was also shown to exhibit outlier eigenvalues which inherently represent the class information in image classification settings (Sagun et al., 2016; 2017; Wu et al., 2017; Papyan, 2018; Ghorbani et al., 2019).

The motivation behind reaching this collapsed state during TPT seems counter-intuitive as one would prefer to avoid over-fitting on the training data. However, recent observations based on "double-descent" by Belkin et al. (2019; 2020) and the benign effects of interpolating on the training data with over-parameterized networks (Ma et al., 2018; Belkin et al., 2018; Allen-Zhu et al., 2019; Feldman, 2020; Papyan et al., 2020; Bartlett et al., 2020; Zhang et al., 2021a) provide sufficient justification for further experimentation and analysis in this regime. To this end, prior work by Cohen et al. (2018) showed that the behaviour of k-Nearest Neighbor(k-NN) and deep classifier neural networks tend to be similar upon memorization. In fact, their experiments on the KL divergence between k-NN and Wide ResNet classifier outputs provide evidence for the emergence of NC4 property.

*The differentiating factor of NC from prior efforts lies in the fact that canonical deep classifier networks naturally exhibit all four properties without explicit structural constraints during training.* Questions pertaining

to theoretical explanations, modelling techniques and implications of NC naturally arise in this situation and we aim to address them in this work. The rest of the paper is organized as follows: section 2 details the preliminaries and setup that we use throughout the paper, section 3 introduces the modelling principles and reviews community efforts in a bottom-up fashion, section 4 presents the implications of NC on generalization and transfer learning, followed by takeaways and potential research directions in section 5.

## 1.1 Contributions

- We review and analyse NC modelling techniques based on the principles of unconstrained features and local elasticity, which is currently missing in the literature.

- We review and analyse the implications of NC on the generalization and transfer learning capabilities of deep classifier neural networks. Through these discussions, we hope to clarify certain misconceptions regarding NC on test data and provide directions for future efforts.

## 2 Preliminaries

In this paper, the term "network" refers to a neural network. Architectural details such as depth, presence of convolution layers, residual connections etc are omitted for brevity and will be mentioned as per the context. Since NC analysis using recurrent neural networks or its variants is absent in the literature, we omit such assumptions in this paper. We primarily focus on classification settings and employ a common notation scheme for all modelling techniques.

### 2.1 Data

Let's consider a data set $\boldsymbol{X} \in \mathbb{R}^{d \times N}$, for which $\mathbb{P}$ is assumed to be the underlying probability distribution. $\boldsymbol{X}$ is associated with labels $[K] = \{1, \ldots, K\}$, where $K \in \mathbb{N}$. The label function $\xi : \mathbb{R}^d \to \{\boldsymbol{e}_1, \ldots, \boldsymbol{e}_K\} \in \mathbb{R}^K$ is a $\mathbb{P}$ measurable ground-truth provider that maps an input to its respective one-hot vector. Formally, by representing the $i^{th}$ data point of the dataset $\boldsymbol{X}$ as $\boldsymbol{x}_i \in \mathbb{R}^d, \forall i \in [N]$ and with a slight abuse of notation, the $i^{th}$ data point of the $k^{th}$ class as $\boldsymbol{x}_i^k \in \mathbb{R}^d$, the function call $\xi(\boldsymbol{x}_i^k)$ outputs the ground truth vector $\boldsymbol{e}_k$. For notational convenience, we overload $\xi(\boldsymbol{x}_i^k)$ as $\xi_{\boldsymbol{x}_i^k}$. We define a set $C_k = \xi^{-1}(\{\boldsymbol{e}_k\})$ as the set of data points belonging to class $k \in [K]$, with $\mathbb{P}_{C_k}$ as their class conditional distribution. When indexing on the data is not necessary, we simply use $\boldsymbol{x} \in \mathbb{R}^d$ to represent a random data point. We represent the size of each class as $n_k, k \in [K]$ where $\sum_{k=1}^{K} n_k = N$. For the majority of this paper, we assume a balanced class setting and consider $n = N/K$ for convenience. Additionally, $\|.\|_F$ denotes the frobenius norm, $\|(\boldsymbol{r}, \boldsymbol{s})\|_E^2 = \|\boldsymbol{r}\|_F^2 + \|\boldsymbol{s}\|_F^2$, $\langle . \rangle$ denotes the inner product, $\mathrm{tr}\{.\}$ denotes the trace of a matrix and † denotes the pseudo-inverse.

### 2.2 Setup

A classifier network $h_L : \mathbb{R}^d \to \mathbb{R}^K$ belonging to a function class $\mathcal{H}$ can be formulated as a composition of $L - 1$ layers followed by a linear function. Formally, $h_L = a_L \circ f_{L-1} = a_L \circ g_{L-1} \circ g_{L-2} \cdots \circ g_1$ where $g_i : \mathbb{R}^{m_{i-1}} \to \mathbb{R}^{m_i}, \forall i \in [L-1]$ are parametric layers of the network, $f_{L-1} = g_{L-1} \circ g_{L-2} \cdots \circ g_1$ is the function obtained by composing $L - 1$ layers and $a_L : \mathbb{R}^{m_{L-1}} \to \mathbb{R}^{m_L}$ is the final layer linear function. For better readability, the number of layers $L$ is implicitly assumed and the sub-scripts are dropped. This gives us $h := h_L, a := a_L, f := f_{L-1}$ and simplifies $m_0 = d, m_L = K$. We also consider $m := m_{L-1}$ for the majority of the analysis that follows. As we will be dealing with data and label matrices, we formulate the matrix representation of a network as:

$$\boldsymbol{H} = \boldsymbol{A}\boldsymbol{F} + \boldsymbol{b} \tag{1}$$

Where $\boldsymbol{H} \in \mathbb{R}^{K \times N}$ is the network output matrix, $\boldsymbol{F} \in \mathbb{R}^{m \times N}$ is the penultimate layer feature matrix, $\boldsymbol{A} \in \mathbb{R}^{K \times m}$ is the final layer weight matrix and $\boldsymbol{b} \in \mathbb{R}^K$ is the final layer bias vector. We represent the outputs of $h : \mathbb{R}^d \to \mathbb{R}^K$ over $\boldsymbol{x}_i \in \mathbb{R}^d, i \in [N]$ as columns of $\boldsymbol{H} \in \mathbb{R}^{K \times N}$, outputs of $f : \mathbb{R}^d \to \mathbb{R}^m$ over $\boldsymbol{x}_i \in \mathbb{R}^d, i \in [N]$ as columns of $\boldsymbol{F} \in \mathbb{R}^{m \times N}$, and treat the one-hot label vectors $\{\xi_{\boldsymbol{x}_i}\}_{i \in [N]}$ as the columns

of label matrix $\boldsymbol{Y}$. For simplicity, we consider the ordered versions of $\boldsymbol{A} = [\boldsymbol{a}_1, \boldsymbol{a}_2, \cdots, \boldsymbol{a}_K]^\top \in \mathbb{R}^{K \times m}$ with rows $\boldsymbol{a}_k \in \mathbb{R}^m, \forall k \in [K]$, $\boldsymbol{F} = [\boldsymbol{f}_{1,1}, \cdots, \boldsymbol{f}_{1,n}, \cdots, \boldsymbol{f}_{K,n}] \in \mathbb{R}^{m \times N}$ with penultimate layer features for $\boldsymbol{x}_i^k$ given by the column $\boldsymbol{f}_{k,i} \in \mathbb{R}^m$, $\boldsymbol{H} = [\boldsymbol{h}_{1,1}, \cdots, \boldsymbol{h}_{1,n}, \cdots, \boldsymbol{h}_{K,n}] \in \mathbb{R}^{K \times N}$ with network output for $\boldsymbol{x}_i^k$ given by the column $\boldsymbol{h}_{k,i} \in \mathbb{R}^m$ and consider label matrix $\boldsymbol{Y}$ as a Kronecker product matrix $\boldsymbol{Y} = \boldsymbol{I}_K \otimes \boldsymbol{1}_n^\top \in \mathbb{R}^{K \times N}$. To measure the performance of network $h$, we use a generic loss function $\ell : \mathbb{R}^K \times \mathbb{R}^K \to [0, \infty)$ and define population risk functional $\mathcal{R} : \mathcal{H} \to [0, \infty)$ as:

$$\mathcal{R}(h) = \int_{\mathbb{R}^d} \ell(h(x), \xi_x) \mathbb{P}(dx) \tag{2}$$

The population risk can be approximated by the empirical risk on data $\boldsymbol{X}$, leading to the ERM problem:

$$\underset{h \in \mathcal{H}}{\arg\min} \, \widehat{\mathcal{R}}(h) = \frac{1}{N} \sum_{i=1}^{N} \ell(h(\boldsymbol{x}_i), \xi_{\boldsymbol{x}_i}) = \frac{1}{Kn} \sum_{k=1}^{K} \sum_{i=1}^{n} \ell(\boldsymbol{A}\boldsymbol{f}_{k,i} + \boldsymbol{b}, \boldsymbol{e}_k) \tag{3}$$

Most of the community efforts that we review in the following sections focus on the theoretical aspects of modelling neural collapse. From an empirical viewpoint, we observed that most of them employ the SGD optimizer for the ERM problem to validate the theory. The experimental details will be provided as per context but the reader can assume SGD with momentum as the default choice of optimizer unless specified otherwise. An extended set of notations for following the theory is available in Appendix.A.1.

### 2.3 Neural collapse

When a sufficiently expressive network $h$ is trained on $\boldsymbol{X}$ to minimize $\widehat{\mathcal{R}}(h)$, a zero training error point is reached when the classification error reaches 0. The network enters TPT when trained beyond this point and exhibits intriguing structural properties as follows:

**NC1:** *Collapse of Variability:* For all classes $k \in [K]$ and data points $i \in [n]$ within a class, the penultimate layer features $\boldsymbol{f}_{k,i}$ collapse to their class means $\boldsymbol{\mu}_k = \frac{1}{n} \sum_{i=1}^{n} \boldsymbol{f}_{k,i}$. We consider the within class covariance $\Sigma_W = \frac{1}{Kn} \sum_{k=1}^{K} \sum_{i=1}^{n} \left( (\boldsymbol{f}_{k,i} - \boldsymbol{\mu}_k)(\boldsymbol{f}_{k,i} - \boldsymbol{\mu}_k)^\top \right) \in \mathbb{R}^{m \times m}$, global mean $\boldsymbol{\mu}_G = \frac{1}{K} \sum_{k=1}^{K} \boldsymbol{\mu}_k \in \mathbb{R}^m$ and between class covariance $\Sigma_B = \frac{1}{K} \sum_{k=1}^{K} \left( (\boldsymbol{\mu}_k - \boldsymbol{\mu}_G)(\boldsymbol{\mu}_k - \boldsymbol{\mu}_G)^\top \right) \in \mathbb{R}^{m \times m}$ to measure the variability collapse.

*Empirical metric:*

$$\mathcal{NC}1 := \frac{1}{K} \text{tr}\{\Sigma_W \Sigma_B^\dagger\} \tag{4}$$

**NC2:** *Preference towards a simplex ETF:* The re-centered class means $\boldsymbol{\mu}_k - \boldsymbol{\mu}_G, \forall k \in [K]$ are equidistant from each other: $\|\boldsymbol{\mu}_k - \boldsymbol{\mu}_G\|_2 = \|\boldsymbol{\mu}_{k'} - \boldsymbol{\mu}_G\|_2$ for every $k, k' \in [K]$ and by concatenating $\frac{\boldsymbol{\mu}_k - \boldsymbol{\mu}_G}{\|\boldsymbol{\mu}_k - \boldsymbol{\mu}_G\|_2} \in \mathbb{R}^m, \forall k \in [K]$ to form a matrix $\boldsymbol{M} \in \mathbb{R}^{K \times m}$, $\boldsymbol{M}$ now represents a simplex ETF such that:

$$\boldsymbol{M}\boldsymbol{M}^\top = \frac{K}{K-1} \boldsymbol{I}_K - \frac{1}{K-1} \boldsymbol{1}_K \boldsymbol{1}_K^\top \tag{5}$$

$$cos(\boldsymbol{\mu}_k - \boldsymbol{\mu}_G, \boldsymbol{\mu}_{k'} - \boldsymbol{\mu}_G) = \frac{\langle \boldsymbol{\mu}_k - \boldsymbol{\mu}_G, \boldsymbol{\mu}_{k'} - \boldsymbol{\mu}_G \rangle}{\|\boldsymbol{\mu}_k - \boldsymbol{\mu}_G\|_2 \|\boldsymbol{\mu}_{k'} - \boldsymbol{\mu}_G\|_2} = -\frac{1}{K-1}, \forall k, k' \in [K], k \neq k' \tag{6}$$

*Empirical metric:*

$$\mathcal{NC}2 := \left\| \frac{\boldsymbol{M}\boldsymbol{M}^\top}{\|\boldsymbol{M}\boldsymbol{M}^\top\|_F} - \frac{1}{\sqrt{K-1}} \left( \boldsymbol{I}_K - \frac{1}{K} \boldsymbol{1}_K \boldsymbol{1}_K^\top \right) \right\|_F \tag{7}$$

**NC3:** *Self-dual alignment:* The last-layer classifier $\boldsymbol{A}$ is in alignment with the simplex ETF of $\boldsymbol{M}$ (up to rescaling) as:

$$\frac{\boldsymbol{A}}{\|\boldsymbol{A}\|_F} = \frac{\boldsymbol{M}}{\|\boldsymbol{M}\|_F}$$

*Empirical metric:*

$$\mathcal{NC}3 := \left\| \frac{\boldsymbol{AM}^\top}{\|\boldsymbol{AM}^\top\|_F} - \frac{1}{\sqrt{K-1}}\left(\boldsymbol{I}_K - \frac{1}{K}\boldsymbol{1}_K\boldsymbol{1}_K^\top\right) \right\|_F \tag{8}$$

**NC4:** *Choose the nearest class mean:* for any new test point $\boldsymbol{x}_{test}$, the classification result is determined by: $\arg\min_{k\in[K]} \|f(\boldsymbol{x}_{test}) - \boldsymbol{\mu}_k\|_2$. During training, one can track this property on $\boldsymbol{X}$ as a sanity check.

*Empirical metric:*

$$\mathcal{NC}4 := \frac{1}{Kn}\sum_{k=1}^{K}\sum_{i=1}^{n}\mathbb{I}(\arg\max_{c\in[K]}(\langle\boldsymbol{a}_c, \boldsymbol{f}_{k,i}\rangle + \boldsymbol{b}_c) \neq \arg\min_{c\in[K]}\|\boldsymbol{f}_{k,i} - \boldsymbol{\mu}_c\|_2) \tag{9}$$

Where $\mathbb{I}: \{True, False\} \to \{0,1\}$ is the indicator function and $\boldsymbol{b}_c \in \mathbb{R}$ is the $c^{th}$ element of the bias vector. The metric essentially represents the fraction of misclassified data points using the nearest class center (NCC) rule on the penultimate layer features. Based on this setup, we consider a network to be collapsed if the empirical metrics $\mathcal{NC}1\text{-}4 \to 0$. Without loss of generality, when we consider a network to exhibit NC1, it means that empirical measure $\mathcal{NC}1 \to 0$. The same holds for NC2-4. In the following sections, we review recent efforts in understanding the desirability, dynamics of occurrence and implications of these properties.

## 3  A Principled Modelling Approach

In this section, we focus on the principles of "*Unconstrained Features*" and "*Local Elasticity*" to model NC. The "*Unconstrained Features Model (UFM)*" analyzes the ideal values of $\boldsymbol{F}, \boldsymbol{A}, \boldsymbol{b}$ for perfect classification and the training dynamics that lead to them. This line of analysis assumes that $\boldsymbol{F}$ is freely optimizable and disconnected from the previous layers, including input. To the contrary, the "*Local Elasticity (LE)*" based model aims to capture the gradual separation of class features using stochastic differential equations (SDE) and similarity kernels. Additionally, this approach imitates the training dynamics of the network in a data-dependent fashion. Although the principle of unconstrained features has been the relatively popular approach, the earlier efforts by Wojtowytsch et al. (2020); Lu & Steinerberger (2020); Ergen & Pilanci (2021) did not explicitly use the UFM terminology. On the other hand, Mixon et al. (2020) and Fang et al. (2021) formalized and termed this approach as UFM and *"(N-)Layer-Peeled Model (LPM)"* respectively[1]. Thus, to avoid confusion due to terminology in the presentation of ideas, we stick to UFM for the rest of the paper.

### 3.1  Networks with "Unconstrained Features"

The unconstrained features model builds on the expressivity assumption of function class $\mathcal{H}$ and attempts to explain NC w.r.t ideal geometries and training dynamics. We consider a network to be expressive enough for $\boldsymbol{X}$ if it achieves perfect classification on $\boldsymbol{X}$. Under this assumption, the penultimate layer is disconnected from the previous layers and treated as free optimization variables during training (see figure 2). Since the last two layers of canonical classifier networks are fully connected/dense, we assume this to be the case for UFM analysis as well. To this end, we study the properties of $(\boldsymbol{F}, \boldsymbol{A}, \boldsymbol{b})$ under various settings pertaining to: *loss functions, regularization,* and *normalization* to derive insights on their collapse properties.

#### 3.1.1  Role of cross-entropy loss without regularization

**A note on desirability:** Prior to analysing the NC properties, we start by briefly reviewing the work of Wojtowytsch et al. (2020) which analyses the ideal outputs of $h$ that lead to minimal risk. By collapsing the network outputs to a single point based on their class, and repeating it for all classes $k \in [K]$, we get:

$$\boldsymbol{z}_k := \frac{1}{n}\int_{C_k} h(\boldsymbol{x}')\mathbb{P}(d\boldsymbol{x}') \tag{10}$$

---

[1]Here $N$ indicates the layer features (from the end of the network) which can be freely optimized. Additionally, note that the UFM and its extended version by Tirer & Bruna (2022) can now be considered as the 1-LPM and 2-LPM variants.

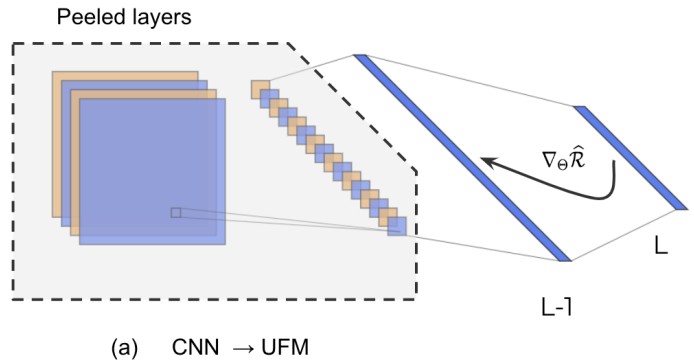
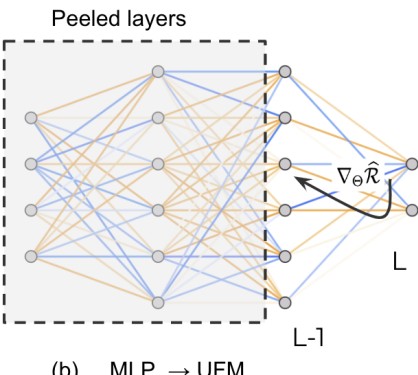

(a)   CNN → UFM                                  (b)   MLP → UFM

Figure 2: From left to right, we illustrate UFMs corresponding to an expressive Convolutional Neural Network (CNN) and MLP respectively. The shaded regions in both plots pertain to the first $L - 2$ layers which are 'peeled away' from the last two layers. Here $\Theta = (\boldsymbol{F}, \boldsymbol{A}, \boldsymbol{b})$ indicates the trainable parameters and $\nabla_\Theta \widehat{\mathcal{R}}$ indicates the gradients of the empirical risk w.r.t $\Theta$ that are backpropagated. Under the expressivity assumption, observe that the nature of the first $L - 2$ layers is impertinent and allows the UFM to encompass a variety of network architectures for analysis.

Without loss of generality, if we consider a network $\bar{h} \in \mathcal{H}$ to exist such that $\bar{h}(\boldsymbol{x}) = \boldsymbol{z}_k, \forall \boldsymbol{x} \in C_k$, then Wojtowytsch et al. (2020) showed that $\mathcal{R}(\bar{h}) \leq \mathcal{R}(h)$ when $\ell = \ell_{CE}$:

$$\ell_{CE}(h(\boldsymbol{x}), \xi_{\boldsymbol{x}}) = -\log\left(\frac{\exp(\langle h(\boldsymbol{x}), \xi_{\boldsymbol{x}}\rangle)}{\sum_{j=1}^{K} \exp(\langle h(\boldsymbol{x}), \boldsymbol{e}_j\rangle)}\right) \tag{11}$$

Thus, indicating the desirability of variance collapse for the network outputs (refer Appendix.A.2 for further details on this result). Note that for sufficiently expressive function classes which are scale-invariant, if $\langle h(\boldsymbol{x}), \xi_{\boldsymbol{x}}\rangle > \max\limits_{\boldsymbol{e}_i \neq \xi_{\boldsymbol{x}}, \forall i \in [K]} \langle h(\boldsymbol{x}), \boldsymbol{e}_i\rangle$ (i.e in TPT), then $\lim_{\lambda \to \infty} \mathcal{R}(\lambda h) = 0$. Out of these infinitely many possible solutions, one can assume norm bounded $\mathcal{H}$ and focus on minimizer results whose structure can be analysed. Based on this assumption, let's consider $\mathcal{H}$ to be an expressive class of functions from the input space to the Euclidean ball of radius $R$ and center at the origin: $B_R(0) \in R^K$. The empirical risk can now be minimized over the class means when variance collapse of network outputs occurs. Observe that when:

$$h^*(\boldsymbol{x}_i^k) = \operatorname*{arg\,min}_{h(\boldsymbol{x}_i^k) \in B_R(\boldsymbol{b})} \left(-\log\left(\frac{\exp(\langle h(\boldsymbol{x}_i^k), \boldsymbol{e}_k\rangle)}{\sum_{j=1}^{K} \exp(\langle h(\boldsymbol{x}_i^k), \boldsymbol{e}_j\rangle)}\right)\right) \tag{12}$$

The Lagrange multiplier equations for this minimization problem lead to:

$$h^*(\boldsymbol{x}_i^k) = \sqrt{\frac{K-1}{K}} R \boldsymbol{e}_k - \frac{R}{\sqrt{K(K-1)}} \sum_{j \neq k}^{K} \boldsymbol{e}_j$$

Thus, forming a simplex ETF of the collapsed network outputs. Note that the bias $\boldsymbol{b}$ led to a re-centering of $B_R$ and didn't affect the simplex ETF formation (Wojtowytsch et al., 2020). In a concurrent line of work, Lu & Steinerberger (2022) showed related results for $\ell_{CE}$, where the collapsed outputs of the network $h$ satisfy the angle property given in equation 6 and indicate the desirability of forming a simplex ETF.

The transformation from the penultimate layer to the final layer is affine w.r.t $\boldsymbol{A}, \boldsymbol{b}$ as per equation 1. Now, by noting that $(\boldsymbol{A} \cdot + \boldsymbol{b})^{-1}(\boldsymbol{z}_i)$ is an $m - K$ dimensional affine subspace of $\mathbb{R}^m$, it is desirable for collapse to occur in the penultimate layer features when $f$ is $l^p$-norm constrained. This is due to the strictly convex

nature of $l^p$-norm ($1 < p < \infty$) which leads to a unique mapping from the collapsed final layer outputs to collapsed penultimate layer features. See section 4 in Wojtowytsch et al. (2020) for detailed proofs and additional analysis of this result.

**Gradient Flow:** *With the desirability of collapse being established w.r.t population risk, a question that naturally arises is whether the dynamics of the optimizers in ERM settings tend towards such states, especially without the norm-constraints which leads to non-unique minimizers?* The work by Ji et al. (2021) addresses this question by analysing the gradient flow[2] of the empirical risk with a cross-entropy loss under the zero bias assumption. We present the key results from their analysis of the gradient flow and its relations with a max-margin separation problem below. Firstly, observe that the empirical risk $\widehat{\mathcal{R}}$ with $\ell = \ell_{CE}$ leads to ERM with $\widehat{\mathcal{R}}_{CE}$ as:

$$\min_{\boldsymbol{F},\boldsymbol{A}} \widehat{\mathcal{R}}_{CE}(\boldsymbol{F},\boldsymbol{A}) = -\frac{1}{N}\sum_{k=1}^{K}\sum_{i=1}^{n}\log\left(\frac{\exp(\boldsymbol{a}_k^\top \boldsymbol{f}_{k,i})}{\sum_{j=1}^{K}\exp(\boldsymbol{a}_j^\top \boldsymbol{f}_{k,i})}\right) \tag{13}$$

This leads to the gradient flow formulation as follows:

$$\begin{aligned}
\frac{d\boldsymbol{F}(t)}{dt} &= -\frac{\partial\widehat{\mathcal{R}}_{CE}(\boldsymbol{F}(t),\boldsymbol{A}(t))}{\partial\boldsymbol{F}} \\
\frac{d\boldsymbol{A}(t)}{dt} &= -\frac{\partial\widehat{\mathcal{R}}_{CE}(\boldsymbol{F}(t),\boldsymbol{A}(t))}{\partial\boldsymbol{A}}
\end{aligned} \tag{14}$$

Where $\boldsymbol{F}(t), \boldsymbol{A}(t)$ are indexed by time $t$ of the gradient flow. Formally, by defining the margin of a data point $\boldsymbol{x}_i^k$ and the associated penultimate layer feature $\boldsymbol{f}_{k,i}$ as: $q_{k,i}(\boldsymbol{F},\boldsymbol{A}) := \boldsymbol{a}_k^\top \boldsymbol{f}_{k,i} - \max_{j\neq k}\boldsymbol{a}_j^\top \boldsymbol{f}_{k,i}$, then reaching TPT indicates that $q_{k,i}(\boldsymbol{F},\boldsymbol{A}) \geq 0, \forall k \in [K], i \in [n]$, i.e, the features can be perfectly separated. In such a setting, Ji et al. (2021) proved that, if $(\boldsymbol{F}(t), \boldsymbol{A}(t))$ evolve as per the gradient flow of equation 14, then any limit point of the form: $\{(\hat{\boldsymbol{F}}(t), \hat{\boldsymbol{A}}(t) := (\frac{\boldsymbol{F}(t)}{\sqrt{\|\boldsymbol{A}(t)\|_F^2+\|\boldsymbol{F}(t)\|_F^2}}, \frac{\boldsymbol{A}(t)}{\sqrt{\|\boldsymbol{A}(t)\|_F^2+\|\boldsymbol{F}(t)\|_F^2}})\}$ is along the direction of an $(\epsilon,\delta)$-approximate Karush-Kuhn-Tucker (KKT) (Gordon & Tibshirani, 2012) point with $\epsilon, \delta \to 0$, of the following minimum-norm separation problem:

$$\min_{\boldsymbol{A},\boldsymbol{F}} \frac{1}{2}\|\boldsymbol{A}\|_F^2 + \frac{1}{2}\|\boldsymbol{F}\|_F^2$$
$$\text{s.t } \boldsymbol{a}_k^\top \boldsymbol{f}_{k,i} - \boldsymbol{a}_j^\top \boldsymbol{f}_{k,i} \geq 1, \ k \neq j \in [K], i \in [n] \tag{15}$$

Now, by considering $q_{min}(\boldsymbol{F},\boldsymbol{A}) := \min_{k\in[K],i\in[n]} q_{k,i}(\boldsymbol{F},\boldsymbol{A})$ as the margin of data set $\boldsymbol{X}$, which is bounded by $q_{min}(\boldsymbol{F},\boldsymbol{A}) \leq \frac{\|\boldsymbol{A}\|_F^2+\|\boldsymbol{F}\|_F^2}{2(K-1)\sqrt{n}}$, Ji et al. (2021) show that the maximum separation is attained when $\|\boldsymbol{A}\|_F = \|\boldsymbol{F}\|_F$ and $(\boldsymbol{F},\boldsymbol{A})$ satisfy the NC properties. Finally, to show that the approximate KKT points are indeed the desired global minima, it is sufficient to show that the separation problem of equation 15 satisfies the Mangasarian-Fromovitz Constraint Qualification (MFCQ) (Mangasarian & Fromovitz, 1967), refer Dutta et al. (2013) and theorem 3.1, 3.2 in Ji et al. (2021) for detailed proofs. Although this line of analysis focuses on NC properties of the max-margin solutions, similar proof sketches were employed by Nacson et al. (2019); Lyu & Li (2019) to study the implicit bias of gradient descent in homogeneous neural networks.

**Loss Landscape:** The non-convex nature of the risk in equation 13 can result in KKT points which are not global minimizers. If $(\boldsymbol{F},\boldsymbol{A})$ don't satisfy NC properties or $\|\boldsymbol{A}\|_F \neq \|\boldsymbol{F}\|_F$, then a direction exists in the tangent space of $(\boldsymbol{F},\boldsymbol{A})$ that leads to lower $\widehat{\mathcal{R}}_{CE}$ values. Formally, by defining the tangent space of $(\boldsymbol{F},\boldsymbol{A})$ as $\{\Delta\boldsymbol{F} \in \mathbb{R}^{m\times N}, \Delta\boldsymbol{A} \in \mathbb{R}^{K\times m} : \text{tr}\{\boldsymbol{F}^\top\Delta\boldsymbol{F}\}+\text{tr}\{\boldsymbol{A}^\top\Delta\boldsymbol{A}\} = 0\}$, then $\widehat{\mathcal{R}}_{CE}(\boldsymbol{F}+\delta\Delta\boldsymbol{F},\boldsymbol{A}+\delta\Delta\boldsymbol{A}) < \widehat{\mathcal{R}}_{CE}(\boldsymbol{F},\boldsymbol{A})$, for a constant $\delta_{max} > 0$ and $\forall 0 < \delta < \delta_{max}$. This result by Ji et al. (2021) was proved using second-order analysis of the empirical risk in equation 13 (without the $1/N$ scaling factor) and analysing the eigenvector corresponding to a negative eigenvalue of the resulting Riemannian hessian matrix.

---

[2]Gradient flow can be treated as gradient descent with infinitesimal step sizes (Chizat & Bach, 2018; Du et al., 2018).

These theoretical observations pertaining to gradient flow were empirically validated by Ji et al. (2021) using ResNet18 and VGG13 networks to classify CIFAR10, MNIST, KMNIST and FashionMNIST data sets. SGD with momentum 0.3, and learning rate 0.01 was employed as the optimizer. Thus, the implicit regularization due to cross-entropy seems to be sufficient for converging to NC solutions.

### 3.1.2 Role of (mean) squared error without regularization

Following the cross-entropy case, let's consider the squared error $\ell = \ell_{SE}$ and the ERM formulated as:

$$\min_{\boldsymbol{F},\boldsymbol{A},\boldsymbol{b}} \widehat{\mathcal{R}}_{SE}(\boldsymbol{F},\boldsymbol{A},\boldsymbol{b}) = \frac{1}{2}\left\|\boldsymbol{A}\boldsymbol{F} + \boldsymbol{b}\mathbf{1}_N^\top - \boldsymbol{Y}\right\|_F^2 \tag{16}$$

Where $\mathbf{1}_N \in \mathbb{R}^N$ is the all ones vector. Unlike cross-entropy, it is convenient to deal with matrices instead of individual feature vectors for analysing the squared error setting. We consider the squared error in our analysis due to its empirical effectiveness on a variety of NLP and vision-based classification tasks (Janocha & Czarnecki, 2017; Hui & Belkin, 2020; Demirkaya et al., 2020).

**Gradient Flow:** In this section, we review the NC properties of squared error minimizers based on the gradient flow analysis presented in a recent effort by Mixon et al. (2020). With near 0 initialization of $(\boldsymbol{F},\boldsymbol{A},\boldsymbol{b})$, Mixon et al. (2020) observed the following 'Strong Neural Collapse (SNC)' properties of the minimizers as follows:

$$\text{SNC1}: \boldsymbol{A}\boldsymbol{A}^\top = \sqrt{n}(\boldsymbol{I}_K - \frac{1}{K}\mathbf{1}_K\mathbf{1}_K^\top)$$
$$\text{SNC2}: \boldsymbol{F} = \frac{1}{\sqrt{n}}(\boldsymbol{A} \otimes \mathbf{1}_n)^\top \tag{17}$$
$$\text{SNC3}: \boldsymbol{b} = \frac{1}{K}\mathbf{1}_K$$

Where $n = N/K$ (as per setup). These properties are called 'strong' as the standard NC properties can be derived from them. For instance, observe from SNC2 that $\boldsymbol{\mu}_k = \frac{1}{\sqrt{n}}\boldsymbol{a}_k$ satisfies NC1. Similarly, from SNC1 we can deduce that:

$$\boldsymbol{A}\boldsymbol{A}^\top\mathbf{1}_K = \sqrt{n}(\boldsymbol{I}_K - \frac{1}{K}\mathbf{1}_K\mathbf{1}_K^\top)\mathbf{1}_K = \sqrt{n}(\mathbf{1}_K - \mathbf{1}_K) = \mathbf{0} \implies \mathbf{1}_K^\top\boldsymbol{A}\boldsymbol{A}^\top\mathbf{1}_K = \left\|\boldsymbol{A}^\top\mathbf{1}_K\right\|_2^2 = 0$$

This result leads to:

$$\left\|\boldsymbol{\mu}_k - \boldsymbol{\mu}_G\right\|_2^2 = \left\|\frac{1}{\sqrt{n}}\boldsymbol{a}_k - \frac{1}{K}\sum_{j=1}^K\frac{1}{\sqrt{n}}\boldsymbol{a}_j\right\|_2^2 = \left\|\frac{1}{\sqrt{n}}\boldsymbol{a}_k - \frac{1}{\sqrt{n}K}\boldsymbol{A}^\top\mathbf{1}_K\right\|_2^2 = \left\|\frac{1}{\sqrt{n}}\boldsymbol{a}_k\right\|_2^2$$

Where $\left\|\frac{1}{\sqrt{n}}\boldsymbol{a}_k\right\|_2^2$ is the $k^{th}$ diagonal element of $\frac{1}{n}\boldsymbol{A}\boldsymbol{A}^\top$ and is equal to $\frac{1}{\sqrt{n}}(1 - \frac{1}{K})$. Thus, when $\boldsymbol{A}$ is normalized, we can see from SNC1 that:

$$\left\langle \frac{\boldsymbol{a}_k}{\|\boldsymbol{a}_k\|}, \frac{\boldsymbol{a}_l}{\|\boldsymbol{a}_l\|} \right\rangle = \frac{(\boldsymbol{A}\boldsymbol{A}^\top)_{kl}}{\sqrt{n}(1 - \frac{1}{K})} = \frac{\sqrt{n}(\boldsymbol{I}_K - \frac{1}{K}\mathbf{1}_K\mathbf{1}_K^\top)_{kl}}{\sqrt{n}(1 - \frac{1}{K})} = \left(\frac{K}{K-1}\boldsymbol{I}_K - \frac{1}{K-1}\mathbf{1}_K\mathbf{1}_K^\top\right)_{kl}$$

Where $(\boldsymbol{A}\boldsymbol{A}^\top)_{kl}$ indicates the $(k,l)^{th}$ element of $\boldsymbol{A}\boldsymbol{A}^\top$, resulting in the formation of simplex ETF by $\boldsymbol{A}$. Now, to understand how these properties were obtained, note that the gradient flow equation with $\Theta = (\boldsymbol{F},\boldsymbol{A},\boldsymbol{b})$ and the respective derivatives is given by:

$$\Theta'(t) = -\nabla\widehat{\mathcal{R}}_{SE}(\Theta(t))$$
$$\nabla_{\boldsymbol{F}}\widehat{\mathcal{R}}_{SE}(\Theta(t)) = \boldsymbol{A}^\top(\boldsymbol{A}\boldsymbol{F} + \boldsymbol{b}\mathbf{1}_N^\top - \boldsymbol{Y})$$
$$\nabla_{\boldsymbol{A}}\widehat{\mathcal{R}}_{SE}(\Theta(t)) = (\boldsymbol{A}\boldsymbol{F} + \boldsymbol{b}\mathbf{1}_N^\top - \boldsymbol{Y})\boldsymbol{F}^\top \tag{18}$$
$$\nabla_{\boldsymbol{b}}\widehat{\mathcal{R}}_{SE}(\Theta(t)) = (\boldsymbol{A}\boldsymbol{F} + \boldsymbol{b}\mathbf{1}_N^\top - \boldsymbol{Y})\mathbf{1}_N$$

The optimal value of $\boldsymbol{b}$ can be partially decoupled from $\boldsymbol{F}, \boldsymbol{A}$ when they are assumed to be small. Thus, the resultant ODE of the parameters:

$$\boldsymbol{F}'(t) = -\boldsymbol{A}(t)^\top(\boldsymbol{b}(t)\mathbf{1}_N^\top - \boldsymbol{Y}), \ \ \boldsymbol{A}'(t) = -(\boldsymbol{b}(t)\mathbf{1}_N^\top - \boldsymbol{Y})\boldsymbol{F}(t)^\top, \ \ \boldsymbol{b}'(t) = -(\boldsymbol{b}(t)\mathbf{1}_N^\top - \boldsymbol{Y})\mathbf{1}_N$$

with initial conditions $\boldsymbol{F}(0) = \boldsymbol{F}_0, \boldsymbol{A}(0) = \boldsymbol{A}_0, \boldsymbol{b}(0) = \mathbf{0}$ has a solution satisfying:

$$\left\|(\boldsymbol{F}(t), \boldsymbol{A}(t)) - e^{\sqrt{n}t} \cdot \Pi_{\mathcal{T}}(\boldsymbol{F}_0, \boldsymbol{A}_0)\right\|_E \leq e^{\frac{1}{K\sqrt{n}}} \cdot \left\|\Pi_{\mathcal{T}^\perp}(\boldsymbol{F}_0, \boldsymbol{A}_0)\right\|_E, \ \ \boldsymbol{b}(t) = \left(\frac{1 - e^{-Nt}}{K}\right)\mathbf{1}_K, \forall t \geq 0 \quad (19)$$

Where $\|(\boldsymbol{F}, \boldsymbol{A})\|_E^2 = \|\boldsymbol{F}\|_F^2 + \|\boldsymbol{A}\|_F^2$ as per setup and $\Pi_{\mathcal{T}}$ is the orthogonal projection onto the subspace:

$$\mathcal{T} := \left\{(\boldsymbol{F}, \boldsymbol{A}) : \boldsymbol{F} = \frac{1}{\sqrt{n}}(\boldsymbol{A} \otimes \mathbf{1}_n)^\top, \mathbf{1}_K^\top \boldsymbol{A} = \mathbf{0}\right\}$$

For a detailed proof, see theorem 2 in Mixon et al. (2020). This result implies that, during the initial stages of the gradient flow, a large component of $\boldsymbol{F}_0, \boldsymbol{A}_0$ resides in subspace $\mathcal{T}$ while $\boldsymbol{b}$ tends towards the span$\{\mathbf{1}_K\}$. Thus, the initial trajectory of $\Theta$, lies along the subspace $\mathcal{S}$ given by:

$$\mathcal{S} = \left\{(\boldsymbol{F}, \boldsymbol{A}, \boldsymbol{b}) : \boldsymbol{F} = \frac{1}{\sqrt{n}}(\boldsymbol{A} \otimes \mathbf{1}_n)^\top, \mathbf{1}_K^\top \boldsymbol{A} = \mathbf{0}, \boldsymbol{b} \in span\{\mathbf{1}_K\}\right\} \quad (20)$$

To this end, when $\Theta$ lies along $\mathcal{S}$, the risk modifies into:

$$\begin{aligned}
\widehat{\mathcal{R}}_{SE}(\boldsymbol{F}, \boldsymbol{A}, \boldsymbol{b}) &= \frac{1}{2}\left\|\boldsymbol{A}\boldsymbol{F} + \boldsymbol{b}\mathbf{1}_N^\top - \boldsymbol{Y}\right\|_F^2 = \frac{1}{2}\left\|\boldsymbol{A}(\frac{1}{\sqrt{n}}(\boldsymbol{A} \otimes \mathbf{1}_n)^\top) + \boldsymbol{b}\mathbf{1}_N^\top - \boldsymbol{Y}\right\|_F^2 \\
&= \frac{1}{2}\left\|\boldsymbol{A}(\frac{1}{\sqrt{n}}(\boldsymbol{A} \otimes \mathbf{1}_n)^\top) + \boldsymbol{b}\mathbf{1}_N^\top - (\boldsymbol{I}_K - \frac{1}{K}\mathbf{1}_K\mathbf{1}_K^\top + \frac{1}{K}\mathbf{1}_K\mathbf{1}_K^\top) \otimes \mathbf{1}_n^\top\right\|_F^2 \\
&= \frac{1}{2}\left\|(\frac{1}{\sqrt{n}}\boldsymbol{A}\boldsymbol{A}^\top - (\boldsymbol{I}_K - \frac{1}{K}\mathbf{1}_K\mathbf{1}_K^\top)) \otimes \mathbf{1}_n^\top + (\boldsymbol{b} - \frac{1}{K}\mathbf{1}_K)\mathbf{1}_N^\top\right\|_F^2 \\
&= \frac{1}{2}\left\|\boldsymbol{A}\boldsymbol{A}^\top - \sqrt{n}(\boldsymbol{I}_K - \frac{1}{K}\mathbf{1}_K\mathbf{1}_K^\top)\right\|_F^2 + \frac{N}{2}\left\|\boldsymbol{b} - \frac{1}{K}\mathbf{1}_K\right\|_F^2
\end{aligned}$$

Thus, showing that $(\boldsymbol{F}, \boldsymbol{A}, \boldsymbol{b})$ satisfying SNC properties are indeed the minimizers of $\widehat{\mathcal{R}}_{SE}$. The last equality is valid since the two terms are orthogonal when $\Theta$ lies along $\mathcal{S}$. The empirical setup to verify this behaviour is simple. For some choice of $K, N, m$, one can randomly initialize $\boldsymbol{A}_0, \boldsymbol{F}_0$ and set $\boldsymbol{b} = \mathbf{0}, \boldsymbol{Y} = \boldsymbol{I}_K \otimes \mathbf{1}_n^\top$. This setup is sufficient for performing gradient descent w.r.t $\widehat{\mathcal{R}}_{SE}$ and tracking the NC properties of weights and features across steps/iterations. Such an empirical analysis by Mixon et al. (2020) showed that SNC properties are highly sensitive to initialization of $(\boldsymbol{F}_0, \boldsymbol{A}_0)$ i.e, by defining the SNC errors as:

$$\delta_{SNC1} = \left\|\boldsymbol{A}\boldsymbol{A}^\top - \sqrt{n}(\boldsymbol{I}_K - \frac{1}{K}\mathbf{1}_K\mathbf{1}_K^\top)\right\|_F, \delta_{SNC2} = \left\|\boldsymbol{F} - \frac{1}{\sqrt{n}}(\boldsymbol{A} \otimes \mathbf{1}_n)^\top\right\|_F, \delta_{SNC3} = \left\|\boldsymbol{b} - \frac{1}{K}\mathbf{1}_K\right\|_F,$$

Figure 3 illustrates SNC errors that are orders of magnitude higher as initialization moves away from 0. Furthermore, Mixon et al. (2020) also confirm that, as $\|\boldsymbol{F}_0\|_F, \|\boldsymbol{A}_0\|_F$ tend towards 0, the entire trajectory of gradient descent tends to stay along $\mathcal{S}$. The theoretical analysis of the full trajectory behaviour, especially pertaining to the implicit bias of gradient descent towards NC solutions is still lacking and open for research. *We show in Appendix.A.3 that the mean squared error can be theoretically analysed in the same fashion, and show resemblance to the above results.*

**Loss Landscape:** As the UFM can be considered as a linear neural network with optimizable inputs, the existing work by Baldi & Hornik (1989); Saxe et al. (2013); Kawaguchi (2016); Freeman & Bruna (2016) is relevant to our study. Based on the key results of these efforts, it can be shown under mild assumptions that the landscape of squared error loss for linear networks contains critical points which are either global minima

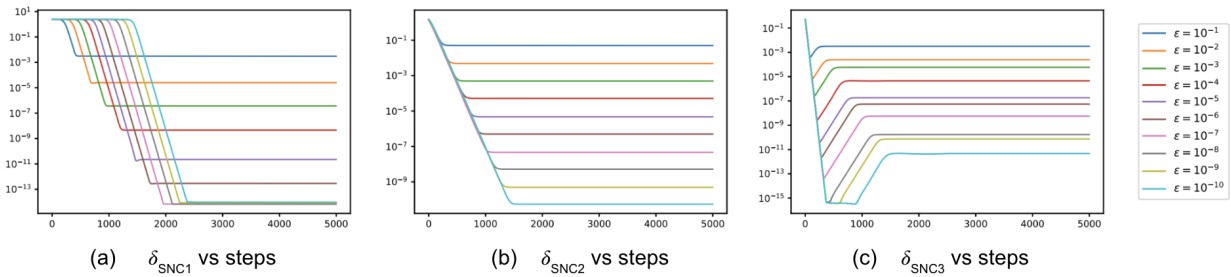

Figure 3: (Left-right) Visualization of $\delta_{SNC1}, \delta_{SNC2}, \delta_{SNC3}$ with initialization $\|\boldsymbol{A}_0\|_F = \epsilon, \|\boldsymbol{F}_0\|_F = \epsilon, \boldsymbol{b}_0 = \boldsymbol{0}, K = 3, N = 9, m = 15$ vs gradient descent steps on $\widehat{\mathcal{R}}_{SE}$. Image credit: Mixon et al. (2020)

or strict saddles with negative curvature. Although the landscape is benign, note that we always need some randomness to escape these saddle points. Due to this reason, the vanilla gradient descent which doesn't include randomness in its updates may get stuck in one, which we believe is a possible reason for large SNC errors in figure 3. Similar observations were made by Ji et al. (2021) for $\widehat{\mathcal{R}}_{CE}$ and gradient descent.

### 3.1.3 Role of cross-entropy loss with regularization

Our analysis till now has been limited to simplified risk formulations. In this section, we move closer to practical settings by incorporating weight and feature regularizations.

**Lower bound:** As the empirical risk is dependent on $\boldsymbol{F}, \boldsymbol{A}, \boldsymbol{b}$, the regularized version of ERM with $\widehat{\mathcal{R}}_{CE}$ can be formulated as follows:

$$\min_{\boldsymbol{F},\boldsymbol{A},\boldsymbol{b}} \widehat{\mathcal{R}}_{CEr}(\boldsymbol{F}, \boldsymbol{A}, \boldsymbol{b}) = \frac{1}{N} \sum_{k=1}^{K} \sum_{i=1}^{n} \ell_{CE}(\boldsymbol{A}\boldsymbol{f}_{k,i} + \boldsymbol{b}, \boldsymbol{e}_k) + \frac{\lambda_{\boldsymbol{A}}}{2} \|\boldsymbol{A}\|_F^2 + \frac{\lambda_{\boldsymbol{F}}}{2} \|\boldsymbol{F}\|_F^2 + \frac{\lambda_{\boldsymbol{b}}}{2} \|\boldsymbol{b}\|_2^2 \quad (21)$$

Where $\lambda_{\boldsymbol{A}}, \lambda_{\boldsymbol{F}}, \lambda_{\boldsymbol{b}} > 0$ are penalty terms. To this end, Zhu et al. (2021) lower bound $\widehat{\mathcal{R}}_{CEr}(\boldsymbol{F}, \boldsymbol{A}, \boldsymbol{b})$ by:

$$\begin{aligned}
\widehat{\mathcal{R}}_{CEr}(\boldsymbol{F}, \boldsymbol{A}, \boldsymbol{b}) &= \frac{1}{N} \sum_{k=1}^{K} \sum_{i=1}^{n} \ell_{CE}(\boldsymbol{A}\boldsymbol{f}_{k,i} + \boldsymbol{b}, \boldsymbol{e}_k) + \frac{\lambda_{\boldsymbol{A}}}{2} \|\boldsymbol{A}\|_F^2 + \frac{\lambda_{\boldsymbol{F}}}{2} \|\boldsymbol{F}\|_F^2 + \frac{\lambda_{\boldsymbol{b}}}{2} \|\boldsymbol{b}\|_2^2 \\
&\geq -\frac{\|\boldsymbol{A}\|_F^2}{(1+c_1)(K-1)} \sqrt{\frac{\lambda_{\boldsymbol{A}}}{n\lambda_{\boldsymbol{F}}}} + c_2 + \lambda_{\boldsymbol{A}} \|\boldsymbol{A}\|_F^2
\end{aligned} \quad (22)$$

Where $c_1 > 0, c_2 = \frac{1}{c_1+1} \log((1+c_1)(K-1)) + \frac{c_1}{1+c_1} \log(\frac{1+c_1}{c_1})$ and equality holds true when $\boldsymbol{F}, \boldsymbol{A}$ satisfy NC properties and $\|\boldsymbol{A}\|$ is finite. Zhu et al. (2021)) arrive at this result by expanding the cross-entropy formulation, identifying this lower bound and showing that it can be achieved when $\boldsymbol{F}, \boldsymbol{A}$ satisfy NC properties.

**Loss landscape:** To analyse the loss landscape, we summarize the approach of Zhu et al. (2021), which leverages the presence of regularizing terms to form a connection between the non-convex problem in equation 21 and a convex program as follows:

$$\min_{\boldsymbol{Z},\boldsymbol{b}} \widetilde{\mathcal{R}}_{CEr}(\boldsymbol{Z}, \boldsymbol{b}) := \frac{1}{N} \sum_{k=1}^{K} \sum_{i=1}^{n} \ell_{CE}(\boldsymbol{Z}_{k,i} + \boldsymbol{b}, \boldsymbol{e}_k) + \sqrt{\lambda_{\boldsymbol{A}}\lambda_{\boldsymbol{F}}} \|\boldsymbol{Z}\|_* + \frac{\lambda_{\boldsymbol{b}}}{2} \|\boldsymbol{b}\|_2^2 \quad (23)$$

Where $\boldsymbol{Z} = \boldsymbol{A}\boldsymbol{F} \in \mathbb{R}^{K \times N}$ and $\|.\|_*$ denotes the nuclear norm. The validity of this connection can be understood from the following result by Zhu et al. (2021) (with similar results in Srebro (2004); Haeffele & Vidal (2015)) given as:

$$\min_{\boldsymbol{A}\boldsymbol{F}=\boldsymbol{Z}} \frac{\lambda_{\boldsymbol{A}}}{2} \|\boldsymbol{A}\|_F^2 + \frac{\lambda_{\boldsymbol{F}}}{2} \|\boldsymbol{F}\|_F^2 = \sqrt{\lambda_{\boldsymbol{A}}\lambda_{\boldsymbol{F}}} \min_{\boldsymbol{A}\boldsymbol{F}=\boldsymbol{Z}} \frac{\sqrt{\lambda_{\boldsymbol{A}}}}{2\sqrt{\lambda_{\boldsymbol{F}}}} \left( \|\boldsymbol{A}\|_F^2 + \frac{\lambda_{\boldsymbol{F}}}{\lambda_{\boldsymbol{A}}} \|\boldsymbol{F}\|_F^2 \right) = \sqrt{\lambda_{\boldsymbol{A}}\lambda_{\boldsymbol{F}}} \|\boldsymbol{Z}\|_* \quad (24)$$

By exploiting this connection with the convex program, Zhu et al. (2021) find the global minimizers of $\widetilde{\mathcal{R}}_{CEr}$ in equation 23 and show that they provide a lower bound for $\widehat{\mathcal{R}}_{CEr}(\boldsymbol{F}, \boldsymbol{A}, \boldsymbol{b})$. Formally, if $(\boldsymbol{Z}_*, \boldsymbol{b}_*)$ is the global minimizer of $\widetilde{\mathcal{R}}_{CEr}(\boldsymbol{Z}, \boldsymbol{b})$, then $\widetilde{\mathcal{R}}_{CEr}(\boldsymbol{Z}_*, \boldsymbol{b}_*) \leq \widehat{\mathcal{R}}_{CEr}(\boldsymbol{F}, \boldsymbol{A}, \boldsymbol{b})$, and this optimal state can be transferred to the non-convex $\widehat{\mathcal{R}}_{CEr}(\boldsymbol{F}, \boldsymbol{A}, \boldsymbol{b})$. For the sake of conciseness, we directly state the result of lemma C.4 in Zhu et al. (2021) that, any critical point $(\boldsymbol{F}, \boldsymbol{A}, \boldsymbol{b})$ of equation 21 satisfying:

$$\left\| \nabla_{\boldsymbol{Z}=\boldsymbol{A}\boldsymbol{F}} \left( \frac{1}{N} \sum_{k=1}^{K} \sum_{i=1}^{n} \ell_{CE}(\boldsymbol{A}\boldsymbol{f}_{k,i} + \boldsymbol{b}, \boldsymbol{e}_k) \right) \right\|_2 \leq \sqrt{\lambda_{\boldsymbol{A}} \lambda_{\boldsymbol{F}}} \tag{25}$$

is a global minimizer of $\widetilde{\mathcal{R}}_{CEr}$ with $\boldsymbol{Z} = \boldsymbol{A}\boldsymbol{F}$. To this end, Zhu et al. (2021) classify the critical points $\mathcal{C}$ of $\widehat{\mathcal{R}}_{CEr}(\boldsymbol{F}, \boldsymbol{A}, \boldsymbol{b})$ into two disjoint subsets as follows:

$$\mathcal{C} = \left\{ \boldsymbol{F}, \boldsymbol{A}, \boldsymbol{b} : \nabla_{\boldsymbol{A}}\widehat{\mathcal{R}}_{CEr}(\boldsymbol{F}, \boldsymbol{A}, \boldsymbol{b}) = \nabla_{\boldsymbol{F}}\widehat{\mathcal{R}}_{CEr}(\boldsymbol{F}, \boldsymbol{A}, \boldsymbol{b}) = \nabla_{\boldsymbol{b}}\widehat{\mathcal{R}}_{CEr}(\boldsymbol{F}, \boldsymbol{A}, \boldsymbol{b}) = \boldsymbol{0} \right\}$$

$$\mathcal{C}_1 := \mathcal{C} \cap \left\{ \boldsymbol{F}, \boldsymbol{A}, \boldsymbol{b} : \left\| \nabla_{\boldsymbol{Z}=\boldsymbol{A}\boldsymbol{F}} \left( \frac{1}{N} \sum_{k=1}^{K} \sum_{i=1}^{n} \ell_{CE}(\boldsymbol{A}\boldsymbol{f}_{k,i} + \boldsymbol{b}, \boldsymbol{e}_k) \right) \right\|_2 \leq \sqrt{\lambda_{\boldsymbol{A}} \lambda_{\boldsymbol{F}}} \right\} \tag{26}$$

$$\mathcal{C}_2 := \mathcal{C} \cap \left\{ \boldsymbol{F}, \boldsymbol{A}, \boldsymbol{b} : \left\| \nabla_{\boldsymbol{Z}=\boldsymbol{A}\boldsymbol{F}} \left( \frac{1}{N} \sum_{k=1}^{K} \sum_{i=1}^{n} \ell_{CE}(\boldsymbol{A}\boldsymbol{f}_{k,i} + \boldsymbol{b}, \boldsymbol{e}_k) \right) \right\|_2 > \sqrt{\lambda_{\boldsymbol{A}} \lambda_{\boldsymbol{F}}} \right\}$$

Note that points in $\mathcal{C}_1$ already satisfy the global minima conditions based on equation 25. For $\mathcal{C}_2$, a stronger assumption of $m > K$ is needed to create a negative curvature direction for the hessian of $\widehat{\mathcal{R}}_{CEr}$ as follows:

$$\Delta = \left( -\left(\frac{\lambda_{\boldsymbol{A}}}{\lambda_{\boldsymbol{F}}}\right)^{1/4} \boldsymbol{w}\boldsymbol{v}^\top, \left(\frac{\lambda_{\boldsymbol{A}}}{\lambda_{\boldsymbol{F}}}\right)^{1/4} \boldsymbol{u}\boldsymbol{w}^\top, \boldsymbol{0} \right) \tag{27}$$

where $\boldsymbol{u} \in \mathbb{R}^K, \boldsymbol{v} \in \mathbb{R}^N$ are the left and right singular vectors corresponding to the largest singular value of $\nabla^2_{\boldsymbol{Z}=\boldsymbol{A}\boldsymbol{F}}(\frac{1}{N} \sum_{k=1}^{K} \sum_{i=1}^{n} \ell_{CE}(\boldsymbol{A}\boldsymbol{f}_{k,i} + \boldsymbol{b}, \boldsymbol{e}_k))$ and $\boldsymbol{w} \in \mathbb{R}^m$ is a non-zero vector such that $\boldsymbol{A}\boldsymbol{w} = \boldsymbol{0}$. Observe that $m > K$ is necessary to obtain a $\boldsymbol{w}$ in the null-space of $\boldsymbol{A}$ (see theorem 3.2 in Zhu et al. (2021)). Thus, stochastic optimizers can escape these strict saddle points along $\Delta$ and reach global minima that satisfy NC.

### 3.1.4 Role of (mean) squared error with regularization

**Lower bound:** similar to cross-entropy, we define the ERM for MSE with regularization as follows:

$$\min_{\boldsymbol{F},\boldsymbol{A},\boldsymbol{b}} \widehat{\mathcal{R}}_{MSEr}(\boldsymbol{F}, \boldsymbol{A}, \boldsymbol{b}) = \frac{1}{2N} \left\| \boldsymbol{A}\boldsymbol{F} + \boldsymbol{b}\boldsymbol{1}_N^\top - \boldsymbol{Y} \right\|_F^2 + \frac{\lambda_{\boldsymbol{A}}}{2} \left\| \boldsymbol{A} \right\|_F^2 + \frac{\lambda_{\boldsymbol{F}}}{2} \left\| \boldsymbol{F} \right\|_F^2 + \frac{\lambda_{\boldsymbol{b}}}{2} \left\| \boldsymbol{b} \right\|_2^2 \tag{28}$$

Where $\lambda_{\boldsymbol{A}}, \lambda_{\boldsymbol{F}}, \lambda_{\boldsymbol{b}} > 0$ are penalty terms. In a series of recent efforts, Han et al. (2021) analysed $\widehat{\mathcal{R}}_{MSEr}(\boldsymbol{F}, \boldsymbol{A}, \boldsymbol{b})$ when $\lambda_{\boldsymbol{F}} = 0$ (with bias $\boldsymbol{b}$ concatenated to $\boldsymbol{A}$) and Tirer & Bruna (2022) analysed the 'bias-free'($\boldsymbol{b} = \boldsymbol{0}$), 'unregularized-bias'($\lambda_{\boldsymbol{b}} = 0$) cases. For simplicity, we consider the $\boldsymbol{b} = \boldsymbol{0}$ case and present the lower bound given by Tirer & Bruna (2022) based on Jensen's inequality and strict convexity of $\|.\|_F^2$ as:

$$\begin{aligned}
\widehat{\mathcal{R}}_{MSEr}(\boldsymbol{F}, \boldsymbol{A}) &= \frac{1}{2N} \left\| \boldsymbol{A}\boldsymbol{F} - \boldsymbol{Y} \right\|_F^2 + \frac{\lambda_{\boldsymbol{A}}}{2} \left\| \boldsymbol{A} \right\|_F^2 + \frac{\lambda_{\boldsymbol{F}}}{2} \left\| \boldsymbol{F} \right\|_F^2 \\
&= \frac{1}{2Kn} \sum_{k=1}^{K} \frac{n}{n} \sum_{i=1}^{n} \left\| \boldsymbol{A}\boldsymbol{f}_{k,i} - \boldsymbol{e}_k \right\|_F^2 + \frac{\lambda_{\boldsymbol{A}}}{2} \left\| \boldsymbol{A} \right\|_F^2 + \frac{\lambda_{\boldsymbol{F}}}{2} \sum_{k=1}^{K} \frac{n}{n} \sum_{i=1}^{n} \left\| \boldsymbol{f}_{k,i} \right\|_2^2 \\
&\geq \frac{1}{2Kn} \sum_{k=1}^{K} n \left\| \boldsymbol{A}\frac{1}{n} \sum_{i=1}^{n} \boldsymbol{f}_{k,i} - \boldsymbol{e}_k \right\|_F^2 + \frac{\lambda_{\boldsymbol{A}}}{2} \left\| \boldsymbol{A} \right\|_F^2 + \frac{\lambda_{\boldsymbol{F}}}{2} \sum_{k=1}^{K} n \left\| \frac{1}{n} \sum_{i=1}^{n} \boldsymbol{f}_{k,i} \right\|_2^2
\end{aligned} \tag{29}$$

Where the final equality holds when NC1 is satisfied, i.e $\boldsymbol{f}_{k,1} = \cdots = \boldsymbol{f}_{k,n}$, leading to $\boldsymbol{F} = \overline{\boldsymbol{F}} \otimes \mathbf{1}_n^\top$. Here $\overline{\boldsymbol{F}} \in \mathbb{R}^{m \times K}$ is a matrix with class feature means $\overline{\boldsymbol{f}}_k = \frac{1}{n} \sum_{i=1}^{n} \boldsymbol{f}_{k,i}, \forall k \in [K]$ as columns. Under the assumption of balanced classes, Tirer & Bruna (2022) consider $\nabla_{\overline{\boldsymbol{F}}} \widehat{\mathcal{R}}_{MSEr}(\boldsymbol{F}, \boldsymbol{A}) = \nabla_{\boldsymbol{A}} \widehat{\mathcal{R}}_{MSEr}(\boldsymbol{F}, \boldsymbol{A}) = \boldsymbol{0}$ and obtain a closed form representation of $\boldsymbol{A} = \overline{\boldsymbol{F}}^\top (\overline{\boldsymbol{F}}\overline{\boldsymbol{F}}^\top + K\lambda_{\boldsymbol{A}} \boldsymbol{I}_m)^{-1}$. This formulation simplifies $\widehat{\mathcal{R}}_{MSEr}$ to solely depend on $\overline{\boldsymbol{F}}$, with the minimizer characterized by a flat spectrum: $\overline{\boldsymbol{F}}^\top \overline{\boldsymbol{F}} \propto \boldsymbol{I}_K$. The tight-frame obtained in this analysis is not a simplex ETF but an orthogonal frame. However, by centering the columns of $\overline{\boldsymbol{F}}$ around their mean $\overline{\boldsymbol{f}}_G := \frac{1}{K} \sum_{k=1}^{K} \overline{\boldsymbol{f}}_k$, we obtain the matrix: $\overline{\boldsymbol{F}} - \overline{\boldsymbol{f}}_G \mathbf{1}_K^\top$ which is indeed a simplex ETF. Note that $\overline{\boldsymbol{f}}_k, \overline{\boldsymbol{f}}_G$ are essentially $\boldsymbol{\mu}_k, \boldsymbol{\mu}_G$ (as per setup). Additionally, Tirer & Bruna (2022) showed that when $\boldsymbol{b} \neq \boldsymbol{0}, \lambda_{\boldsymbol{b}} = 0$, the closed form of bias for each class is $\boldsymbol{b}_k^* = \frac{1}{K} - \boldsymbol{a}_k^\top \boldsymbol{\mu}_G$. *Thus, the ideal bias turns out to be the global mean subtractor that was necessary for $\overline{\boldsymbol{F}}$ to form a simplex ETF.* A similar observation was made by Han et al. (2021) in their bias concatenated setup.

**Loss landscape:** Although the results by Saxe et al. (2013); Kawaguchi (2016); Freeman & Bruna (2016) have been influential, they don't deal with regularized settings for squared error. To this end, it is essential to characterize the deviations of critical points from global minima or strict saddles when regularization is introduced. By considering $\lambda_{\boldsymbol{F}} = \lambda_{\boldsymbol{b}} = 0$ and small values of $\lambda_{\boldsymbol{A}} \to 0^+$, Taghvaei et al. (2017) models this regularized ERM problem for linear networks as an optimal control problem (Farotimi et al., 1991) and show that not all local minimizers are global minimizers in the presence of regularization (see Mehta et al. (2021) for an empirical analysis). However, a second-order analysis of this regularized landscape is yet to be fully studied, especially the nature of critical points and their NC properties.

### 3.1.5   Does normalization facilitate collapse?

**Gradient flow perspective:** In the gradient flow analysis of the squared error without regularization, recall that it was necessary for $(\boldsymbol{F}, \boldsymbol{A})$ to lie along the sub-space $\mathcal{S}$ (in equation 20) to exhibit NC. In the regularized squared error setting with $\boldsymbol{b} = \boldsymbol{0}, \lambda_{\boldsymbol{F}} = 0$, Han et al. (2021) draw similar yet rigorous conclusions by decomposing the mean squared error into terms that depend on the least-squares optimal value of $\boldsymbol{A}$ (which is a function of $\boldsymbol{F}$), and the ones that capture the deviation of $\boldsymbol{A}$ from this optimal value. The terms which come under the former category are of particular interest. Formally, when $\boldsymbol{A}$ is set to the least squares optimal value $\boldsymbol{A}_{LS}$, let $\widetilde{\boldsymbol{F}} = \boldsymbol{F} - \boldsymbol{\mu}_G \mathbf{1}_N^\top \in \mathbb{R}^{m \times N}$, $\widetilde{\boldsymbol{M}} = [\boldsymbol{\mu}_1 - \boldsymbol{\mu}_G, \cdots, \boldsymbol{\mu}_K - \boldsymbol{\mu}_G] \in \mathbb{R}^{m \times K}$ and $\Sigma_{\widetilde{\boldsymbol{F}}} = \frac{1}{N} \widetilde{\boldsymbol{F}}\widetilde{\boldsymbol{F}}^\top \in \mathbb{R}^{m \times m}$, then Han et al. (2021) show that the parameter space of $\{(\boldsymbol{A}_{LS}, \widetilde{\boldsymbol{F}}) : \boldsymbol{A}_{LS} = \frac{1}{K} \widetilde{\boldsymbol{M}}^\top \Sigma_{\widetilde{\boldsymbol{F}}}^{-1}\}$ holds the following property for any symmetric full rank matrix $\boldsymbol{D} \in \mathbb{R}^{m \times m}$:

$$\frac{1}{K}(\boldsymbol{D}\widetilde{\boldsymbol{M}})^\top \big[(\boldsymbol{D}\widetilde{\boldsymbol{F}})(\boldsymbol{D}\widetilde{\boldsymbol{F}})^\top\big]^{-1} \boldsymbol{D}\widetilde{\boldsymbol{F}} = \frac{1}{K} \widetilde{\boldsymbol{M}}^\top \Sigma_{\widetilde{\boldsymbol{F}}}^{-1} \widetilde{\boldsymbol{F}} \tag{30}$$

The proof is a straightforward expansion of the transpose and inverse terms. This result implies that $\boldsymbol{A}_{LS}\widetilde{\boldsymbol{F}}$ is invariant to the transformation $\widetilde{\boldsymbol{F}} \to \boldsymbol{D}\widetilde{\boldsymbol{F}}$. Han et al. (2021) exploit the freedom to choose $\boldsymbol{D}$ and set $\boldsymbol{D} = \Sigma_W^{-1/2}$, resulting in 'renormalized' features $\boldsymbol{D}\widetilde{\boldsymbol{F}}$ (similar to 'whitened' features in statistical terms). By incorporating this continual renormalization ($\boldsymbol{N} = \boldsymbol{D}\widetilde{\boldsymbol{F}}$) into the gradient flow, they obtain:

$$\frac{d}{dt}\boldsymbol{N} = \Pi_{T_{\boldsymbol{N}}\mathcal{I}}\big(\nabla_{\boldsymbol{N}} \widehat{\mathcal{R}}_{MSEr-LS}(\boldsymbol{N})\big) \tag{31}$$

Where $\widehat{\mathcal{R}}_{MSEr-LS}(\boldsymbol{N})$ is the empirical risk pertaining to the parameter space of $(\boldsymbol{A}_{LS}, \widetilde{\boldsymbol{F}})$ and $\Pi_{T_{\boldsymbol{N}}\mathcal{I}}$ is a projection operator onto the tangent space of the manifold $\mathcal{I}$, of all identity-covariance features (refer Absil et al. (2009) for additional information on matrix manifolds and optimization). Informally, one can think of this operator as applying 'renormalization' at every step of the flow. Han et al. (2021) show in this setting that as $t \to \infty$, the non-zero singular values of $\Sigma_W^{-1/2}\widetilde{\boldsymbol{M}}$ tend to infinity while approaching equality. In simpler terms, the signal dominates the 'noise' (where 'noise' pertains to the deviation terms of $\boldsymbol{A}$ from $\boldsymbol{A}_{LS}$ during the gradient flow) and the limiting matrix of $(\Sigma_W^{-1/2}\widetilde{\boldsymbol{M}})^\top \in \mathbb{R}^{K \times m}$ is a simplex ETF. Thus, demonstrating the role of such renormalization steps in exhibiting NC. In addition to these intriguing theoretical properties, the benefits of such 'whitening' techniques have been widely studied and are typical in

modern-day deep learning settings (LeCun et al., 2012; Wiesler et al., 2014; Ioffe & Szegedy, 2015; Salimans & Kingma, 2016; Ulyanov et al., 2016; Ba et al., 2016; Wu & He, 2018). To further demonstrate the effects of normalization, we complement the UFM analysis with the results of Ergen & Pilanci (2021) for ReLU networks with batch-normalization and some minor additional calculations of our own in Appendix.A.4

### 3.1.6 Discussion

The simplicity of UFM allowed us to leverage the rich literature on matrix factorization, and optimization theory and identify the ideal configurations for $F, A, b$. In this section, we discuss additional properties and limitations of this modelling technique.

**Data independence:** In data-independent models such as the UFM, we are mainly concerned with $F, A, b$ and $Y$. As a consequence, under the balanced class assumption, networks can exhibit NC after sufficiently long training on completely random $(X, Y)$. This interpolating behaviour leading to NC properties was observed by Zhu et al. (2021) in canonical networks such as ResNet18 and MLP when trained on a randomly labelled CIFAR10 dataset (see figure 4). Thus, if a network has sufficient capacity to memorize the training data and reach TPT, we can expect its penultimate layer features and final layer weights to satisfy NC properties. To the contrary, experiments by Papyan et al. (2020) (see figure 5) show varying magnitude/extent of variance collapse depending on the complexity of data. For smaller data sets such as CIFAR10, a ResNet18 network attains a $\mathcal{NC}1$ value of $\approx 10^{-2}$, while for ImageNet, a ResNet152 network attains a $\mathcal{NC}1$ value of $\approx 1$. Thus, even after memorizing the training data, empirical results show deviation from the ideal UFM behaviour for larger, complex data sets. To understand the behaviour in figure 5, one needs to incorporate a notion of data complexity into the UFM, which is not straightforward as it goes against the premise on which the UFM is based. Instead, one can attempt to analyse the role of a large number of classes $K$ on the NC properties while enjoying the simplifications of UFM.

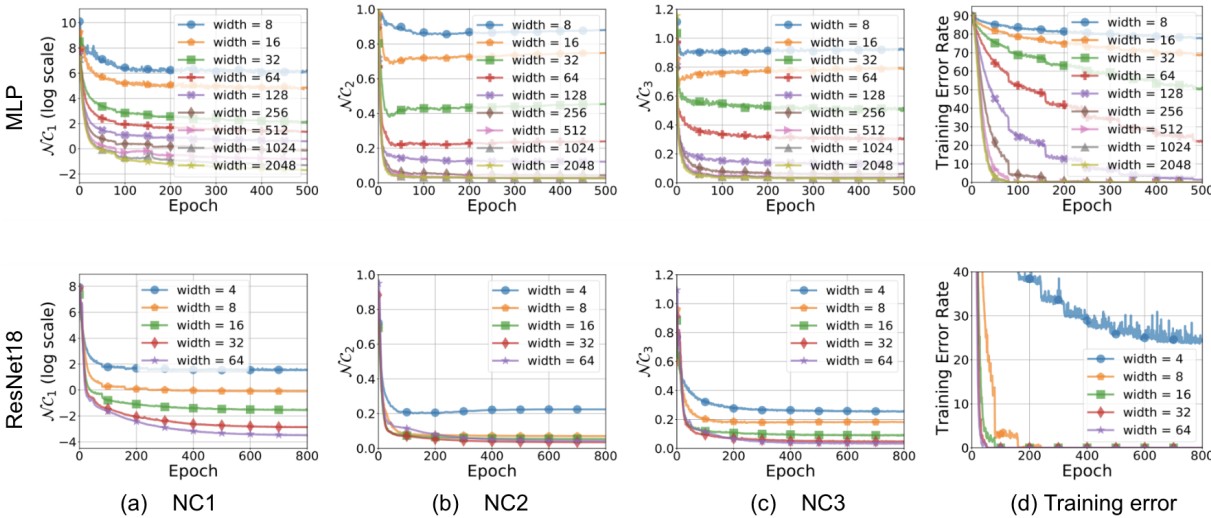

Figure 4: $\mathcal{NC}$ metrics of MLP and ResNet18 on a randomly labelled CIFAR10 dataset using cross-entropy loss. The width of the network is maintained across layers and varied across experiments. The first row corresponds to a 4-layer MLP, optimized using SGD with a learning rate 0.01 and weight decay $10^{-4}$. The second row corresponds to ResNet18, optimized using SGD with momentum 0.9, weight decay $5 \times 10^{-4}$, initial learning rate 0.05, decreased by a factor of 10 every 40 epochs. Image credit: Zhu et al. (2021).

**Implicit label dependence:** Unlike cross-entropy and (mean) squared error losses that explicitly require $Y$, contrastive losses such as Noise Contrastive Estimation based InfoNCE (Oord et al., 2018), Jensen-Shannon Divergence (JSD) (Lin, 1991) etc are independent of it. In the absence of labels, such losses aim to maximize the feature similarity of closely related training samples (for instance, of samples which inherently belong to the same class) while maximizing the dissimilarity with unrelated ones. A recent surge in unsupervised

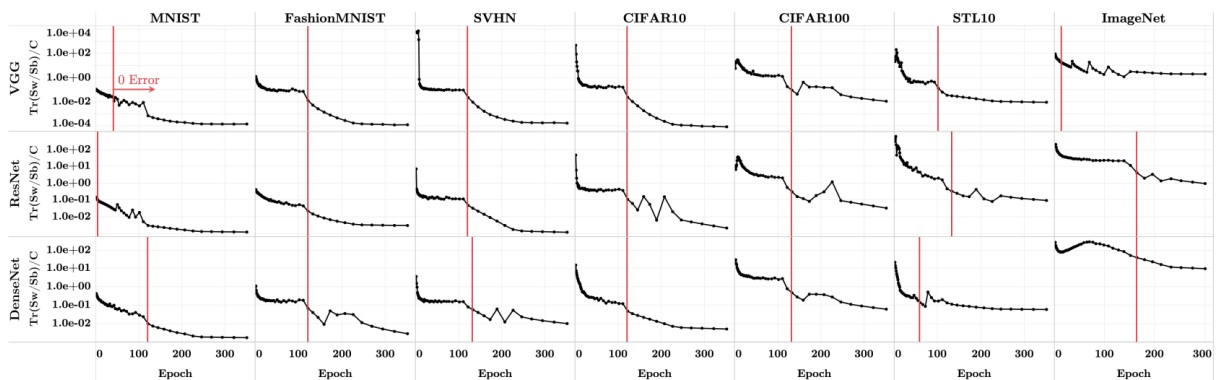

Figure 5: Plots of $\mathcal{NC}1$ (variability collapse) for combinations of data sets and canonical networks. VGG11, ResNet18 and DenseNet40 were chosen for MNIST and SVHN, VGG11, ResNet18 and DenseNet250 for FashionMNIST, VGG13, ResNet18, and DenseNet40 for CIFAR10, VGG13, ResNet50, and DenseNet250 for CIFAR100, VGG13, ResNet50, and DenseNet250 for STL10, VGG19, ResNet152, and DenseNet201 for ImageNet. The networks were trained using SGD with momentum 0.9 and weight decay of $10^{-4}$ for ImageNet and $5 \times 10^{-4}$ for other data sets. The learning rates were chosen by sweeping over logarithmically spaced values between $10^{-4}$ and 0.25 and the value resulting in the best test error was chosen. During the parameter search, the learning rates were reduced by a factor of 10 after $100, 200$ epochs for ImageNet (total=300 epochs) and $175, 265$ epochs for the rest (total=350 epochs). Image credit: Papyan et al. (2020).

representation learning can be attributed to the effectiveness of such objectives (Saunshi et al., 2019; Chen et al., 2020; Baevski et al., 2020; Jaiswal et al., 2020; Jing & Tian, 2020). However, with an unknown number of inherent classes, when $\boldsymbol{F}$ has a rank $m < K$, it is impossible for $\boldsymbol{F}$ to form a K-simplex. Nevertheless, the supervised contrastive learning (Khosla et al., 2020) settings provide interesting insights on NC where the label information is implicitly used in the objective:

$$\min_{\boldsymbol{F}} \widehat{\mathcal{R}}_{CL} = \frac{1}{N} \sum_{k=1}^{K} \sum_{i=1}^{n} -\frac{1}{n} \sum_{j=1}^{n} \log \left( \frac{\exp(\boldsymbol{f}_{k,i}^{\top} \boldsymbol{f}_{k,j}/\tau)}{\sum_{k'=1}^{K} \sum_{i'=1}^{n} \exp(\boldsymbol{f}_{k,i}^{\top} \boldsymbol{f}_{k',i'}/\tau)} \right) \tag{32}$$

Where $\tau > 0$ is known as the 'temperature' parameter, which controls the hardness of negative samples (Wang & Liu, 2021). Owing to its similarity with $\widehat{\mathcal{R}}_{CE}$, Fang et al. (2021) obtained a lower bound for $\widehat{\mathcal{R}}_{CL}$ as follows:

$$\frac{1}{N} \sum_{k=1}^{K} \sum_{i=1}^{n} -\frac{1}{n} \sum_{j=1}^{n} \log \left( \frac{\exp(\boldsymbol{f}_{k,i}^{\top} \boldsymbol{f}_{k,j}/\tau)}{\sum_{k'=1}^{K} \sum_{i'=1}^{n} \exp(\boldsymbol{f}_{k,i}^{\top} \boldsymbol{f}_{k',i'}/\tau)} \right) \geq -\frac{c_2 K \Omega_{\boldsymbol{F}}}{(c_1 + c_2)(K-1)\tau} + c_3 + \log n \tag{33}$$

Where $c_1 = \exp(\sqrt{\Omega_{\boldsymbol{A}}\Omega_{\boldsymbol{F}}}), c_2 = (K-1)\exp(-\sqrt{\Omega_{\boldsymbol{A}}\Omega_{\boldsymbol{F}}}), c_3 = \frac{c_1}{(c_1+c_2)} \log(\frac{c_1+c_2}{c_1}) + \frac{c_2}{(c_1+c_2)} \log(\frac{(c_1+c_2)(K-1)}{c_2})$, $\frac{1}{K} \sum_{k=1}^{K} \|\boldsymbol{a}_k\|_2^2 \leq \Omega_{\boldsymbol{A}}, \frac{1}{Kn} \sum_{k=1}^{K} \sum_{i=1}^{n} \|\boldsymbol{f}_{k,i}\|_2^2 \leq \Omega_{\boldsymbol{F}}$. Similar to our previous observations, equality is attained when variance collapse occurs and the columns formed by class means in optimal $\boldsymbol{F}$ resembles a simplex ETF (Fang et al., 2021). *As a takeaway, observe that even without explicit label matrix $\boldsymbol{Y}$, a loss function which promotes variability collapse and maximum separation leads to neural collapse based solutions.*

**Class imbalance:** Assuming an equal number of training examples for all the classes has been critical for analysis till now. Having $n_1 = n_2 = \cdots, n_K = n = N/K$ gives a symmetric structure to $\boldsymbol{Y}$ in the form of $\boldsymbol{Y} = \boldsymbol{I}_K \otimes \boldsymbol{1}_n^{\top}$ which results in a relatively easier derivation of variance collapse and simplex ETF. When the classes are imbalanced, the analysis is not straightforward. By considering $\ell$ to be any convex loss function,

the ERM objective with norm constraints can be given by:

$$\min_{\boldsymbol{F},\boldsymbol{A}} \frac{1}{N} \sum_{k=1}^{K} \sum_{i=1}^{n_k} \ell(\boldsymbol{A}\boldsymbol{f}_{k,i}, \boldsymbol{e}_k) \,, s.t \,\, \frac{1}{K} \sum_{k=1}^{K} \|\boldsymbol{a}_k\|_2^2 \le \Omega_{\boldsymbol{A}}, \frac{1}{K} \sum_{k=1}^{K} \frac{1}{n_k} \sum_{i=1}^{n_k} \|\boldsymbol{f}_{k,i}\|_2^2 \le \Omega_{\boldsymbol{F}} \tag{34}$$

Without loss of generality, one can consider the cross-entropy loss and analyze the ERM by performing a convex relaxation into a semi-definite program. Fang et al. (2021) analyse such a convex program by considering a subset of CIFAR10 dataset with $K_{maj}$ majority classes such that $n_1 = n_2 \cdots = n_{K_{maj}} = n_{maj}$ and $K_{min}$ minority classes such that $n_{K_{maj}+1} = \cdots = n_K = n_{min}$. By defining a class imbalance ratio of $r_{ib} = n_{maj}/n_{min} > 1$, the authors empirically observed that when $r_{ib} \ge t_0$, for some threshold $t_0 > 1$, the average angle between the minority class classifiers becomes zero, i.e, the rows of $A$ pertaining to these minority classes collapse to a single vector. The authors term this phenomenon 'Minority Collapse'. Additionally, they observed that the threshold $t_0$ tends to get smaller (larger) with smaller (larger) $\Omega_{\boldsymbol{A}}, \Omega_{\boldsymbol{F}}, K_{min}$. Intuitively, when the constraints $\Omega_{\boldsymbol{A}}, \Omega_{\boldsymbol{F}}$ are tighter, observe from equation 34 that majority classes ($K_{maj}$) dominate the objective and there is a little 'budget' in the gradient updates for data in $K_{min}$ minority classes. In a formal sense, let's consider the gradient of cross-entropy loss w.r.t $\boldsymbol{a}_k, k \in [K]$:

$$\frac{\partial \ell_{CE}}{\partial \boldsymbol{a}_k} = \underbrace{\sum_{i=1}^{n_k} \boldsymbol{f}_{k,i} \left( \frac{\exp(\boldsymbol{a}_k^\top \boldsymbol{f}_{k,i})}{\sum_{k'=1}^{K} \exp(\boldsymbol{a}_{k'}^\top \boldsymbol{f}_{k,i})} - 1 \right)}_{\text{``pull''}} + \underbrace{\sum_{k' \neq k}^{K} \sum_{j=1}^{n_{k'}} \boldsymbol{f}_{k',j} \frac{\exp(\boldsymbol{a}_k^\top \boldsymbol{f}_{k',j})}{\sum_{k''=1}^{K} \exp(\boldsymbol{a}_{k''}^\top \boldsymbol{f}_{k',j})}}_{\text{``push''}} \tag{35}$$

The "pull" term represents the tendency of $\boldsymbol{a}_k$ to move towards features of the same class while the "push" term represents the tendency to move away from them (see Yang et al. (2022) for analysis based on this formulation). In the case of minority classes, the "push" term dominates the gradient, potentially leading to minority collapse. The manifestation of minority collapse was even observed in ResNet18 networks on CIFAR10, FashionMNIST data sets for sufficiently large $r_{ib}$. In practical settings, one way of avoiding this state is to oversample data from the minority classes or under-sample from the majority class to decrease $r_{ib}$ (Drummond et al., 2003; Zhou & Liu, 2005; He & Garcia, 2009; Huang et al., 2016; Buda et al., 2018; Johnson & Khoshgoftaar, 2019; Cui et al., 2019; Fang et al., 2021). Alternatively, when we are aware of the imbalance, fixing the last layer classifier to the desired simplex ETF seems like a clever hack to prevent minority collapse. Yang et al. (2022) confirm this intuition and achieve improved performance even in fine-grained image classification tasks.

**Extensibility:** Extending the UFM with multiple non-linear layers quickly turns a tractable model into an involved one. Thus, a good starting point in this direction is to add a single linear layer and analyse the model properties. To this end, consider the following ERM based on MSE with regularization and an extra linear layer as proposed by Tirer & Bruna (2022):

$$\min_{\boldsymbol{F},\boldsymbol{A}_1,\boldsymbol{A}_2} \widehat{\mathcal{R}}_{MSE-ext} = \frac{1}{2N} \|\boldsymbol{A}_2\boldsymbol{A}_1\boldsymbol{F} - \boldsymbol{Y}\|_F^2 + \frac{\lambda_{\boldsymbol{A}_2}}{2} \|\boldsymbol{A}_2\|_F^2 + \frac{\lambda_{\boldsymbol{A}_1}}{2} \|\boldsymbol{A}_1\|_F^2 + \frac{\lambda_{\boldsymbol{F}}}{2} \|\boldsymbol{F}\|_F^2 \tag{36}$$

Where $\boldsymbol{F} \in \mathbb{R}^{m \times N}$ are the unconstrained features, $\boldsymbol{A}_1 \in \mathbb{R}^{m \times m}, \boldsymbol{A}_2 \in \mathbb{R}^{K \times m}$ are the linear layer weights and $\lambda_{\boldsymbol{A}_2}, \lambda_{\boldsymbol{A}_1}, \lambda_{\boldsymbol{F}} > 0$ are penalty terms. Tirer & Bruna (2022) lower bound this risk by following the same sketch as equation 29:

$$\begin{aligned}
&\frac{1}{2N} \|\boldsymbol{A}_2\boldsymbol{A}_1\boldsymbol{F} - \boldsymbol{Y}\|_F^2 + \frac{\lambda_{\boldsymbol{A}_2}}{2} \|\boldsymbol{A}_2\|_F^2 + \frac{\lambda_{\boldsymbol{A}_1}}{2} \|\boldsymbol{A}_1\|_F^2 + \frac{\lambda_{\boldsymbol{F}}}{2} \|\boldsymbol{F}\|_F^2 \\
&= \frac{1}{2Kn} \sum_{k=1}^{K} \frac{n}{n} \sum_{i=1}^{n} \|\boldsymbol{A}_2\boldsymbol{A}_1\boldsymbol{f}_{k,i} - \boldsymbol{e}_k\|_F^2 + \frac{\lambda_{\boldsymbol{A}_2}}{2} \|\boldsymbol{A}_2\|_F^2 + \frac{\lambda_{\boldsymbol{A}_1}}{2} \|\boldsymbol{A}_1\|_F^2 + \frac{\lambda_{\boldsymbol{F}}}{2} \sum_{k=1}^{K} \frac{n}{n} \sum_{i=1}^{n} \|\boldsymbol{f}_{k,i}\|_2^2 \\
&\ge \frac{1}{2Kn} \sum_{k=1}^{K} n \left\| \boldsymbol{A}_2\boldsymbol{A}_1 \frac{1}{n} \sum_{i=1}^{n} \boldsymbol{f}_{k,i} - \boldsymbol{e}_k \right\|_F^2 + \frac{\lambda_{\boldsymbol{A}_2}}{2} \|\boldsymbol{A}_2\|_F^2 + \frac{\lambda_{\boldsymbol{A}_1}}{2} \|\boldsymbol{A}_1\|_F^2 + \frac{\lambda_{\boldsymbol{F}}}{2} \sum_{k=1}^{K} n \left\| \frac{1}{n} \sum_{i=1}^{n} \boldsymbol{f}_{k,i} \right\|_2^2
\end{aligned} \tag{37}$$

Where the final equality holds when within-class variability in $\boldsymbol{F}$ is 0, i.e, $\boldsymbol{F} = \overline{\boldsymbol{F}} \otimes \mathbf{1}_n^\top, \overline{\boldsymbol{F}} \in \mathbb{R}^{m \times K}$ (similar to the $\widehat{\mathcal{R}}_{MSEr}$ case). The authors connect this three-factor minimization problem with two-factor objectives based on the result in equation 24 and split the risk into two sub-problems as follows:

$$
\begin{aligned}
\widehat{\mathcal{R}}_{MSE-ext1} &= \min_{\boldsymbol{A}_2, \boldsymbol{Z}_{\boldsymbol{A}_1 \overline{\boldsymbol{F}}}} \frac{1}{2K} \left\| \boldsymbol{A}_2 \boldsymbol{Z}_{\boldsymbol{A}_1 \overline{\boldsymbol{F}}} - \boldsymbol{I}_K \right\|_F^2 + \frac{\lambda_{\boldsymbol{A}_2}}{2} \left\| \boldsymbol{A}_2 \right\|_F^2 + \sqrt{n \lambda_{\boldsymbol{A}_1} \lambda_{\boldsymbol{F}}} \left\| \boldsymbol{Z}_{\boldsymbol{A}_1 \overline{\boldsymbol{F}}} \right\|_* \\
\widehat{\mathcal{R}}_{MSE-ext2} &= \min_{\boldsymbol{Z}_{\boldsymbol{A}_2 \boldsymbol{A}_1}, \overline{\boldsymbol{F}}} \frac{1}{2K} \left\| \boldsymbol{Z}_{\boldsymbol{A}_2 \boldsymbol{A}_1} \overline{\boldsymbol{F}} - \boldsymbol{I}_K \right\|_F^2 + \frac{n \lambda_{\boldsymbol{F}}}{2} \left\| \overline{\boldsymbol{F}} \right\|_F^2 + \sqrt{\lambda_{\boldsymbol{A}_2} \lambda_{\boldsymbol{A}_1}} \left\| \boldsymbol{Z}_{\boldsymbol{A}_2 \boldsymbol{A}_1} \right\|_*
\end{aligned}
\tag{38}
$$

Where sub-problems $\widehat{\mathcal{R}}_{MSE-ext1}, \widehat{\mathcal{R}}_{MSE-ext2}$ have a close resemblance to the one layer UFM risk formulation. If $(\boldsymbol{F}^*, \boldsymbol{A}_1^*, \boldsymbol{A}_2^*)$ is the global minimizer of $\widehat{\mathcal{R}}_{MSE-ext}$ such that $\boldsymbol{F}^* = \overline{\boldsymbol{F}} \otimes \mathbf{1}_n^\top$, then:

$$
(\boldsymbol{A}_2^* \boldsymbol{A}_1^*) \overline{\boldsymbol{F}} \propto \overline{\boldsymbol{F}}^\top \overline{\boldsymbol{F}} \propto (\boldsymbol{A}_2^* \boldsymbol{A}_1^*)(\boldsymbol{A}_2^* \boldsymbol{A}_1^*)^\top \propto \boldsymbol{I}_K
\tag{39}
$$

Thus, $\overline{\boldsymbol{F}}$ collapses to an orthogonal frame (to a simplex ETF if we recenter around the column means) and is aligned with $(\boldsymbol{A}_2^* \boldsymbol{A}_1^*)^\top$. Similar results can be shown for $\boldsymbol{A}_2^*$ and $\boldsymbol{A}_1^* \boldsymbol{F}^*$ (see theorem 4.1 in Tirer & Bruna (2022) for proofs). Now, by converting the linear layer $\boldsymbol{A}_1 \boldsymbol{F}$ into $\sigma(\boldsymbol{A}_1 \boldsymbol{F})$, where $\sigma$ represents a non-linear activation (such as ReLU), we are presented with a non-linear UFM model. Factorization of $\boldsymbol{A}_2 \sigma(\boldsymbol{A}_1 \boldsymbol{F})$ into 2 factors is possible in the case of $\widehat{\mathcal{R}}_{MSE-ext1}$ defined above but not for $\widehat{\mathcal{R}}_{MSE-ext2}$. Tirer & Bruna (2022) analyse the former by considering ReLU activations as a non-negativity constraint on $\boldsymbol{A}_1 \boldsymbol{F}$ and proceed with the 2-factor problem. Refer to theorem 4.2 and Appendix.E in Tirer & Bruna (2022) for further details.

## 3.2 Models of "Locally Elastic" Networks

Theoretically modelling the training dynamics in neural networks has been a long-standing challenge and is mostly tackled in the shallow settings (Jacot et al., 2018; Arora et al., 2019; Goldt et al., 2019; Yang, 2019; Hu et al., 2020) or in linear networks (Saxe et al., 2013; Kawaguchi, 2016; Ji & Telgarsky, 2018; Arora et al., 2018; Lampinen & Ganguli, 2018). Nevertheless, with an understanding of the 'desired' structures that a sufficiently expressive network should achieve, we transition to a modelling technique which attempts to imitate the feature separation dynamics in canonical deep classifier neural networks and demonstrates neural collapse as a by-product.

### 3.2.1 Primer

**Local Elasticity (LE):** Introduced in the work of He & Su (2019), local elasticity is a phenomenon which describes classifiers whose prediction of a training sample $\boldsymbol{x}_i^k, i \in [n], k \in [K]$, is insignificantly affected by SGD updates pertaining to gradients of dissimilar samples $\boldsymbol{x}_i^{k'}, i \in [n], k' \neq k \in [K]$. *Intuitively, local elasticity represents the influence that training samples have on each other during training* (see figure 6). Analysis along these lines can be traced back to the seminal work on influence functions and curves by Hampel (1974), followed by efforts in developing robust statistical models (Reid & Crépeau, 1985; Weisberg, 2005; Huber, 2011; Kalbfleisch & Prentice, 2011; Koh & Liang, 2017). While influence functions have been widely adopted in the machine learning community for studying interpretable learning techniques (Adadi & Berrada, 2018; Molnar, 2020), we restrict our focus to locally elastic networks and their training dynamics. He & Su (2019) present a preliminary analysis of this idea in image classification settings and a geometric interpretation using the Neural Tangent Kernel (NTK).

**Stochastic Differential Equations (SDE):** Stochastic differential equations are statistical variants of classical differential equations and can be traced back to the works of Itô (1951); Van Kampen (1976); Kloeden & Platen (1992). Their presence can be found in a wide range of important settings such as filtering (Welch et al., 1995), boundary value problems (including the influential 'Dirichlet problem'), the Fokker-Plank equation (Risken, 1996), the Black-Scholes model (Karoui et al., 1998), Ornstein-Uhlenbeck processes (Ikeda & Watanabe, 2014) and many more. From a deep learning perspective, since we are modelling the impact of SGD updates on the gradual separability of features, the idea is to represent these changes as a stochastic differential equation (Li et al., 2021; Zhang et al., 2021b) and study its implications.

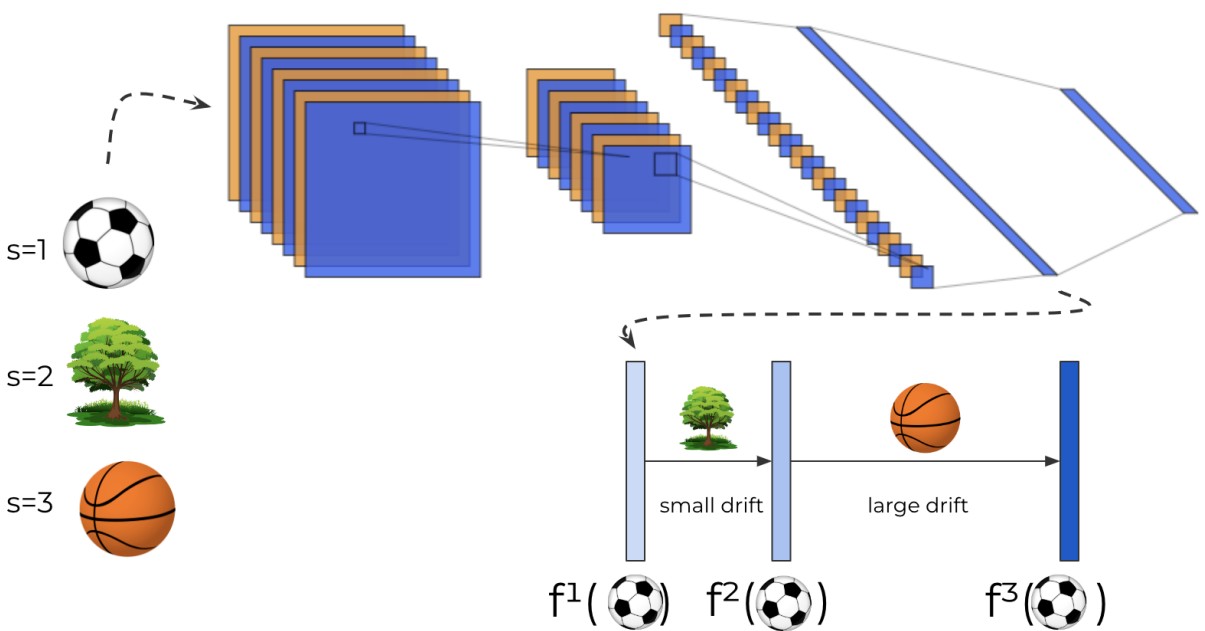

Figure 6: Illustration of the local elasticity phenomenon in neural networks with a toy example. At step $s = 1$, the image of a football is passed to the network, resulting in its penultimate layer representation $f^1$. In the next step $s = 2$, the image of a tree is passed, which is dissimilar to the football and results in a small feature drift in the learnt representation of a football. Finally, when an image of a basketball is passed to the network at step $s = 3$, the drift from $f^2$ to $f^3$ will be larger than $f^1$ to $f^2$ due to visual similarity.

### 3.2.2 Feature separation via Locally elastic stochastic differential equations (LE-SDE)

**Intuition:** To provide an intuitive understanding of this model, we take a bottom-up approach to present the ideas. Recall that the penultimate layer features for the $i^{th}$ data point of class $k$ are given by $\boldsymbol{f}_{k,i} \in \mathbb{R}^m$. We extend the notation to denote this feature at iteration/step $s$ of training as $\boldsymbol{f}_{k,i}^s$. Without loss of generality, by randomly sampling $\boldsymbol{x}_{i'}^{k'}, i' \sim Unif([n]), k' \sim Unif([K])$ at iteration $s$, our goal is to model the impact of training a network with $\boldsymbol{x}_{i'}^{k'}$ on $\boldsymbol{f}_{k,i}$. A reasonable formulation can be given by:

$$\boldsymbol{f}_{k,i}^s - \boldsymbol{f}_{k,i}^{s-1} = E^s \boldsymbol{f}_{k',i'}^{s-1} + \phi^{s-1}(\boldsymbol{x}_i^k) \tag{40}$$

This indicates that the 'drift' in features $(\boldsymbol{f}_{k,i}^s - \boldsymbol{f}_{k,i}^{s-1})$ is proportional to $\boldsymbol{f}_{k',i'}^{s-1}$, scaled by some impact term $E^s$, plus data dependent noise $\phi^{s-1}(\boldsymbol{x}_i^k)^3$.

**LE-SDE Model:** The work of Zhang et al. (2021b) formally captures this intuitive idea in their LE-SDE model. Since back-propagation iteratively updates the features, Zhang et al. (2021b) capture the impact of learning rate as well as transformations of $\boldsymbol{f}_{k',i'}^{s-1}$ on $\boldsymbol{f}_{k,i}^s$. Thus, the formal and refined version of 'drift' is presented by the authors as:

$$\boldsymbol{f}_{k,i}^s - \boldsymbol{f}_{k,i}^{s-1} = \eta(\boldsymbol{E}^s)_{kk'}(\boldsymbol{T}^s)_{kk'} \boldsymbol{f}_{k',i'}^{s-1} + \sqrt{\eta}\phi^{s-1}(\boldsymbol{x}_i^k) \tag{41}$$

Where $\eta$ is the step size, $\phi^{s-1}(\boldsymbol{x}_i^k)$ is the Gaussian noise associated with the data point $\boldsymbol{x}_i^k$ (independent of it's feature $\boldsymbol{f}_{k,i}^{s-1}$). $\boldsymbol{E}^s \in \mathbb{R}^{K \times K}$ is the local elasticity impact matrix at iteration $s$ and $(\boldsymbol{E}^s)_{kk'} \in \mathbb{R}$ is the $(k, k')^{th}$ entry of $\boldsymbol{E}^s$ which represents the LE impact that $\boldsymbol{x}_{i'}^{k'}$ has on $\boldsymbol{x}_i^k$. Similarly $(\boldsymbol{T}^s)_{kk'} \in \mathbb{R}^{m \times m}$

---

[3]Observe that this formulation allows us to track the separability of features in the pre-TPT phases as well.

is the transformation matrix on features at iteration $s$. Next, by considering $\tilde{\boldsymbol{f}}_k^s$ to be a random sample from class k (can be informally thought of as a representative sample as well), the authors represent $\widetilde{\boldsymbol{F}}^s = [\tilde{\boldsymbol{f}}_1^s | \cdots | \tilde{\boldsymbol{f}}_K^s] \in \mathbb{R}^{Km}$ to be a concatenation of per-class representative features and $\overline{\boldsymbol{F}}^s = [\boldsymbol{\mu}_1^s | \cdots | \boldsymbol{\mu}_K^s] \in \mathbb{R}^{Km}$ as the concatenation of per-class feature means at iteration $s$. This assumption and setup follows from NC1 where $\tilde{\boldsymbol{f}}_k^s$ eventually collapses to $\boldsymbol{\mu}_k^s$. Thus, a single representative data point for each class is amenable for analysis. Now, the continuous version of the SDE in equation 41 with $t = s\eta, \eta \to 0$ can be represented as:

$$d\widetilde{\boldsymbol{F}}^t = \boldsymbol{B}^t \overline{\boldsymbol{F}}^t dt + (\Sigma^t)^{1/2} d\boldsymbol{W}^t \tag{42}$$

This is the **LE-SDE** formulation where $\boldsymbol{W}^t$ represents a standard Wiener process $\in \mathbb{R}^{Km}$, $\Sigma^t \in \mathbb{R}^{Km \times Km}$ is the covariance matrix of representative features and $\boldsymbol{B}^t \in \mathbb{R}^{Km \times Km}$ is a $K \times K$ block matrix with $m \times m$ sized blocks, which models the combined effect of LE impact $\boldsymbol{E}^t$ and transformations $\boldsymbol{T}^t$. The $(k, k')^{th}$ block of $\boldsymbol{B}^t$ is $(\boldsymbol{E}^t)_{kk'}(\boldsymbol{T}^t)_{kk'}/K, \forall k, k' \in [K]$. To account for randomness involved in selecting the representative samples $\widetilde{\boldsymbol{F}}^t$, the class means $\overline{\boldsymbol{F}}^t$ satisfy:

$$\frac{d}{dt}(\overline{\boldsymbol{F}}^t) = \boldsymbol{B}^t \overline{\boldsymbol{F}}^t \tag{43}$$

This is obtained by taking expectation on both sides of equation 42 and noting that the Wiener process can be characterized as a martingale with $\boldsymbol{W}_0 = \boldsymbol{0}$. This implies $\mathbb{E}[\boldsymbol{W}^t] = \boldsymbol{0}$, resulting in equation 43. Zhang et al. (2021b) call it the **LE-ODE**. With this setup in place, the authors assume the diagonal entries of $\boldsymbol{E}^t$ to be $\alpha_t$ and the off-diagonal entries to be $\beta_t$. Here $\alpha_t, \beta_t \in \mathbb{R}$ pertain to "intra-class" and "inter-class" LE impacts respectively. Now, as $t \to \infty$, when $\nu_t = \min\{\alpha_t - \beta_t, \alpha_t + (K-1)\beta_t\} > 0$, and $\boldsymbol{T}$ is a positive semi-definite matrix with positive diagonal entries, then Theorem 3.1 in Zhang et al. (2021b) states that:

- The features are separable with a probability $p \to 1$ when $\nu_t = \omega(1/t)$

- The features are asymptotically pairwise separable with a probability $p \to 0$ when $\nu_t = o(1/t)$ and $n \to \infty$ at an arbitrarily slow rate.

Here, pairwise separation at any time $t$ implies, for $1 \leq k < k' \leq K$, there exists a direction $\boldsymbol{v}_{k,k'}^t$ such that:

$$\min_i \langle \boldsymbol{v}_{k,k'}^t, \boldsymbol{f}_{k,i}^t \rangle > \max_j \langle \boldsymbol{v}_{k,k'}^t, \boldsymbol{f}_{k',j}^t \rangle \tag{44}$$

Similarly, asymptotically pairwise separable implies:

$$P\big(\min_i \langle \boldsymbol{v}_{k,k'}^t, \boldsymbol{f}_{k,i}^t \rangle > \max_j \langle \boldsymbol{v}_{k,k'}^t, \boldsymbol{f}_{k',j}^t \rangle\big) \to 1 \tag{45}$$

### 3.2.3 Neural collapse as a by-product

A powerful yet simplified aspect of this model is the freedom to choose $\boldsymbol{T}$. This flexibility was exploited by the authors by setting it as the outer product of residuals $\boldsymbol{d}_j \boldsymbol{d}_j^\top$, where residual roughly aligns along $\boldsymbol{d}_j = \boldsymbol{e}_j - \frac{1}{K}\boldsymbol{1}_m, \forall j \in [K]$. *Intuitively, these residuals roughly indicate the direction in which $\tilde{\boldsymbol{f}}_j$ need to be pushed for perfect classification.* Thus, by setting:

$$(\boldsymbol{T})_{ij} = \frac{\boldsymbol{d}_j \boldsymbol{d}_j^\top}{\|\boldsymbol{d}_j\|_2^2} \in \mathbb{R}^{m \times m}, \text{ where } \boldsymbol{d}_j = \boldsymbol{e}_j - \frac{1}{K}\boldsymbol{1}_m \in \mathbb{R}^m, j \in [K] \tag{46}$$

the transformation $\boldsymbol{T}$ always aligns the changes in $\tilde{\boldsymbol{f}}_j$ along $\boldsymbol{d}_j$ for all $i \in [K]$. Formally, by considering the "intra-class" and "inter-class" LE impacts $\alpha_t, \beta_t$ to be constants $\alpha, \beta$ and $\boldsymbol{B} = \frac{1}{K}(\boldsymbol{E} \otimes \boldsymbol{I}_K) \odot \boldsymbol{T}$, where $\odot$ represents the Hadamard product, $(\boldsymbol{E} \otimes \boldsymbol{I}_K) \odot \boldsymbol{T}$ has eigen values $\{\alpha - \beta, \alpha + \frac{\beta}{K-1}, 0\}$ with multiplicities $\{1, K(K-1), K-1\}\}$ respectively. Additionally, by assuming $m = K$ for satisfying the psd property of the transformation matrix and using these quantities to solve the LE-ODE in equation 43, the authors get:

$$\overline{\boldsymbol{F}}^t = \boldsymbol{c}_0 + \tau_1 \boldsymbol{d} e^{\frac{1}{K}(A_t - B_t)} + \left( \sum_{l=1}^{K-1} \tau_{2l} \boldsymbol{u}_l \right) e^{\frac{1}{K}(A_t + \frac{1}{K-1} B_t)} \tag{47}$$

Where $\boldsymbol{d}$ and $\boldsymbol{u}_l \in \mathbb{R}^{K^2}, l \in [K-1]$ are the eigen vectors of $(\boldsymbol{E} \otimes \boldsymbol{I}_K) \odot \boldsymbol{T}$ with eigen values $\alpha - \beta, \alpha + \frac{\beta}{K-1}$ respectively, $\boldsymbol{c}_0 \in \mathbb{R}^{K^2}, \tau_1, \tau_{2l} \in \mathbb{R}, l \in [K-1]$ are constants and $A_t = \int_0^t \alpha d\tau, B_t = \int_0^t \beta d\tau$. With the assumption of local elasticity ($\nu_t > 0$), $B_t < 0$, as $t \to \infty$, $\overline{\boldsymbol{F}}^t$ will eventually align towards $\boldsymbol{v}$. Since $\boldsymbol{d}$ is a concatenation of residuals $\boldsymbol{d}_j, j \in [K]$ from equation 46, the matrix formed by $\boldsymbol{d}_j$'s as columns forms a simplex ETF (refer Appendix.C.2.4 in Zhang et al. (2021b) for details of the proof). Thus, after a certain point in time $t \geq t_0$ of evaluating the LE-ODE, the class means $\boldsymbol{\mu}_k^t, \forall k \in [K]$ which were evolving through the concatenated matrix $\overline{\boldsymbol{F}}^t$, tend to a simplex ETF as $t \to \infty$.

### 3.2.4 Discussion

The LE-SDE/ODE approach implicitly captures the impact of data complexity on the feature separation dynamics and provides a unique way of modelling feature evolution. Note that the purpose of this modelling technique is not to approximate the actual non-linear dynamics of deep classifier neural networks, but to mimic the dynamics of the LE phenomenon during training. The key takeaway here is that by choosing the LE transformations to be biased towards an orthogonal structure of labels $\boldsymbol{e}_j, j \in [K]$, neural collapse is manifested as a by-product. For additional results and experiments pertaining to the study of LE, please refer to He & Su (2019); Zhang et al. (2021b). As we have already observed the effects of realignment in our analysis of normalization and UFM, the LE-SDE/ODE model reinforces the role of continuous realignment towards a maximally separable configuration for facilitating NC. As a follow-up of this observation let's consider the Taylor approximation of feature drifts as presented in Zhang et al. (2021b) to better understand the realignment behaviour. As a first step, assuming that a network is being trained using the cross-entropy loss, and considering the 'logits-as-features' model of Zhang et al. (2021b), observe that the drift in features can be approximately given by:

$$\boldsymbol{f}_{k,i}^s - \boldsymbol{f}_{k,i}^{s-1} \approx \eta \left[ \boldsymbol{G}_k^s \big( \boldsymbol{e}_{k'} - \sigma_{softmax}(\boldsymbol{f}_{k',i'}^{s-1}) \big) \right] \tag{48}$$

Where $\boldsymbol{G}_k^s = \frac{\partial \boldsymbol{f}_{k,i}^{s-1}}{\partial \theta} {\frac{\partial \boldsymbol{f}_{k',i'}^{s-1}}{\partial \theta}}^{\top} \in \mathbb{R}^{K \times K}$ is the time dependent gram matrix for class $k$, trainable parameters $\theta$ and softmax function $\sigma_{softmax}$. Recall that we assumed $m = K$ for the transformation matrix to be psd. With this approximation, the feature drift $D(\widetilde{\boldsymbol{F}}^t, t)$ can be defined for all representative features $\widetilde{\boldsymbol{F}}^t$ when $t = s\eta, \eta \to 0$ as:

$$D(\widetilde{\boldsymbol{F}}^t, t) = \boldsymbol{G}^t \left( \left[ (\boldsymbol{e}_1 - \sigma_{softmax}(\tilde{\boldsymbol{f}}_1^t))^{\top}, \cdots, (\boldsymbol{e}_k - \sigma_{softmax}(\tilde{\boldsymbol{f}}_K^t))^{\top} \right]^{\top} \right) \tag{49}$$

Where $\boldsymbol{G}^t \in \mathbb{R}^{K^2 \times K^2}$ is the gram matrix of all classes. As the actual dynamics of separation are non-linear, the LE-SDE attempts to model it by linearizing the drift around the mean value of $\overline{\boldsymbol{F}}^t$ (see Särkkä & Solin (2019) for a similar linearization approach) to obtain:

$$D(\widetilde{\boldsymbol{F}}^t, t) \approx D(\overline{\boldsymbol{F}}^t, t) + \nabla_{\widetilde{\boldsymbol{F}}^t} D(\overline{\boldsymbol{F}}^t, t)(\widetilde{\boldsymbol{F}}^t - \overline{\boldsymbol{F}}^t)$$
$$\approx \boldsymbol{G}^t \left( \nabla_{\widetilde{\boldsymbol{F}}^t} D(\overline{\boldsymbol{F}}^t, t)\widetilde{\boldsymbol{F}}^t + \underbrace{\left[ (\boldsymbol{e}_1 - \sigma(\overline{\boldsymbol{f}}_1^t))^{\top}, \cdots, (\boldsymbol{e}_k - \sigma(\overline{\boldsymbol{f}}_K^t))^{\top} \right]^{\top} - \nabla_{\widetilde{\boldsymbol{F}}^t} D(\overline{\boldsymbol{F}}^t, t)\overline{\boldsymbol{F}}^t}_{\text{``residue''}} \right) \tag{50}$$

Where $\nabla_{\widetilde{\boldsymbol{F}}^t} D(\overline{\boldsymbol{F}}^t, t)$ is the Jacobian of the drift of class means w.r.t $\widetilde{\boldsymbol{F}}^t$. The "residue" term derived in the original proof by Zhang et al. (2021b) is an approximation of the one mentioned above and is shown to tend to 0 around convergence. *The negligible residue implies a close resemblance of the class mean drifts by the*

*representative feature drifts for all classes. Thus demonstrating the emergence of NC properties under the effective training assumption.* We found that the experimental setup of Zhang et al. (2021b) focuses mainly on LE behaviour, and analysis pertaining to $\mathcal{NC}$1-4 metrics was lacking. For closure, we mention that, despite the non-negligible residues and linearization effects of the LE-SDE/ODE model, this approach provided a faithful approximation of the feature separation dynamics on CIFAR10 and a synthetic GeoMNIST dataset. Furthermore, if one wishes to track $\mathcal{NC}$1-4 metrics, a batch size of 1 is needed to explicitly track the drift of each and every feature, making the model susceptible to noise and extremely slow for large scale data sets such as ImageNet. Furthermore, the complexity of data sets (w.r.t number of classes) might have different implications on the drift behaviour, leading to unexpected deviations from exhibiting NC as a by-product.

### 3.3 Retrospect

After a detailed analysis of UFM and LE-SDE modelling techniques, we summarize our observations in this section. In the work of Lu & Steinerberger (2020); Wojtowytsch et al. (2020), the authors impose norm constraints on the weights, and features and observe that the simplex ETF configuration leads to a global minimizer solution for the ERM with cross-entropy loss. Additionally, Wojtowytsch et al. (2020) analyse two-layer neural networks in the mean-field setting and for non-convex input classes in their paper to conclude that NC might not be desirable in such cases. Thus, suggesting an inherent difference in the expressiveness of shallow networks with infinite-width and deep neural networks. In the concurrent work of Mixon et al. (2020), the authors presented a gradient flow analysis of $F$, $A$ by employing a linearized ODE. By identifying an invariant subspace $\mathcal{S}$ along which strong neural collapse holds, the authors show that solutions along $\mathcal{S}$ are indeed the minimizers of the ERM with squared error loss. However, the linearization by partially decoupling $F$, $A$ is valid in regimes where their norms are small, which is typically not desirable in practical settings with deep networks suffering from vanishing gradients.

In the following work by Zhu et al. (2021), the authors take a different approach and analyse the loss-landscape with cross-entropy and regularization. Unlike the work of Ji et al. (2021), which presented an analysis of the cross-entropy loss and concluded that the implicit regularization of gradient flow is sufficient for attaining NC solutions, Zhu et al. (2021) emphasizes the benign nature of the regularized landscape which allows stochastic optimizers to reach global minima and exhibit NC properties. Additionally, by fixing the last layer classifier of ResNet50 and DenseNet169 as a simplex ETF for classification on ImageNet, Zhu et al. (2021) report $\approx 10\%$ reduction in the number of trainable parameters with a negligible drop in performance. For the regularized (mean) squared error setting, the works of Tirer & Bruna (2022) and Han et al. (2021) present a multifaceted approach to analysing NC. Han et al. (2021) obtain a closed-form solution for $A$ in terms of $F$ and show that the gradient flow trajectory of normalized features along this 'central path' is of particular importance to the learning dynamics. Their experiments employ canonical deep classifier networks and datasets to validate the theoretical results and in a sense complement the trajectory analysis of Mixon et al. (2020) in a more rigorous fashion. The work of Tirer & Bruna (2022) analyses a similar ERM setting with varying choices pertaining to bias and bias-regularization. Furthermore, Tirer & Bruna (2022) and Han et al. (2021) draw similar conclusions that the ideal bias value pertains to that of a global mean subtractor for the penultimate layer features. Additionally, Tirer & Bruna (2022) presented an extended version of the UFM by adding an extra linear layer. In the absence of non-linearity, they show that NC solutions are indeed desirable by splitting the 3-factor optimization problem into multiple 2-factor variants. Although they partially address the non-linear case, a rigorous analysis is still open for research. To this end, the work of Ergen & Pilanci (2021) analyses this extended variant of the UFM with respect to practical settings such as ReLU activations and batch-normalization[4]. Their approach is based on a convex-dual formulation of the ERM for any convex loss function and highlights the importance of batch-normalization in obtaining bounded and NC-satisfying optimal solutions.

Unlike the above efforts which rely on a balanced class setting, Fang et al. (2021) explored the implications of class imbalance on the geometry of minimizers. Their approach relies on the work of Sturm & Zhang (2003) and relaxes a quadratically constrained quadratic program as a semidefinite program. Their work highlights the 'minority collapse' phenomenon and studies the effects of class imbalance ratio on the features of minority classes. To this end, the empirical results of Yang et al. (2022) complement the work of Zhu et al.

---

[4]We present a simple application of their results in the Appendix.A.4

Table 1: Unified comparison of modelling techniques (MT) (UFM, LE or a generic analysis) based on weight/feature constraints or regularization (Reg), loss function $\ell$, class distribution constraints (balanced (B), imbalanced (IB)) for theoretical modelling, the main approach for analysis, the NC properties (NC1-4) that are derived and finally whether the authors provide the relevant empirical analysis (EA).

| | **MT** | **Reg** | $\ell$ | $n_k$ | **Approach** | **NC** | **EA** |
|---|---|---|---|---|---|---|---|
| Wojtowytsch et al. (2020) | UFM | - | CE | B | Global Minimizer | 1, 2 | - |
| Lu & Steinerberger (2020) | UFM | - | CE | B | Global Minimizer | 1, 2 | - |
| Ji et al. (2021) | UFM | - | CE | B | Gradient Flow + KKT + Loss Landscape | 1, 2, 3 | ✓ |
| Zhu et al. (2021) | UFM | ✓ | CE | B | Global Minimizer + Loss Landscape | 1, 2, 3 | ✓ |
| Fang et al. (2021) | UFM | ✓ | CE,CL | B, IB | Convex Relaxation + Global Minimizer | 1, 2, 3 + MC | ✓ |
| Mixon et al. (2020) | UFM | - | SE | B | Gradient Flow at Initialization + Linearization | 1, 2, 3 | ✓ |
| Han et al. (2021) | UFM | ✓ | MSE | B | Gradient Flow + Central Path | 1, 2, 3 | ✓ |
| Tirer & Bruna (2022) | UFM | ✓ | MSE | B | Global Minimizer + Extended UFM | 1, 2, 3 | ✓ |
| Poggio & Liao (2019) | - | ✓ | SE | B, IB | Gradient Flow | 1, 2, 3 | - |
| Ergen & Pilanci (2021) | - | ✓ | SE | B, IB | Convex Dual + Whitening | 1, 2, 3 | - |
| Zhang et al. (2021b) | LE | - | CE | B, IB | Dynamics SDE + Linearization | 1, 2 | - |

(2021) and highlight the practical relevance of fixing the final layer weights as a simplex ETF to address the minority collapse issue.

The 'local elasticity' based approach is relatively less explored than the UFM and is primarily analyzed in the work of Zhang et al. (2021b). In the case of UFM, the expressivity of the network was exploited by considering the penultimate layer features to be freely optimizable. Whereas in the LE-SDE setup, the local elasticity of the network was expected to result in sufficiently expressive features. Based on this assumption, Zhang et al. (2021b) showed that similar drift patterns of representative features and class means as $t \to \infty$ lead to NC-based solutions. However, the authors did not present any experimental results to validate these theoretical results. On a related and non-rigorous note, the LE-SDE formulation can be considered as a flexible version of the gradient flow of features/logits. Especially, the linearization of the drift in LE-SDE to model the non-linear dynamics allowed Zhang et al. (2021b) to faithfully approximate the training dynamics around convergence (see Appendix.B.2 in Zhang et al. (2021b) for additional discussion on the initialization regime). Such an approach avoids the negligible weight norm assumptions that Mixon et al. (2020) had to employ for linearization of the gradient flow ODE.

Overall, by employing the UFM approach, one can analyse the optimal closed-form solutions and gradient flow dynamics of $\boldsymbol{F}$, $\boldsymbol{A}$ while also leveraging convex relaxation approaches to study the otherwise non-convex ERM problem. On the other hand, the LE-SDE approach is limited to gradient flow-like analysis and heavily relies on the linearization of the SDE to derive insights on NC. Although both approaches fail to provide an accurate picture of the pre-TPT phases of training[5], the drift formulation and transformation matrix choices in the LE-SDE approach present a good starting point to analyse the initial phases of training. Similarly,

---

[5]Note that the difficulty primarily lies in accurately modelling the non-linear dynamics of the gradient flow. Thus, a majority of the efforts aim to faithfully approximate it through linearization.

in the case of UFM, we believe that incorporating non-linearity and additional layers can be a first step in deriving new insights for inner layers. Especially, as discussed above, the difficulty in solving this problem lies in factorizing the matrices after an activation function has been applied. A detailed comparison of various implementations of these models is presented in Table 1[6].

## 4 Implications on Generalization and Transfer Learning

In this section, we primarily focus on the connections between NC and the generalization capabilities of over-parameterized networks. The modelling techniques discussed in the previous section are limited to the training regime and explain the desirability/dynamics of attaining NC-based minimizers. To this end, we shed light on the empirical results by Zhu et al. (2021), which indicate the discrepancy in test performance of networks exhibiting NC during the training phase (see figure 7). This observation highlights that NC during training (let's call it train-collapse for convenience) doesn't necessarily guarantee good generalization. Also, one might wonder if train-collapse is just a result of the optimization process and doesn't necessarily describe the effectiveness of the learnt features. So, what is an effective approach to capture the impact of train collapse on test data? If train collapse doesn't guarantee generalization, how does it affect transfer learning? We address the questions by analysing recently proposed generalization and transfer learning bounds based on NC and discuss their validity.

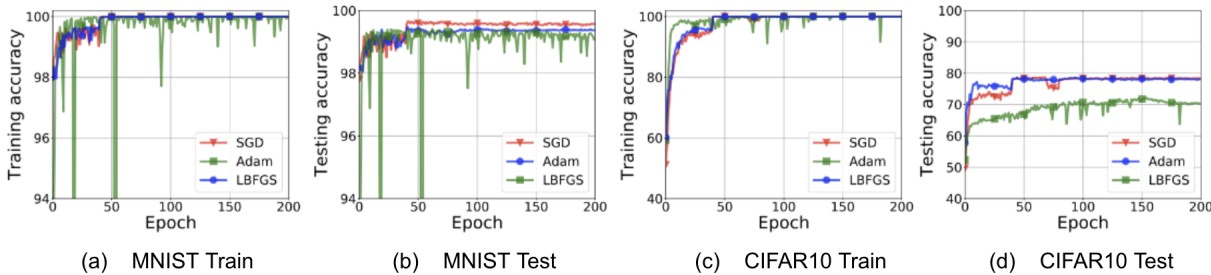

Figure 7: Train vs test performance of ResNet18 on MNIST (first two plots) and CIFAR10 (last two plots) with SGD, Adam and LBFGS optimizers. SGD with momentum 0.9 and Adam with $\beta_1 = 0.9, \beta_2 = 0.999$ were initialized with learning rate $0.05, 0.001$ respectively and scheduled to decrease by a factor of 10 every 40 epochs. LBFGS was initialized with memory size 10, learning rate 0.1 and employed a Wolfe line-search strategy for following iterations. Weight decay is commonly set to $5 \times 10^{-4}$. Image credit: Zhu et al. (2021)

### 4.1 How to evaluate a test collapse?

NC properties during training are measured when perfect classification has already been achieved by the network. Since we generally cannot guarantee perfect classification on test data, we consider a relaxation of perfect classification during testing as proposed by Hui et al. (2022) and define:

**Weak test-collapse**: This variant of collapse mandates that test samples should collapse to either one of the $K$ class means: $\boldsymbol{\mu}_1, \boldsymbol{\mu}_2, \ldots, \boldsymbol{\mu}_K$, and not necessarily to the mean of the class that it actually belongs to.

**Strong test-collapse** This variant mandates that test samples should collapse to the 'correct' class mean.

Intuitively, the "weak" and "strong" notions of test collapse seem reasonable but they bring forward additional challenges. Strong test-collapse requires a Bayes-optimal classifier to exist based on the features of a limited number of samples. This is infeasible and too rigid of a requirement. On the other hand, weak test-collapse can be satisfied by fixing the penultimate layer of a network as an orthogonal frame representing the one-hot logits. However, this setup doesn't guarantee good performance as the network can misclassify all the test points to the wrong class means and still attain weak test-collapse.

---

[6]Although most of the approaches derive NC1-3 properties in their results, it is sufficient as Papyan et al. (2020) showed that NC1-2 imply NC3-4. Additionally, the experiments column is ticked only if the paper presents NC-related experiments.

In the empirical studies of Hui et al. (2022), when a ResNet18 was trained and tested on CIFAR10, the magnitude/extent of weak/strong test collapse was observed to be lesser than the train collapse. *Surprisingly, we observed that the value of weak/strong test collapse of ResNet18 on CIFAR10 in Hui et al. (2022) is on-par with the train-collapse of ResNet152 on ImageNet.* Thus, the notion of NC occurring on test data is entirely data-dependent and is not convincing enough for understanding generalization.

To the contrary, recent work by Galanti et al. (2021) presents a generalization bound based on the variance collapse property (NC1) and states that train collapse favours generalization. Formally, by defining a **"Class Distance Normalized Variance (CDNV)"** metric over class conditional distributions:

$$V_f(\mathbb{P}_{C_i}, \mathbb{P}_{C_j}) = \frac{\text{Var}_f(\mathbb{P}_{C_i}) + \text{Var}_f(\mathbb{P}_{C_j})}{2 \left\| \mu_f(\mathbb{P}_{C_i}) - \mu_f(\mathbb{P}_{C_j}) \right\|_2^2} \tag{51}$$

Where $\mu_f(\mathbb{P}_{C_i}) = \mathbb{E}_{x \sim \mathbb{P}_{C_i}}[f(x)]$ and $Var_f(\mathbb{P}_{C_i}) = \mathbb{E}_{x \sim \mathbb{P}_{C_i}}[\|f(x) - \mu_f(\mathbb{P}_{C_i})\|_2^2]$, the generalization bound by Galanti et al. (2021) states that:

$$P\left( V_f(\mathbb{P}_{C_i}, \mathbb{P}_{C_j}) \leq (V_f(\mathcal{D}_{C_i}, \mathcal{D}_{C_j}) + B)(1 + A)^2 \right) \geq 1 - \delta \tag{52}$$

Where $B \propto 1/(\left\| \mu_f(\mathcal{D}_{C_i}) - \mu_f(\mathcal{D}_{C_j}) \right\|_2^2), A \propto 1/(\left\| \mu_f(\mathbb{P}_{C_i}) - \mu_f(\mathbb{P}_{C_j}) \right\|_2)$, $\mu_f(\mathcal{D}_{C_i})$ is the mean of features $f(x_j^i), \forall j \in [n_i]$ pertaining to class $i$[7], sampled from the empirical distribution $\mathcal{D}_{C_i}$ and $\mathcal{D}_{C_i} \sim \mathbb{P}_{C_i}^{n_i}$ denotes an empirical distribution on class $i$ with $n_i$ data points. The bound in equation 52 essentially relates the population CDNV with empirical CDNV and the generalization gap (details omitted here for brevity). From the UFM analysis presented in the previous sections, we saw that $\left\| \mu_f(\mathcal{D}_{C_i}) - \mu_f(\mathcal{D}_{C_j}) \right\|^2$ is maximized when the class means $\mu_f(\mathcal{D}_{C_i}), i \in [K]$ attain the ideal simplex ETF configuration. Thus, by bounding $A, B$ and considering sufficiently large values of $n_i, n_j$, the generalization gap can be reduced and a train-collapse on classes $i, j$ i.e, $V_f(\mathcal{D}_{C_i}, \mathcal{D}_{C_j}) \to 0$ would indicate $V_f(\mathbb{P}_{C_i}, \mathbb{P}_{C_j}) \to 0$ with a high probability.

From a theoretical standpoint, the assumption of large $n_i, n_j$ is a natural way to approximate a true data distribution and seldom applies to practical settings (since a large amount of labelled data is usually hard to obtain). Thus, even though the experimental values of strong/weak test-collapse in Hui et al. (2022) showed higher values than train-collapse, their analysis conveys the same observation that, with an increase in the size of train data, the weak/strong test collapse can potentially tend to lower values (although the setting itself is not a good indicator of typical practical scenarios). Furthermore, from the experimental results on Mini-ImageNet by Galanti et al. (2021) in figure 8, we observed that for the same number of classes, the test CDNV is approximately an order of magnitude larger than train CDNV and holds resemblance with the trend of plots presented by Hui et al. (2022). *This observation is of paramount importance as the absence of thresholds for CDNV or $\mathcal{NC}1\text{-}4$, which indicate whether collapse has occurred or not, can lead to misleading/contradicting results in the community. We believe that better metrics or thresholds can help in drawing objective conclusions on the occurrence of collapse (train/test) and avoid subjective interpretations in future efforts.* Developing such objective metrics can prove to be tricky, especially when considering the varying number of classes in data sets, network architectures, optimizers etc and we open this question to the community for further thought.

## 4.2 A 'depth' based generalization bound

Till now, we observed that networks exhibit NC when they interpolate on train data and such memorization doesn't necessarily guarantee good test performance (for instance, based on the choice of optimizers). The CDNV-based generalization bound fails to explain such discrepancies. In this section, we analyse a generalization bound based on the seemingly obvious NC4 property and shed light on the occurrence of NC in random label settings. We begin by presenting simplified definitions from Galanti (2022) as follows:

---

[7]Think of it as an empirical approximation of $\mu_f(\mathbb{P}_{C_i})$ based on $\mathcal{D}_{C_i}$.

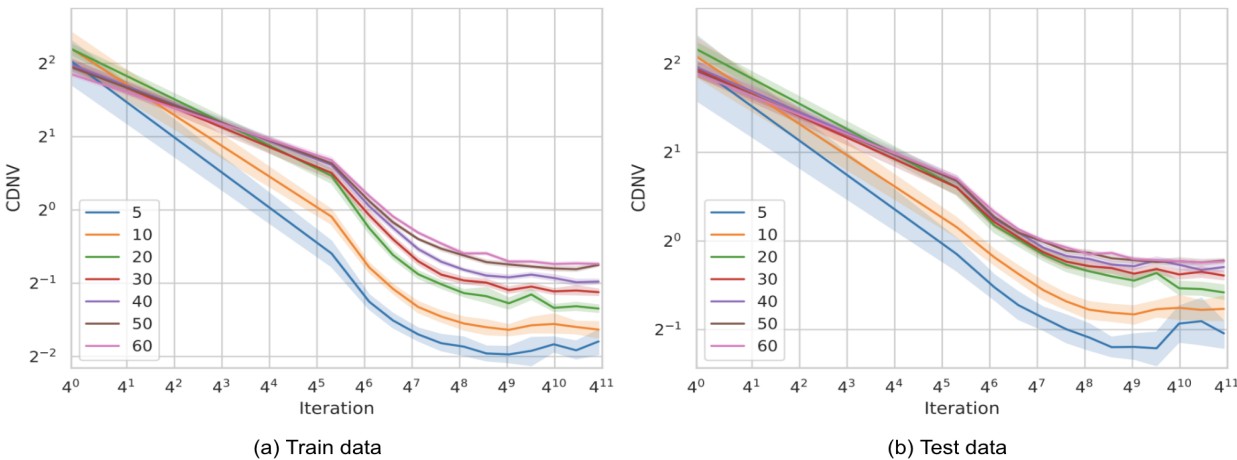

(a) Train data

(b) Test data

Figure 8: CDNV on train(left) and test(right) data of a Wide ResNet-28-4 (i.e depth factor of 28, width factor of 4) trained on Mini-ImageNet using SGD with momentum 0.9, learning rate $2^{-4}$. The legend indicates randomly selected classes for training/testing. Image credit: Galanti et al. (2021)

**$\epsilon$-effective depth:** For a network $h$ composed of $L$ layers, let $\hat{h}_l(\boldsymbol{x}) = \arg\min_{k \in [K]} \|f_l(\boldsymbol{x}) - \mu_{f_l}(\boldsymbol{X}_{C_k})\|, l \in [L]$.

The $\epsilon$-effective depth $\rho_{\boldsymbol{X}}^\epsilon(h)$ of $h$ on $\boldsymbol{X}$ is the minimum depth $l \in [L]$ for which $err_{\boldsymbol{X}}(\hat{h}_l) \leq \epsilon$. If such an $l$ doesn't exist then $\rho_{\boldsymbol{X}}^\epsilon(h) = L$.

Here $err_{\boldsymbol{X}}(\hat{h}_l) = \frac{1}{N}\sum_{i=1}^{N}\mathbb{I}[\hat{h}_l(\boldsymbol{x}_i) \neq \arg\max_{k \in [K]} \xi(\boldsymbol{x}_i)]$ is the nearest class center (NCC) misclassification error, $\mu_{f_l}(\boldsymbol{X}_{C_k})$ indicates the mean of features $f_l(.)$ for samples of class $k$, and finally recall from the preliminaries that $f_l = g_l \circ g_{l-1} \cdots \circ g_1 : \mathbb{R}^d \to \mathbb{R}^{m_l}$ is the composition of $l$ layers of the network. *Essentially, the $\epsilon$-effective depth $\rho_{\boldsymbol{X}}^\epsilon(h)$ represents the minimum depth at which the features can be classified by the NCC decision rule and achieve at most $\epsilon$ classification error.*

**$\epsilon$-Minimal NCC depth:** Let $\mathcal{G}$ represent a function class of layers in a network, the $\epsilon$-Minimal NCC depth $\rho_{min}^\epsilon(\mathcal{G}, \boldsymbol{X})$ on data set $\boldsymbol{X}$ is the minimum number of layers (belonging to $\mathcal{G}$) that can be composed to result in an output function $\tilde{f}$ such that $err_{\boldsymbol{X}}(\tilde{h}) \leq \epsilon$, where $\tilde{h}(\boldsymbol{x}) := \arg\min_{k \in [K]} \left\|\tilde{f}(\boldsymbol{x}) - \mu_{\tilde{f}}(\boldsymbol{X}_{C_k})\right\|$.

Now, consider the following setup: $\boldsymbol{X}_1, \boldsymbol{X}_2 \sim \mathbb{P}$ are two balanced data sets of size $N$. With a slight abuse of notation, we represent $h_{\boldsymbol{X}_1}^\kappa : \mathbb{R}^d \to \mathbb{R}^K$ as a network with weight initialization $\kappa$ and trained on dataset $\boldsymbol{X}_1$. When this network is evaluated on $\boldsymbol{X}_2$, we assume that the misclassified samples are uniformly distributed over the samples in $\boldsymbol{X}_2$ with probability $1 - \delta_N^1$. In practical settings, this assumption can be thought of as representing scenarios with non-hierarchical classes. As a second assumption, if $\boldsymbol{X}_1, \boldsymbol{X}_2$ contain noisy/random labels and both were to be used for training, we consider that with probability $1 - \delta_{N,p,\alpha}^2, p \in (0, 1/2), \alpha \in (0, 1)$, the $\epsilon$-minimal NCC depth to fit $(2 - p)N$ correct labels and $pN$ random labels is upper bounded by the expected $\epsilon$-minimal NCC depth to fit $(2 - q)N$ correct labels and $qN$ random labels for any $q \geq (1 + \alpha)p$. Under these assumptions, the generalization bound proposed by Galanti (2022) can now be formulated as follows:

$$\mathbb{E}_{\boldsymbol{X}_1}\mathbb{E}_\kappa[err_{\mathbb{P}}(h_{\boldsymbol{X}_1}^\kappa)] \leq P_{\boldsymbol{X}_1, \boldsymbol{X}_2, \widetilde{\boldsymbol{Y}}_2}\left[\mathbb{E}_\kappa[\rho_{\boldsymbol{X}_1}^\epsilon(h_{\boldsymbol{X}_1}^\kappa)] \geq \rho_{min}^\epsilon(\mathcal{G}, \boldsymbol{X}_1 \cup \widetilde{\boldsymbol{X}_2})\right] + (1 + \alpha)p + \delta_N^1 + \delta_{N,p,\alpha}^2 \qquad (53)$$

Where $err_{\mathbb{P}}(h) = \mathbb{E}_{\mathbb{P}}\mathbb{I}[\arg\max_{k \in [K]} h(\boldsymbol{x}) \neq \arg\max_{k \in [K]} \xi(\boldsymbol{x})]$ and $\widetilde{\boldsymbol{X}_2}$ is obtained by randomly relabelling $pN$ samples from $\boldsymbol{X}_2$. The noisy labels are now represented as $\widetilde{\boldsymbol{Y}}_2$. *Essentially, the bound indicates that, by randomly selecting $p \in (0, 1/2)$, if the expected $\epsilon$-effective depth of $h_{\boldsymbol{X}_1}^\kappa$ for all $\kappa$ is smaller than the $\epsilon$-minimal NCC depth $\rho_{min}^\epsilon(\mathcal{G}, \boldsymbol{X}_1 \cup \widetilde{\boldsymbol{X}_2})$, then the network is bound to do well on the test data.* For comprehensive proofs and tightness comparison of the bound, refer Galanti (2022).

Additionally, based on the empirical results of Galanti (2022) presented in figure 9, one can observe a clear indication of CDNV collapse as NCC train accuracy reaches 1 after a certain depth. Cohen et al. (2018) performed a similar study using a Wide ResNet on MNIST, CIFAR10, CIFAR100 & their random counterparts. It was observed that the layers gradually learn the k-NN based features when the dataset is clean and demand sufficient depth for randomly labelled data. The experiments by Galanti (2022) corroborate this observation and align such behaviour with the learning gap induced by $\delta^2_{N,p,\alpha}$ in equation 53 presented above. Furthermore, observe from figure 9 that NCC test accuracy and CDNV test of the final layer of the MLP saturates at $\approx 0.6, \approx 2^{-1}$ respectively in both the cases. Interestingly, we can observe a similar pair of these values during training, i.e when NCC train is $\approx 0.6$ in the pre-TPT stages, the CDNV train is $\approx 2^{-1}$. On the other hand, a CNN whose final layer was able to achieve $\approx 0.85$ NCC test accuracy showed a CDNV Test of $\approx 2^{-2}$ (see figure.5 in Appendix.A of Galanti (2022)). Such a pattern underscores the definition of strong test-collapse and a Bayes-optimal classifier that Hui et al. (2022) demand is the one we would ideally require to achieve that state. As a takeaway, note that for a network which is collapsed on train data, attaining collapse on test data depends on various factors such as data distribution, the implicit bias of a network and the optimizers used. Since attaining strong test collapse is usually an ambitious setting, one can nevertheless employ multifaceted approaches to track the extent of test collapse and gain a better understanding of the learning dynamics. For instance, Ben-Shaul & Dekel (2022) extend the cross-entropy loss by enforcing NCC behaviour across intermediate layers and show improved performance across vision and NLP sequence classification tasks.

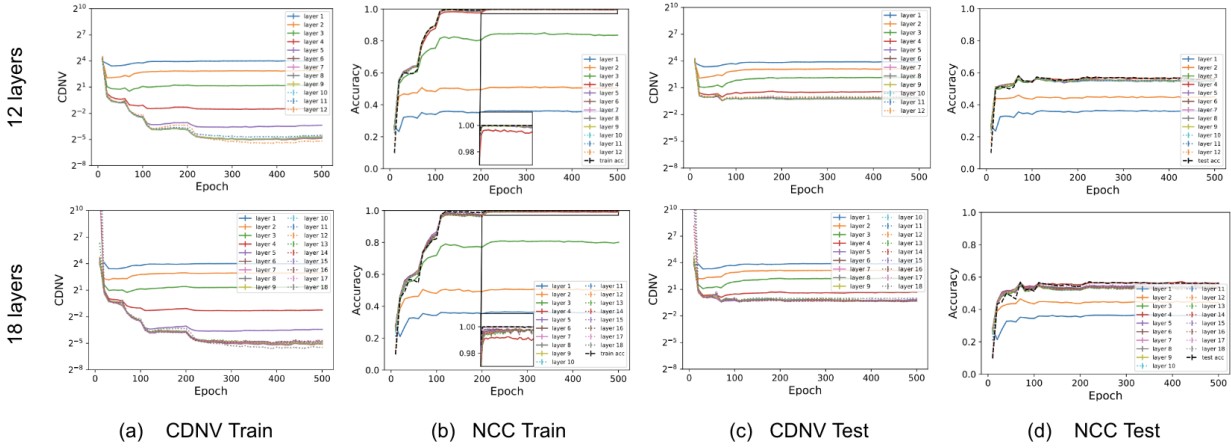

Figure 9: Layer-wise CDNV and NCC plots of an MLP trained on CIFAR10. Each hidden layer has a width of 300 and undergoes batch-normalization followed by ReLU activation. Cross-entropy loss is minimized using SGD with momentum 0.9, weight decay $5 \times 10^{-4}$, and initial learning rate 0.1, which is reduced by a factor of 10 at epochs $60, 120, 160$. Image credit: Galanti (2022)

### 4.3    Does collapse favour transferable representations?

Good classification performance is achieved when the networks are powerful enough and sufficient data is available for ERM (in a statistical sense). In settings where quality labelled data is scarce, transfer learning is a widely adopted technique for classification (Caruana, 1994; Thrun, 1998; Pan & Yang, 2010; Bengio, 2012; Weiss et al., 2016). In typical transfer learning settings, a large network is trained on a plethora of source tasks, followed by fine-tuning on downstream target tasks. The literature on the empirical effectiveness of this approach is quite rich (Long et al., 2015; Zamir et al., 2018; Houlsby et al., 2019; Raghu et al., 2019; Raffel et al., 2020; Kolesnikov et al., 2020; Brown et al., 2020) and various attempts have been made to theoretically understand this capability (Ben-David et al., 2006; Blitzer et al., 2007; Mansour et al., 2008; Zhang et al., 2013; Tripuraneni et al., 2020). A natural question that arises in this context is the role of collapse in learning transferable features.

The work by Galanti et al. (2021; 2022) addresses this question by extending the theoretical analysis of collapse from unseen data (as discussed in the previous section) to unseen classes. Specifically, their analysis focuses on multi-task settings where the source and target class conditionals are sampled i.i.d from a distribution over class conditionals. For a better understanding of the framework, consider a transfer learning setup with a $t$-class classification problem as the downstream task and a $s$-class classification problem as the source/auxiliary task. Formally, we extend our notation and assume the data for downstream and source tasks is sampled from $\mathbb{P}, \widetilde{\mathbb{P}}$ respectively. Next, similar to the setup for CDNV-based generalization bound, let $\mathcal{D}_{C_i} \sim \mathbb{P}_{C_i}^n$ denote a target data set for class $i \in [t]$ where $n$ data points have been sampled from $\mathbb{P}_{C_i}$. Along these lines, the target data set is comprised of $\mathcal{D} = \mathcal{D}_{C_1} \cup \cdots \cup \mathcal{D}_{C_t}$, and the source data set is comprised of $\widetilde{\mathcal{D}} = \widetilde{\mathcal{D}}_{C_1} \cup \cdots \cup \widetilde{\mathcal{D}}_{C_s}$. Finally, the class conditionals $\mathbb{P}_{C_i}, \forall i \in [t]$ and $\widetilde{\mathbb{P}}_{C_j}, \forall j \in [s]$ are assumed to be sampled i.i.d from a distribution $\mathbb{Q}$ over class conditional distributions $\mathbb{U}$[8]. In the work of Galanti et al. (2021), the authors randomly select two classes $k, k' \in [t]$ from the target task and bound the expected CDNV between them, $\mathbb{E}_{\mathbb{P}_{C_k} \neq \mathbb{P}_{C_{k'}}}[V_f(\mathbb{P}_{C_k}, \mathbb{P}_{C_{k'}})]$ using the average CDNV of the source classes, $\frac{2}{s(s-1)} \sum_{i=1}^{s} \sum_{i' \neq i}^{s} [V_f(\widetilde{\mathbb{P}}_{C_i}, \widetilde{\mathbb{P}}_{C_{i'}})]$, and terms which inversely depend on $\inf_f \inf_{\mathbb{P}_{C_k}, \mathbb{P}_{C_{k'}}} \left\| \mu_f(\mathbb{P}_{C_k}) - \mu_f(\mathbb{P}_{C_{k'}}) \right\|_2$. Now, by defining the expected transfer error as follows:

$$\mathcal{L}_{\mathbb{Q}}(f) := \mathbb{E}_{\mathbb{P}}\mathbb{E}_{\mathcal{D}}\left[\mathbb{E}_{(x,y)\sim\mathbb{P}}\left[\mathbb{I}[(a \circ f)_{\mathcal{D}}(x) \neq y]\right]\right] \tag{54}$$

the authors bound the transfer error by $\mathbb{E}_{\mathbb{P}_{C_k} \neq \mathbb{P}_{C_{k'}}}[V_f(\mathbb{P}_{C_k}, \mathbb{P}_{C_{k'}})]$ up to scaling (see Proposition.2 and Appendix.D in Galanti et al. (2021)). Here the expectation is taken over randomly selected target tasks from $\mathbb{P}$ and the limited available data $\mathcal{D}$, and $(a \circ f)_{\mathcal{D}}$ indicates the network (as per preliminaries) trained on $\mathcal{D}$. The issue with this bound pops up when $\left\| \mu_f(\mathbb{P}_{C_i}) - \mu_f(\mathbb{P}_{C_j}) \right\|_2$ tends to be very small. Thus, in settings where the transfer error is small, there is a possibility that the upper bound is very large and not indicative of the network's performance. Such scenarios occur when the support $\mathbb{U}$ for $\mathbb{Q}$ is infinitely large and an anomalous pair of target classes can turn this bound vacuous. To address this issue, Galanti et al. (2022) consider a specific case of ReLU networks with depth $r$ and bound the transfer error with the averaged CDNV over source classes (instead of target classes presented above) plus additional terms that depend on $t, s, r$ and a spectral complexity term which bounds the Lipschitz constant of $f$ (Golowich et al., 2018). Their theoretical results indicate that, with a sufficiently large number of source classes $s$ and data samples per source class, the prevalence of neural collapse on source data leads to small transfer errors, even with limited data samples in target data sets.

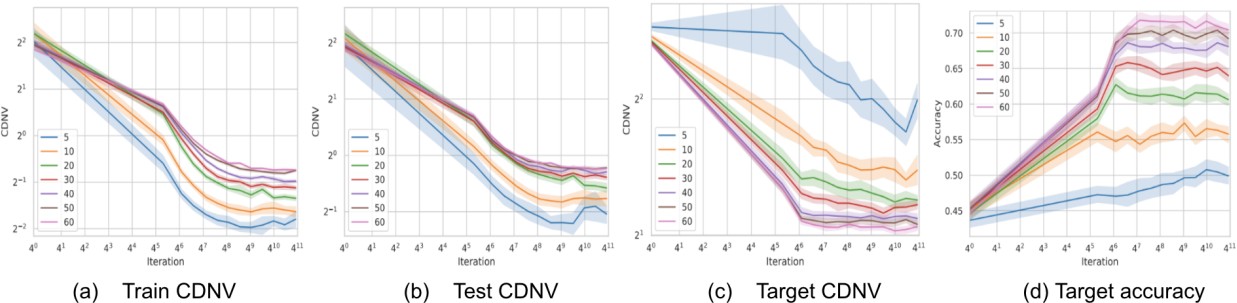

(a) Train CDNV      (b) Test CDNV      (c) Target CDNV      (d) Target accuracy

Figure 10: (a)-(b) Source train and test CDNV, (c)-(d) Target CDNV and accuracy for 5-shot classification on Mini-ImageNet using Wide ResNet-28-4. A varying number of classes (as per legend) are chosen for the source task and 5 classes are randomly selected for target tasks. SGD with momentum 0.9, learning rate $2^{-4}$ was employed to minimize the cross-entropy loss during training. Image credit: Galanti et al. (2021)

Empirical results on Mini-ImageNet pertaining to 100 5-shot classification experiments by Galanti et al. (2021) are shown in figure 10. In this experimental setup, a varying number of classes are randomly chosen

---

[8]On a lateral note, the setup is also amenable to covariate shift (Shimodaira, 2000) analysis and has been studied by Gretton et al. (2006); Huang et al. (2006); Zhang et al. (2013)

from the data set for pre-training a Wide ResNet-28-4 network. Next, a ridge-regression classifier is used as the final layer (with all the previous layers kept fixed) and trained on 5 randomly selected target classes. Finally, the network is tested on 100 random test samples from each of the 5 target classes for reporting the metrics. Observe that a clear pattern emerges in this plot, showcasing the benefits of pre-training on a large number of source classes as per the theoretical analysis.

Although the results look promising, recall that the bound and empirical analysis is restricted to a setting where the class conditional distributions are sampled from a common $\mathbb{Q}$. Thus, we cannot guarantee similar results for settings where the source and target distributions come from different $\mathbb{Q}$. The empirical results presented in Kornblith et al. (2021) are of significance in this context. The authors show that networks pre-trained on ImageNet tend to perform poorly on different downstream data sets (such as CIFAR10, CIFAR100, Flowers etc) when exhibiting a higher extent of collapse during pre-training. They especially show that softmax cross-entropy loss leads to relatively smaller margins between classes during pre-training and in turn results in better downstream performance when compared to losses such as squared error and cosine softmax. The impact of distribution changes (also known as model shift (Wang & Schneider, 2014)) is clearly evident from the experiments by Kornblith et al. (2021) and the bounds presented above fail to analyze this generalized setting.

## 5    Takeaways and Future Research

In retrospect, the study of neural collapse is essentially a study of desirable geometries and feature evolution dynamics in deep neural networks. Although each of the four NC properties has been studied in the literature, their unification under a common notion of 'collapse' has certainly piqued the interest of the community. Firstly, an extensive theoretical analysis of the global minimizers for cross-entropy loss from an NC perspective by Wojtowytsch et al. (2020); Lu & Steinerberger (2020) highlighted the difference in the expressive power of shallow and deep neural networks. In the following work by Han et al. (2021), the authors decomposed the MSE loss based on the gradient flow trajectory of features aligned with NC properties and demonstrated this behaviour for canonical deep classifier networks. Thus, shedding light on their training dynamics. Additionally, Zhu et al. (2021) leverage the simplex ETF property (NC2) and show $\approx 20\%$ reduction in the memory footprint of training ResNet18 on MNIST and $\approx 52\%$ reduction in the number of trainable parameters of ShuffleNet for ImageNet with 1000 classes. On a related note, Fang et al. (2021); Yang et al. (2022) propose solutions to address class imbalance training by analysing the collapse properties of under-represented classes. Furthermore, the study of NC has led to a series of efforts which attempt to explain the effectiveness of learnt representations for generalization and transfer learning. In summary, the studies pertaining to neural collapse have presented a unique approach to questioning and understanding the heuristic design choices and training dynamics involved in deep network training. In the following, we discuss open questions and future efforts that can have a broader impact:

**Modelling techniques:** The "unconstrained features" and "local elasticity" based models have provided a good theoretical ground for analysis. However, both are simplified presentations of the actual networks. Since it is extremely challenging to model every aspect of training a deep neural network in a theoretically tangible fashion, one can either extend these models incrementally (such as adding more layers to the UFM analysis) or approach this problem in a radically different way. The complexity lies in modelling the role of depth, non-linearities, normalization, loss functions, optimizers etc, all in a single model. During our review, we analysed each of these aspects individually but the challenge of unification is still an open problem and is of significance.

**Desirable geometries:** A $m$-simplex is one of the three regular polytopes that can exist in an arbitrarily high dimension $m$. The other two are $m$-cube and $m$-orthoplex (a cross-polytope) (Coxeter, 1973). Pernici et al. (2019; 2021) empirically show that by fixing the last layer of a VGG19 network as a $m$-cube with $m = \lceil \log_2 K \rceil$ or as a $m$-orthoplex with $m = \lceil K/2 \rceil$, the performance on CIFAR10, MNIST, EMNIST and FashionMNIST is comparable to that of a learnable baseline. However, if we desire maximum intra-class separation from our network, a $m$-simplex fits our needs. Interestingly, when $m$ increases, the angle between the weight vectors that form a $m$-simplex tends towards $\pi/2$, which is similar to the case of $m$-orthoplex. The question that arises in this context is whether a deep classifier network attains the $m$-orthoplex configuration

if the penultimate later features dimension is fixed to $m = \lceil K/2 \rceil$. Also, how does this structure affect class-imbalance training? Similar questions arise for $m$-cube as well.

**Learning objectives:** In addition to supervised classification settings using cross-entropy or squared error losses, we analysed that contrastive losses in supervised settings also favour collapse. This implies that either explicit or implicit label information is necessary for attaining the collapse state. It would be interesting to explore a similar phenomenon in unsupervised clustering tasks where labels are entirely absent (Xie et al., 2016; Min et al., 2018). Additionally, based on the results in Kornblith et al. (2021), a network exhibiting a relatively high extent of collapse on ImageNet was shown to transfer relatively badly to downstream classification tasks. To this end, it would be interesting to explore schemes such as *"maximal coding rate reduction (MCR$^2$)"* (Yu et al., 2020; Chan et al., 2020; Wu et al., 2021; Chan et al., 2022), which aims to preserve the intrinsic structure of within-class features along a subspace, while also increasing the distance between these subspaces. A rigorous analysis of such objectives from an NC perspective can shed light on the seemingly elusive implications of collapse.

**Generalization:** From our review of efforts which analyse test-collapse, it was highly subjective whether a certain value of $\mathcal{NC}1$ or CDNV can be deemed as exhibiting collapse or not. It is necessary for the community to standardize such observations in the early stages of this research direction and promote objective results. Furthermore, we observed that empirical results by Zhu et al. (2021) showcased the disparity in generalization performance of networks which attained train collapse using different optimizers. This is at odds with the CDNV-based generalization bounds that Galanti et al. (2021) propose, which heavily rely on train collapse. Further empirical and theoretical analysis is required to model such observations and improve the bounds.

**The special case of large-language models (LLM):** Based on the transfer learning bounds and experiments by Kornblith et al. (2021); Galanti et al. (2021; 2022), it would be interesting to track collapse metrics on the learned representations of large language models. Specifically, consider a case where an LLM is pre-trained in an unsupervised fashion on billions of text sequences and fine-tuned for a variety of classification tasks. In this setting, how would the collapse metrics differ when fine-tuning on tasks such as sentiment classification, document classification etc? Is it possible to propose transfer learning bounds when pre-training is unsupervised? Although such questions are primarily relevant at the scale of LLMs, novel methods to analyse such settings at a smaller scale can be of broad interest.

**Data domains:** From a geometric deep learning perspective (Bronstein et al., 2017), neural networks have been quite effective in learning on non-euclidean data such as graphs and manifolds. Since the architecture of such networks is highly dependent on the topological structure of data, novel modelling techniques are required to empirically and theoretically analyse NC in such settings.

# 6 Conclusion

In this work, we presented a principled review of modelling techniques that analyse NC and discussed its implications on the generalization and transfer learning capabilities of deep classifier neural networks. We presented a comprehensive review of the unconstrained features and local elasticity-based models by analysing their assumptions, settings and limitations under a common lens. Next, we discussed the possibility of neural collapse on test data, followed by an analysis of recently proposed generalization and transfer learning bounds. We hope our review, discussions and open questions would be of broad interest to the community and will lead to intriguing research outcomes.

**Acknowledgments**

The author would like to thank Joan Bruna, Tom Tirer and Federico Pernici for their informative discussions and the anonymous reviewers for their valuable feedback and time.

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

# A  Appendix

## A.1  Extended notations

Table 2: An extended set of notations for lookup.

| Notation | Description |
|---|---|
| $\boldsymbol{X} \in \mathbb{R}^{d \times N}$ | Input data to the network |
| $n_k, n \in \mathbb{N}$ | Imbalanced and balanced class sizes |
| $\boldsymbol{e}_i$ | one-hot vector w.r.t index $i$ |
| $\boldsymbol{0}$ | vector of all zeros of suitable dimension |
| $\boldsymbol{1}_K \in \mathbb{R}^K, \boldsymbol{I}_K \in \mathbb{R}^{K \times K}$ | Vector of all ones, Identity matrix |
| $\boldsymbol{x}_i \in \mathbb{R}^d$ | $i^{th}$ data point ($i^{th}$ column of $\boldsymbol{X}$) |
| $\boldsymbol{x}_i^k \in \mathbb{R}^d$ | $i^{th}$ data point of $k^{th}$ class |
| $C_k$ | Set of all data points belonging to class $k$ |
| $\xi : \mathbb{R}^d \to \{\boldsymbol{e}_1, \cdots, \boldsymbol{e}_K\}$ | A ground truth labelling function |
| $\mathbb{P}, \mathbb{P}_{C_k}$ | Data and class conditional data distributions |
| $\mathcal{H}$ | A function class of networks |
| $h_L : \mathbb{R}^d \to \mathbb{R}^K$, overload: $h$ | A network composed of $L$ layers |
| $a_L : \mathbb{R}^m \to \mathbb{R}^K$, overload: $a$ | Final layer linear function |
| $g_l : \mathbb{R}^{m_{l-1}} \to \mathbb{R}^{m_l}$ | The feature function for layer $l$ |
| $f_{L-1} : \mathbb{R}^d \to \mathbb{R}^m$, overload: $f$ | Composition of $L-1$ feature functions |
| $\boldsymbol{H}_L \in \mathbb{R}^{K \times N}$, overload: $\boldsymbol{H}$ | The network output matrix |
| $\boldsymbol{A}_L \in \mathbb{R}^{K \times m}$, overload: $\boldsymbol{A}$ | The final layer classifier matrix |
| $\boldsymbol{F}_{L-1} \in \mathbb{R}^{m \times N}$, overload: $\boldsymbol{F}$ | Penultimate layer feature matrix |
| $\boldsymbol{b}_L \in \mathbb{R}^K$, overload: $\boldsymbol{b}$ | Bias vector for layer $L$ |
| $\boldsymbol{Y} \in \mathbb{R}^{K \times N}$ | Label matrix |
| $\ell : \mathbb{R}^K \times \mathbb{R}^K \to [0, \infty)$ | Generic loss function |
| $\mathcal{R} : \mathcal{H} \to [0, \infty)$ | Population risk function |
| $\widehat{\mathcal{R}} : \mathcal{H} \to [0, \infty)$ | Empirical risk function |
| $\boldsymbol{\mu}_k \in \mathbb{R}^m$ | Mean of penultimate layer features of class $k$ |
| $\boldsymbol{\mu}_G \in \mathbb{R}^m$ | Mean of penultimate layer features class means |
| $\Sigma_W \in \mathbb{R}^{m \times m}$ | Within class covariance matrix for $\boldsymbol{F}$ |
| $\Sigma_B \in \mathbb{R}^{m \times m}$ | Between class covariance matrix for $\boldsymbol{F}$ |
| $\mathbb{I} : \{True, False\} \to \{0, 1\}$ | Indicator function |
| $q_{k,i}(\boldsymbol{F}, \boldsymbol{A}) \in \mathbb{R}$ | Margin of a data point $x_i^k$ |
| $\gamma_j, \upsilon_j \in \mathbb{R}$ | Batch-normalization scaling, shifting constants for $\boldsymbol{F}_{j:}$ |
| $\alpha_t, \beta_t \in \mathbb{R}$ | Intra-class, inter-class LE impact at time $t$ |
| $\nu_t \in \mathbb{R}$ | Local elasticity condition for feature separability |
| $\boldsymbol{f}_{k,i}^s \in \mathbb{R}^m$ | Penultimate layer features for $x_i^k$ at iteration $s$ |
| $\boldsymbol{E}^t \in \mathbb{R}^{K \times K}$ | LE impact matrix at time $t$ |
| $\boldsymbol{T}^t \in \mathbb{R}^{Km \times Km}$ | LE-SDE transformation matrix at time $t$ |
| $\boldsymbol{B}^t \in \mathbb{R}^{Km \times Km}$ | LE-SDE block transformation matrix at time $t$ |
| $\boldsymbol{W}^t \in \mathbb{R}^{Km}$ | Wiener process at time $t$ |
| $V_f(\mathbb{P}_{C_i}, \mathbb{P}_{C_j}) \in \mathbb{R}$ | Population CDNV for classes $i, j$ |
| $V_f(\mathcal{D}_{C_i}, \mathcal{D}_{C_j}) \in \mathbb{R}$ | Empirical CDNV for classes $i, j$ |
| $\rho_{\boldsymbol{X}}^\epsilon(h) \in \mathbb{R}$ | $\epsilon$-effective depth of $h$ w.r.t $\boldsymbol{X}$ |
| $err_{\boldsymbol{X}}(\hat{h}_l), err_{\mathbb{P}}(\hat{h}_l) \in \mathbb{R}$ | Empirical, population NCC misclassification error |
| $\mathcal{L}_{\mathbb{Q}}(f) \in \mathbb{R}$ | Transfer learning error/Transfer error |
| $\|\cdot\|_F, \|\cdot\|_*$ | Frobenius norm, Nuclear norm |
| $\langle . \rangle, \odot$ | Inner product, Hadamard product |
| $\mathrm{tr}\{.\}, \dagger, \top$ | Trace, Pseudo-inverse and Transpose of a matrix |

## A.2   A note on lower bounds for population risk with collapsed outputs and cross-entropy loss

*Without loss of generality, for a subspace* $\mathbb{B} \in \mathbb{R}^d$ *where* $\ell_{CE}$ *is strictly-convex, we can show that* $\mathcal{R}(\bar{h}) \le \mathcal{R}(h)$ *for* $\mathbb{P}$*-almost every* $\boldsymbol{x}$*, where* $\bar{h} \in \mathcal{H}$ *such that* $\bar{h}(\boldsymbol{x}) = \boldsymbol{z}_k, \forall \boldsymbol{x} \in C_k$ *(Where* $\boldsymbol{z}_k$ *is given by Eq.10).* A brief sketch of the proof by Wojtowytsch et al. (2020) is as follows:

Let $\Phi : \mathbb{R}^k \to \mathbb{R}^k$ be the softmax function, given by:

$$\Phi(\boldsymbol{h}) = \left( \frac{\exp(\langle \boldsymbol{h}, \boldsymbol{e}_1 \rangle)}{\sum_{i=1}^{K} \exp(\langle \boldsymbol{h}, \boldsymbol{e}_i \rangle)}, \cdots, \frac{\exp(\langle \boldsymbol{h}, \boldsymbol{e}_K \rangle)}{\sum_{i=1}^{K} \exp(\langle \boldsymbol{h}, \boldsymbol{e}_i \rangle)} \right), \Phi_k(\boldsymbol{h}) = \frac{\exp(\langle \boldsymbol{h}, \boldsymbol{e}_k \rangle)}{\sum_{i=1}^{K} \exp(\langle \boldsymbol{h}, \boldsymbol{e}_i \rangle)} \tag{55}$$

Where $\boldsymbol{h}$ is a vector input. Now, by taking the Taylor expansion of $\Phi_k$ near $\boldsymbol{z}_k$, we get:

$$\int_{C_k} \Phi_k(h(\boldsymbol{x})) \mathbb{P}(d\boldsymbol{x}) \approx \int_{C_k} \left[ \Phi_k(\boldsymbol{z}_k) + \nabla \Phi_k(\boldsymbol{z}_k) \cdot (h(\boldsymbol{x}) - \boldsymbol{z}_k) + \frac{1}{2}(h(\boldsymbol{x}) - \boldsymbol{z}_k)^\top D^2 \Phi_k(\boldsymbol{z}_k)(h(\boldsymbol{x}) - \boldsymbol{z}_k) \right] \mathbb{P}(d\boldsymbol{x})$$

$$\ge \int_{C_k} \Phi_k(\boldsymbol{z}_k) \mathbb{P}(d\boldsymbol{x})$$

since $\int_{C_k} \nabla \Phi_k(\boldsymbol{z}_k) \cdot (h(\boldsymbol{x}) - \boldsymbol{z}_k) = \boldsymbol{0}$ from equation 10, and the hessian of $\Phi_k$ is positive semi-definite. The equality holds when $h(\boldsymbol{x}) - \bar{h}(\boldsymbol{x}) \in span\{(1, \dots, 1)\}$ since $\ell_{CE}$ is strictly convex on the orthogonal complement of $(1, \dots, 1)$. See Lemma 2.1, 3.1 in Wojtowytsch et al. (2020) for the complete proof.

## A.3   Gradient flow analysis of Mean Squared Error without regularization

Based on our analysis of the squared error without regularization, we show that the same line of reasoning holds true for MSE as well. At first glance, the scaling factor of $\frac{1}{N}$ seems benign as $(\boldsymbol{F}, \boldsymbol{A}, \boldsymbol{b})$ satisfying SNC for the squared error, minimize MSE as well. However, the purpose of this analysis is to observe how the subspace $\mathcal{S}$ and the rate of convergence of $b$ to $\frac{1}{K}\mathbf{1}_K$ is modified due to this scaling factor. We follow the same sketch as Mixon et al. (2020) and define the ERM for MSE as:

$$\min_{\boldsymbol{F}, \boldsymbol{A}, \boldsymbol{b}} \widehat{\mathcal{R}}_{MSE}(\boldsymbol{F}, \boldsymbol{A}, \boldsymbol{b}) = \frac{1}{2N} \left\| \boldsymbol{A}\boldsymbol{F} + \boldsymbol{b}\mathbf{1}_N^\top - \boldsymbol{Y} \right\|_F^2 \tag{56}$$

The corresponding gradient flow equation for $\Theta = (\boldsymbol{F}, \boldsymbol{A}, \boldsymbol{b})$ is given by:

$$\begin{aligned} \Theta'(t) &= -\nabla \widehat{\mathcal{R}}_{MSE}(\Theta(t)) \\ \nabla_{\boldsymbol{F}} \widehat{\mathcal{R}}_{MSE} &= \frac{1}{N} \boldsymbol{A}^\top (\boldsymbol{A}\boldsymbol{F} + \boldsymbol{b}\mathbf{1}_N^\top - \boldsymbol{Y}) \\ \nabla_{\boldsymbol{A}} \widehat{\mathcal{R}}_{MSE} &= \frac{1}{N} (\boldsymbol{A}\boldsymbol{F} + \boldsymbol{b}\mathbf{1}_N^\top - \boldsymbol{Y}) \boldsymbol{F}^\top \\ \nabla_{\boldsymbol{b}} \widehat{\mathcal{R}}_{MSE} &= \frac{1}{N} (\boldsymbol{A}\boldsymbol{F} + \boldsymbol{b}\mathbf{1}_N^\top - \boldsymbol{Y})\mathbf{1}_N \end{aligned} \tag{57}$$

Assuming that $\boldsymbol{F}, \boldsymbol{A}$ have small norms (leading to uncoupled bias), we analyse the following ODE:

$$\boldsymbol{F}'(t) = -\frac{1}{N} \boldsymbol{A}(t)^\top (\boldsymbol{b}(t)\mathbf{1}_N^\top - \boldsymbol{Y}), \ \boldsymbol{A}'(t) = -\frac{1}{N} (\boldsymbol{b}(t)\mathbf{1}_N^\top - \boldsymbol{Y})\boldsymbol{F}(t)^\top, \ \boldsymbol{b}'(t) = -\frac{1}{N}(\boldsymbol{b}\mathbf{1}_N^\top - \boldsymbol{Y})\mathbf{1}_N$$

with initial conditions $\boldsymbol{F}(0) = \boldsymbol{F}_0, \boldsymbol{A}(0) = \boldsymbol{A}_0, \boldsymbol{b}(0) = \boldsymbol{0}$. Let's begin by solving for the bias term:

$$\boldsymbol{b}'(t) = -\frac{1}{N}(\boldsymbol{b}(t)\mathbf{1}_N^\top - \boldsymbol{Y})\mathbf{1}_N = \frac{1}{N}(\boldsymbol{I}_K \otimes \mathbf{1}_n^\top - \boldsymbol{b}(t)\mathbf{1}_N^\top)\mathbf{1}_N = \frac{1}{N}(n\mathbf{1}_K - N\boldsymbol{b}(t)) = \frac{1}{K}(\mathbf{1}_K - K\boldsymbol{b}(t))$$

Based on the initial condition $\boldsymbol{b}(0) = \boldsymbol{0}$, if we consider $\boldsymbol{b}(t) = \beta(t)\mathbf{1}_K$, then:

$$\beta'(t) = \frac{1}{K}(1 - K\beta(t)) \implies \int \frac{K\beta'(t)}{1 - K\beta(t)}dt = \int 1 dt \implies \beta(t) = \frac{1 - e^{-t}}{K} \implies b(t) = (\frac{1 - e^{-t}}{K})\mathbf{1}_K$$

Let $U = (\boldsymbol{F}, \boldsymbol{A})$ and $U(t)' = L_t(U(t))$, where:

$$L_t(\boldsymbol{F}, \boldsymbol{A}) = \left( \boldsymbol{A}^\top \frac{1}{N}\big(\boldsymbol{I}_K \otimes \mathbf{1}_n^\top - \beta(t)\mathbf{1}_K\mathbf{1}_N^\top\big), \frac{1}{N}\big(\boldsymbol{I}_K \otimes \mathbf{1}_n^\top - \beta(t)\mathbf{1}_K\mathbf{1}_N^\top\big)\boldsymbol{F}^\top \right) \tag{58}$$

The self-adjoin property of $L_t$ is straightforward to check and is shown in Mixon et al. (2020). Next, we define the following sub spaces over which $\{L_t\}_{t \geq 0}$ are simultaneously diagonalizable.

$$E_1^\epsilon = \{(\boldsymbol{F}, \boldsymbol{A}) : \boldsymbol{F} = \frac{\epsilon}{\sqrt{n}}(\boldsymbol{A} \otimes \mathbf{1}_n)^\top, \mathbf{1}_K^\top \boldsymbol{A} = \boldsymbol{0}\}$$

$$E_2^\epsilon = \{(\boldsymbol{F}, \boldsymbol{A}) : \boldsymbol{F} = \epsilon \cdot \boldsymbol{z}\mathbf{1}_N^\top, \boldsymbol{A} = \sqrt{n}\mathbf{1}_K\boldsymbol{z}^\top, \boldsymbol{z} \in \mathbb{R}^m\}$$

$$E_3 = \{(\boldsymbol{F}, \boldsymbol{A}) : (\boldsymbol{I}_K \otimes \mathbf{1}_n^\top)\boldsymbol{F}^\top = \boldsymbol{0}, \boldsymbol{A} = \boldsymbol{0}\}$$

where $\epsilon \in \{\pm 1\}$. Now let's verify that these spaces are indeed the eigen spaces of $L_t$.

**Case 1:** $(\boldsymbol{F}, \boldsymbol{A}) \in E_1^\epsilon$

$$\boldsymbol{A}^\top \frac{1}{N}\big(\boldsymbol{I}_K \otimes \mathbf{1}_n^\top - \beta(t)\mathbf{1}_K\mathbf{1}_N^\top\big) = \frac{1}{N}(\boldsymbol{A}^\top \otimes \mathbf{1}_n^\top) = \epsilon\frac{\sqrt{n}}{N}\boldsymbol{F}$$

$$\frac{1}{N}\big(\boldsymbol{I}_K \otimes \mathbf{1}_n^\top - \beta(t)\mathbf{1}_K\mathbf{1}_N^\top\big)\boldsymbol{F}^\top = \frac{\epsilon}{N\sqrt{n}}\big(\boldsymbol{I}_K \otimes \mathbf{1}_n^\top - \beta(t)\mathbf{1}_K\mathbf{1}_K^\top \otimes \mathbf{1}_n^\top\big)(\boldsymbol{A} \otimes \mathbf{1}_n) = \epsilon\frac{\sqrt{n}}{N}\boldsymbol{A}$$

This implies, $\{(\boldsymbol{F}, \boldsymbol{A}), \epsilon\frac{\sqrt{n}}{N}\}$ form the eigen-pair for $L_t$ in $E_1^\epsilon$.

**Case 2:** $(\boldsymbol{F}, \boldsymbol{A}) \in E_2^\epsilon$

$$\boldsymbol{A}^\top \frac{1}{N}\big(\boldsymbol{I}_K \otimes \mathbf{1}_n^\top - \beta(t)\mathbf{1}_K\mathbf{1}_N^\top\big) = \frac{1}{N}(\sqrt{n}\boldsymbol{z}\mathbf{1}_K^\top)\big(\boldsymbol{I}_K \otimes \mathbf{1}_n^\top - \beta(t)\mathbf{1}_K\mathbf{1}_N^\top\big)$$

$$= \frac{\sqrt{n}}{N}\boldsymbol{z}\big(\mathbf{1}_K^\top \otimes \mathbf{1}_n^\top - K\beta(t)\mathbf{1}_N^\top\big)$$

$$= \frac{\sqrt{n}}{N}(\boldsymbol{I} - K\beta(t))\boldsymbol{z}\mathbf{1}_N^\top = \frac{\epsilon\sqrt{n}}{N}(1 - K\beta(t))\boldsymbol{F}$$

$$\frac{1}{N}\big(\boldsymbol{I}_K \otimes \mathbf{1}_n^\top - \beta(t)\mathbf{1}_K\mathbf{1}_N^\top\big)\boldsymbol{F}^\top = \frac{1}{N}\big(\boldsymbol{I}_K \otimes \mathbf{1}_n^\top - \beta(t)\mathbf{1}_K\mathbf{1}_N^\top\big)(\epsilon \cdot \mathbf{1}_N\boldsymbol{z}^\top)$$

$$= \frac{1}{N}\big((\boldsymbol{I}_K \otimes \mathbf{1}_n^\top)(\mathbf{1}_K \otimes \mathbf{1}_n) - \beta(t)\mathbf{1}_K\mathbf{1}_N^\top\mathbf{1}_N\big)(\epsilon \cdot \boldsymbol{z}^\top)$$

$$= \frac{\epsilon}{N}\big(n\mathbf{1}_K - N\beta(t)\mathbf{1}_K\big)\boldsymbol{z}^\top$$

$$= \frac{n\epsilon}{N}(\boldsymbol{I} - K\beta(t))\mathbf{1}_K\boldsymbol{z}^\top = \frac{\epsilon\sqrt{n}}{N}\big(1 - K\beta(t)\big)\boldsymbol{A}$$

This implies, $\{(\boldsymbol{F}, \boldsymbol{A}), \frac{\epsilon\sqrt{n}}{N}\big(1 - K\beta(t)\big)\}$ form the eigen-pair for $L_t$ in $E_2^\epsilon$.

**Case 3:** $(\boldsymbol{F}, \boldsymbol{A}) \in E_3$

$$\boldsymbol{A}^\top \frac{1}{N}\big(\boldsymbol{I}_K \otimes \mathbf{1}_n^\top - \beta(t)\mathbf{1}_K\mathbf{1}_N^\top\big) = \boldsymbol{0}$$

$$\frac{1}{N}\big(\boldsymbol{I}_K \otimes \mathbf{1}_n^\top - \beta(t)\mathbf{1}_K\mathbf{1}_N^\top\big)\boldsymbol{F}^\top = -\beta(t)\mathbf{1}_K\mathbf{1}_N^\top\boldsymbol{F}^\top = \boldsymbol{0}$$

since the eigen value is 0 in $E_3$, we can ignore it and represent the spectral decomposition of $L_t$ as:

$$L_t = \frac{\sqrt{n}}{N} \left( \Pi_1^+ - \Pi_1^- + (1 - K\beta(t))\Pi_2^+ - (1 - K\beta(t))\Pi_2^- \right) \tag{59}$$

Where $\Pi_i^\epsilon$ is the orthogonal projection onto $E_i^\epsilon$. This allows us to solve $U(t)' = L_t(U(t))$ by finding the orthogonal projection of $U(t)$ onto $E_i^\epsilon$. Thus, we get:

$$\Pi_1^\epsilon U'(t) = \frac{\epsilon\sqrt{n}}{N}\Pi_1^\epsilon U(t) \implies \Pi_1^\epsilon U(t) = e^{\frac{\epsilon\sqrt{n}}{N}t}U(0)$$

$$\Pi_2^\epsilon U'(t) = \frac{\epsilon\sqrt{n}}{N}(1 - K\beta(t))\Pi_2^\epsilon U(t) \implies \frac{\Pi_2^\epsilon U'(t)}{\Pi_2^\epsilon U(t)} = \frac{\epsilon\sqrt{n}}{N}\left(1 - K\left(\frac{1 - e^{-t}}{K}\right)\right) = \frac{\epsilon\sqrt{n}}{N}(e^{-t})$$

$$\implies \Pi_2^\epsilon U(t) = \exp\left(\frac{\epsilon\sqrt{n}}{N}(1 - e^{-t})\right)\Pi_2^\epsilon U(0)$$

As the final step in the analysis, we can apply the Pythagoras theorem and get:

$$\left\| U(t) - e^{\frac{\sqrt{n}}{N}t}\Pi_1^+(0) \right\|_E^2 = \left\| e^{-\frac{\sqrt{n}}{N}t}\Pi_1^-(0) + \exp\left(\frac{\sqrt{n}}{N}(1 - e^{-t})\right)\Pi_2^+ U(0) + \exp\left(\frac{-\sqrt{n}}{N}(1 - e^{-t})\right)\Pi_2^- U(0) \right\|_E^2$$

$$= e^{-\frac{2\sqrt{n}}{N}t}\left\| \Pi_1^-(0) \right\|_E^2 + \exp\left(\frac{2\sqrt{n}}{N}(1 - e^{-t})\right)\left\| \Pi_2^+ U(0) \right\|_E^2$$

$$+ \exp\left(\frac{-2\sqrt{n}}{N}(1 - e^{-t})\right)\left\| \Pi_2^- U(0) \right\|_E^2$$

$$\leq \left\| \Pi_1^-(0) \right\|_E^2 + e^{\frac{2\sqrt{n}}{N}}\left\| \Pi_2^+ U(0) \right\|_E^2 + \left\| \Pi_2^- U(0) \right\|_E^2$$

$$\leq e^{\frac{2\sqrt{n}}{N}}\left\| (I - \Pi_1^+)U(0) \right\|_E^2$$

Thus, by consider the subspace $\mathcal{T} = E_1^+$, we get the final result that $(\boldsymbol{F}, \boldsymbol{A}, \boldsymbol{b})$ along the gradient flow satisfy:

$$\left\| (\boldsymbol{F}(t), \boldsymbol{A}(t)) - e^{\frac{\sqrt{n}}{N}t} \cdot \Pi_{\mathcal{T}}(\boldsymbol{F}_0, \boldsymbol{A}_0) \right\|_E \leq e^{\frac{\sqrt{n}}{N}} \cdot \left\| \Pi_{\mathcal{T}^\perp}(\boldsymbol{F}_0, \boldsymbol{A}_0) \right\|_E, \quad \boldsymbol{b}(t) = \left(\frac{1 - e^{-t}}{K}\right)\mathbf{1}_K, \forall t \geq 0$$

Where $\|(\boldsymbol{F}, \boldsymbol{A})\|_E^2 = \|\boldsymbol{F}\|_F^2 + \|\boldsymbol{A}\|_F^2$ and $\Pi_{\mathcal{T}^\perp}$ is the orthogonal projection onto the subspace:

$$\mathcal{T} := \left\{ (\boldsymbol{F}, \boldsymbol{A}) : \boldsymbol{F} = \frac{1}{\sqrt{n}}(\boldsymbol{A} \otimes \mathbf{1}_n)^\top, \mathbf{1}_K^\top \boldsymbol{A} = \mathbf{0} \right\}$$

Note that the subspace $\mathcal{T}$ which in turn leads to subspace $\mathcal{S}$ is the same as the squared error case, but the rate at which $b \to \frac{1}{K}\mathbf{1}_K$ has changed from $e^{-Nt}$ to $e^{-t}$. The empirical consequences of this modified setting on SNC would be interesting to observe (especially the role of initialization), which we defer to future work.

### A.4 Applying the results for ReLU networks with batch-normalization to UFM

As a follow-up of the role of normalization in attaining NC based on unconstrained features and gradient flow, we leverage the results of Ergen & Pilanci (2021) for ReLU networks with the popular batch-normalization (BN) layer to present a different perspective on UFM based NC. Firstly, let the BN output of $\boldsymbol{F}$ be given as:

$$(\boldsymbol{F}_{\mathrm{BN}})_{j:} = \mathrm{BN}_{\gamma,\upsilon}(\boldsymbol{F}_{j:}) = \frac{(\boldsymbol{I}_N - \frac{1}{N}\mathbf{1}_N\mathbf{1}_N^\top)\boldsymbol{F}_{j:}}{\left\| (\boldsymbol{I}_N - \frac{1}{N}\mathbf{1}_N\mathbf{1}_N^\top)\boldsymbol{F}_{j:} \right\|_2} \cdot \gamma_j + \frac{\mathbf{1}_N}{\sqrt{N}} \cdot \upsilon_j, \ \forall j \in [m] \tag{60}$$

Where the batch-normalization output for every row of unconstrained features $\boldsymbol{F}_{j:} \in \mathbb{R}^N, \forall j \in [m]$ is denoted by $(\boldsymbol{F}_{\mathrm{BN}})_{j:}$. Here $\gamma_j \in \mathbb{R}$ is the scaling factor and $\upsilon_j \in \mathbb{R}$ is the shift factor. For the sake of analysis, we consider the modified risk based on the squared loss, similar to Ergen & Pilanci (2021); Ergen et al. (2021):

$$\min_{\boldsymbol{F},\boldsymbol{A}} \widehat{\mathcal{R}}_{SE-BN} = \frac{1}{2} \|\boldsymbol{A}(\boldsymbol{F}_{BN})_+ - \boldsymbol{Y}\|_F^2 + \frac{\lambda_{\boldsymbol{A}}}{2} \sum_{j=1}^{m} (\gamma_j^2 + \upsilon_j^2 + \|\boldsymbol{a}_j\|_2^2) \tag{61}$$

Where $(\boldsymbol{F}_{BN})_+$ indicates a ReLU non-linearity on $\boldsymbol{F}_{BN}$. In our analysis till now, the non-linearity was implicitly assumed for sufficient expressivity of $\boldsymbol{F}$. In this example, we take a step further and break down the role of batch-normalization and ReLU on the optimal configurations for such expressive features. We can obtain a closed-form solution to this optimization problem based on a convex dual formulation $\widetilde{\mathcal{R}}_{SE-BN}$ for $\widehat{\mathcal{R}}_{SE-BN}$ (see Ergen & Pilanci (2021) for an elaborate formulation). Let $\boldsymbol{y}_j$ be the $j^{th}$ row of $\boldsymbol{Y}$, then:

$$\gamma_j^* = \frac{\left\|\boldsymbol{y}_j - \frac{1}{N}\mathbf{1}_N\mathbf{1}_N^\top\boldsymbol{y}_j\right\|_2}{\|\boldsymbol{y}_j\|_2}, \ \upsilon_j^* = \frac{\mathbf{1}_N^\top\boldsymbol{y}_j}{\sqrt{N}\|\boldsymbol{y}_j\|_2}, \boldsymbol{F}^* = \begin{bmatrix} \boldsymbol{Y} \\ \mathbf{0}_{(m-K)\times N} \end{bmatrix} \tag{62}$$

The proof by Ergen & Pilanci (2021) was originally given for ReLU networks with batch-normalization where strong duality was shown to hold true, i.e the global optimum for $\widetilde{\mathcal{R}}_{SE-BN}$ are the solutions for $\widehat{\mathcal{R}}_{SE-BN}$ as well. The values in equation 62 are obtained as a direct application of theorem 4.4 from Ergen & Pilanci (2021) to the UFM, where the output of penultimate layer is now unconstrained. The optimal values can now be used to calculate $\boldsymbol{F}_{BN}^*$ as:

$$(\boldsymbol{F}_{\text{BN}}^*)_{j:} = \frac{(\boldsymbol{I}_N - \frac{1}{N}\mathbf{1}_N\mathbf{1}_N^\top)\boldsymbol{F}_{j:}^*}{\left\|(\boldsymbol{I}_N - \frac{1}{N}\mathbf{1}_N\mathbf{1}_N^\top)\boldsymbol{F}_{j:}^*\right\|_2} \cdot \gamma_j^* + \frac{\mathbf{1}_N}{\sqrt{N}} \cdot \upsilon_j^*$$

$$\implies \boldsymbol{F}_{\text{BN}}^* = \sqrt{\frac{K}{N}} \begin{bmatrix} \boldsymbol{Y} \\ \mathbf{0}_{(m-K)\times N} \end{bmatrix} \tag{63}$$

For simplicity, if we consider $m = K$ and center $\boldsymbol{F}_{BN}^*$ around its global mean, we get:

$$\boldsymbol{F}_{BN}^*(\boldsymbol{I}_N - \frac{1}{N}\mathbf{1}_N\mathbf{1}_N^\top) = \sqrt{\frac{K}{N}}(\boldsymbol{I}_K \otimes \mathbf{1}_n^\top)(\boldsymbol{I}_N - \frac{1}{N}\mathbf{1}_N\mathbf{1}_N^\top) = \sqrt{\frac{K}{N}}(\boldsymbol{I}_K \otimes \mathbf{1}_n - \frac{1}{K}\mathbf{1}_N\mathbf{1}_K^\top) \tag{64}$$

Thus, the features of class $k$ have collapsed to their mean value of $\sqrt{\frac{K}{N}}(\boldsymbol{e}_k - \frac{\mathbf{1}_K}{K})$ and one can verify that the class means lie on the rotated and scaled version of the simplex ETF. Although this setting enables us to obtain a closed form solution for $\widehat{\mathcal{R}}_{SE-BN}$ using techniques such as interior points methods on $\widehat{\mathcal{R}}_{SE-BN}^*$ (Alizadeh, 1995; Nemirovski & Todd, 2008), this convexity doesn't come for free as the convex program now consists of exponentially more terms to optimize (Ergen & Pilanci, 2021; Ergen et al., 2021).

Interestingly, equation 62 shows that the first $K$ rows of $\boldsymbol{F}^*, \boldsymbol{F}_{BN}^*$ are just the scaled versions of $\boldsymbol{Y}$. Does this indicate that batch-normalization layers in canonical deep neural networks facilitates collapse even in the earlier layers? Furthermore, can this intrinsic bias towards a symmetric structure in batch-normalization layers explain its role in faster convergence (Ioffe & Szegedy, 2015; Luo et al., 2018; Wei et al., 2019)? A recent work by Poggio & Liao (2019; 2020) shows an interesting relationship between norms of weights and margins of classification for squared error loss with regularization in ReLU networks. A consequence of this result is that batch-normalization leads to weights with smaller norms which allows the network to learn large margins for classification. A comprehensive gradient flow analysis is presented in Poggio & Liao (2019) and is a valuable follow up of the UFM to canonical deep neural networks.

## A.5 Code availability

Table 3: List of NC related open-source implementations.

| Model | Implementation | Reference |
|---|---|---|
| Neural Collapse (Papyan et al. (2020)) | pytorch | `neuralcollapse` |
| LE-SDE (Zhang et al. (2021b)) | pytorch | `zjiayao/le_sde` |
| SVAG (Li et al. (2021)) | pytorch | `sadhikamalladi/svag` |
| Layer Peeled Model (Fang et al. (2021)) | pytorch | `HornHehhf/LPM` |
| Local Elasticity (He & Su (2019)) | pytorch | `hornhehhf/le` |
| Max Margin (Lyu & Li (2019)) | tensorflow | `vfleaking/max-margin` |
| Separation and Concentration (Zarka et al. (2020)) | pytorch | `j-zarka/separation` |
| Unconstrained Feature Model (Zhu et al. (2021)) | pytorch | `tding1/Neural-Collapse` |

