# OpenReview forum: "Neural Collapse: A Review on Modelling Principles and Generalization"
_TMLR — Accepted by TMLR_

### Review · Reviewer_sBKs · 2023-02-16

**Summary Of Contributions:**

The submission is a review of two different approaches of studying neural collapse:

1. Unconstrained Feature Models, assuming that the penultimate layer is independent of the inputs and can be independently optimzed
2. Locally elastic models, assuming that the output of the network on sample $x$ is not significantly affected when performing SGD updates on an unrelated sample $x'$.

The authors review a number of papers and scenarios in each modelling approach and summarize their findings. They conclude with future directions and comments on how neural collapse affects transfer learning.

**Audience:**

Yes

**Claims And Evidence:**

Yes

**Requested Changes:**

See weaknesses.

Minor point: some stylistic / grammatical corrections are in order. For example, "it's" is often used instead of "its". Informal language is used e.g. "can't" instead of "cannot".

**Strengths And Weaknesses:**

**Strengths**
- The work is on a relevant topic and in my opinion certainly satisfies the TMLR requirements for being of interest to the community.
- The submission seems thorough in its review of literature that is directly related (however see weaknesses below).
- Plenty of mathematical details are provided such that the conclusions can be derived without needing to read the reference materials.

I reviewed the submission as a review article, that is with the assumption that the major scientific claims and contributions have been already submitted and peer reviewed independently. With this in mind, the article suffers from the following mild weaknesses.

**Weaknesses**

- Many interesting conclusions are buried underneath a lot of mathematics. A prospective reader needs to work through many lines of non-trivial math to get to the conclusions of each subsection. It would be very beneficial for the prospective reader if the authors had a section detailing all the conclusions (e.g. role of CE with or without regularization, role of normalization etc.). The reader can then delve into the mathematics in as much or as little detail as they would like.

- As a review article with two major modeling approaches, the authors simply ask the reader to "consider the best aspects of each for future research." A table is provided for simpler comparison but I believe it is a shortcoming that a more extensive comparison of the approaches - i.e. directly comparing their conclusions, whether or not the results are compatible etc. - is not provided. The reader could in principle read each section and draw their own conclusion, but the point of a review article is to facilitate these comparisons.

- It is not always clear what parts of the article are summaries / reviews of previous work and which parts are new. For example, are sections 3.2.2 / 3.2.3 reviews based on published work? The authors provide some citations but it is not clear if these citations cover the idea behind the entire subsection or are just related. I would recommend being 100% clear about these, i.e. "in this section we review ... following [...]" or "We summarize the results of [...] and [...] with different notation" or "These are novel conclusions based on the work of [...]".

---

> ### Author Response · Authors · 2023-02-27
> **Answer to Review of Paper758 by Reviewer sBKs #1**
>
> We appreciate the reviewer for their encouraging feedback.
>
> **Reviewer:** Many interesting conclusions are buried underneath a lot of mathematics. A prospective reader needs to work through many lines of non-trivial math to get to the conclusions of each subsection. It would be very beneficial for the prospective reader if the authors had a section detailing all the conclusions (e.g. role of CE with or without regularization, role of normalization etc.). The reader can then delve into the mathematics in as much or as little detail as they would like.
>
> **Answer:** Thank you for the suggestion. We have updated our manuscript with a summary of the ideas in section 3.3 (text highlighted in blue). Additionally, we have deferred the analysis on ReLU networks and batch normalization for UFM settings to the Appendix to keep the reader on track with the UFM analysis in section 3.1.5.
>
> **Reviewer:** As a review article with two major modeling approaches, the authors simply ask the reader to "consider the best aspects of each for future research." A table is provided for simpler comparison but I believe it is a shortcoming that a more extensive comparison of the approaches - i.e. directly comparing their conclusions, whether or not the results are compatible etc. - is not provided. The reader could in principle read each section and draw their own conclusion, but the point of a review article is to facilitate these comparisons.
>
> **Answer:** We have addressed this shortcoming in section 3.3 by improving the comparison of ideas based on Local elasticity and unconstrained features. Additionally, Table 1 has been updated to summarize and compare the approaches based on the modelling technique, regularization/weight constraints, loss functions, class distribution constraints, the main approach for analysis, the NC properties that the body of work derives/analyses and finally the experiments columns has been updated to reflect empirical analysis relevant to NC. Especially, we have added the `Approach` and `NC` column which directly compares the key techniques used by the authors and the NC properties that they guarantee. Additionally note that although most of the approaches derive NC1-3 properties in their results, it is sufficient as Papyan et al. (2020) showed that NC1-2 imply NC3-4.
>
> **Reviewer:** It is not always clear what parts of the article are summaries / reviews of previous work and which parts are new. For example, are sections 3.2.2 / 3.2.3 reviews based on published work? The authors provide some citations but it is not clear if these citations cover the idea behind the entire subsection or are just related. I would recommend being 100% clear about these, i.e. "in this section we review ... following [...]" or "We summarize the results of [...] and [...] with different notation" or "These are novel conclusions based on the work of [...]".
>
> **Answer:** We have updated every section of the manuscript to clearly state whether we are reviewing existing work or presenting our own calculations.
>
> **Reviewer:** Minor point: some stylistic / grammatical corrections are in order. For example, "it's" is often used instead of "its". Informal language is used e.g. "can't" instead of "cannot".
>
> **Answer:** These grammatical issues have been addressed in the updated manuscript.
>
> References:
>
> - Vardan Papyan, XY Han, and David L Donoho. Prevalence of neural collapse during the terminal phase of
> deep learning training. Proceedings of the National Academy of Sciences, 117(40):24652–24663, 2020.

---

> > ### Comment · Reviewer_sBKs · 2023-03-12
> > **Rebuttal response**
> >
> > The authors have addressed my primary concerns. I think the author's review article would be useful reference point for researchers in the field.

---

### Review · Reviewer_cr7x · 2023-02-21

**Summary Of Contributions:**

This paper presents a survey of recent results concerning the phenomenon of neural collapse. It summarizes the findings of a number of recent works and contrasts their assumptions, problem settings, and resulting findings to give an over-arching view of the research landscape. Additionally, the paper discusses the implications of neural collapse on generalization and transfer learning, drawing from the findings of

**Audience:**

Yes

**Claims And Evidence:**

Yes

**Requested Changes:**

 - Section 3 should more clearly delineate where the derivations are pulled from prior work and where the "from the ground up" modeling principle is being used to reproduce prior results.
  - The tables of results are a nice way to concisely summarize the contributions of a number of papers, however simply showing whether or not the authors conducted a type of analysis (as in Table 1) is not a particularly helpful format. Instead, what would be more useful is to directly compare theoretical results -- e.g. what types of NC properties are guaranteed, or what dynamics are followed by the gradent flow.
  - Section 4 should justify its large use of space for what appear to be relatively loose generalization bounds which hold in a relatively limited number of settings. Additionally, the discussion of transfer should be better-motivated: at the moment, the transfer result places very strong assumptions on the distribution of classes seen during training which makes the setting closer to a multitask iid setting than the generic distribution shift considered more broadly as the transfer learning setting.


**Strengths And Weaknesses:**

**Strengths:**

  - The paper consolidates a quickly-developing body of work into a single document, allowing the reader to easily compare results in a variety of settings. This is aided in particular by the parallel structure of the subsections found in section 3.
  - The paper provides helpful intuitions to the reader in the interpretation of many technical results. The additional space provided by the TMLR format allows for greater discussion than would be permitted in a conference paper, allowing for a more didactic approach.
  - The coverage of the literature seems quite thorough: while neural collapse is not my main research focus, I am somewhat familiar with the area and didn't notice any glaring omissions.

**Weaknesses:**
In contrast to other TMLR submissions I have read, this paper does not offer any novel theoretical or empirical results: instead, it provides a didactic summary of existing work on neural collapse. While surveys can certainly be useful tools to help bring people up to speed on an emerging literature, this one in particular faces two principal limitations: the field of neural collapse is still very nascent and so the burden of reading the literature is relatively light, and the paper doesn't provide much significant insight above and beyond that provided by the works it discusses.

  - While the paper claims to "analyse NC modelling techniques by unifying them under a common set of principles,", I was not able to determine what exactly this common set of principles was that was missing from the literature. Most of Section 3 seems to consist of summarizing the contributions of prior works according to their primary theoretical result, followed by a discussion of their interaction with the loss landscape. Aside from the shared notation, I don't see what additional principles are added by this paper on top of the contributions that already exist in the cited works.
  - It is not always clear where a derivation is new (and is thus replicating prior results under a unified framework) vs where it closely follows the cited text. In most cases that I dug into the latter seemed to be the case.
  -  Section 4 seems to be largely an expanded summary of two papers by Galanti (et al.), for which it takes up disproportionate space. This is particularly salient given that the practical relevance of the generalization bounds discussed in this section is quite limited.

---

> ### Author Response · Authors · 2023-02-27
> **Answer to Review of Paper758 by Reviewer cr7x #1**
>
> We are grateful for the detailed review of our manuscript by the reviewer and for the positive comments.
>
> **Reviewer:** While the paper claims to "analyse NC modelling techniques by unifying them under a common set of principles,", I was not able to determine what exactly this common set of principles was that was missing from the literature. Most of Section 3 seems to consist of summarizing the contributions of prior works according to their primary theoretical result, followed by a discussion of their interaction with the loss landscape. Aside from the shared notation, I don't see what additional principles are added by this paper on top of the contributions that already exist in the cited works
>
> **Answer:** Thank you for bringing up this confusion. We have adjusted our claims in the updated manuscript (text highlighted in blue) to clarify that we are not presenting any novel principles to unify the current modelling techniques. Instead, we present a review based on the existing principles of unconstrained features and local elasticity. Additionally, to avoid confusion due to UFM terminology, we have mentioned a brief note on the initial efforts that propose UFM and the N-layer peeled model in Section 3.
>
> **Reviewer:** It is not always clear where a derivation is new (and is thus replicating prior results under a unified framework) vs where it closely follows the cited text. In most cases that I dug into the latter seemed to be the case
>
> **Answer:** We have updated every section of the manuscript to clearly state whether we are reviewing existing work or presenting our own calculations. Additionally, we have deferred the analysis on ReLU networks and batch normalization for UFM settings to the Appendix to keep the reader on track with the UFM analysis in section 3.1.5.
>
> **Reviewer:** Section 4 seems to be largely an expanded summary of two papers by Galanti (et al.), for which it takes up disproportionate space. This is particularly salient given that the practical relevance of the generalization bounds discussed in this section is quite limited.
>
> **Answer:** Thank you for this suggestion. We have refined this section by removing figures and deleting text which is not significant in our presentation but was using up disproportionate space. However, since our aim is to present intuitive ideas (of the limited works on NC + generalization bounds) in a self-contained manuscript, the definitions, assumptions and setup for the generalization bounds seem to occupy up to 2 pages with the relevant figures in the updated manuscript. Please let us know if there is any additional scope for enhancement.
>
> **Reviewer:** The tables of results are a nice way to concisely summarize the contributions of a number of papers, however simply showing whether or not the authors conducted a type of analysis (as in Table 1) is not a particularly helpful format. Instead, what would be more useful is to directly compare theoretical results -- e.g. what types of NC properties are guaranteed, or what dynamics are followed by the gradient flow
>
> **Answer:** Thank you for the suggestion. We have modified Table 1 to summarize and compare the approaches based on the modelling technique, regularization/weight constraints, loss functions, class distribution constraints, the main approach for analysis, the NC properties that the body of work derives/analyses and finally the experiments columns have been updated to reflect empirical analysis relevant to NC. Especially, we have added the `Approach` and `NC` column which directly compares the key techniques used by the authors and the NC properties that they guarantee. Additionally note that although most of the approaches derive NC1-3 properties in their results, it is sufficient as Papyan et al. (2020) showed that NC1-2 imply NC3-4.
>
> **Reviewer:** Section 4 should justify its large use of space for what appear to be relatively loose generalization bounds which hold in a relatively limited number of settings. Additionally, the discussion of transfer should be better-motivated: at the moment, the transfer result places very strong assumptions on the distribution of classes seen during training which makes the setting closer to a multitask iid setting than the generic distribution shift considered more broadly as the transfer learning setting
>
> **Answer:** The spacing issue has been addressed in the above comment. Additionally, we agree with the reviewer that the assumptions are quite strongly related to the i.i.d setting for class conditional probabilities. We have updated section 4.3 to highlight this assumption before diving into the setup and results.

---

> > ### Author Response · Authors · 2023-02-27
> > **Answer to Review of Paper758 by Reviewer cr7x #2**
> >
> > **Reviewer:** In contrast to other TMLR submissions I have read, this paper does not offer any novel theoretical or empirical results: instead, it provides a didactic summary of existing work on neural collapse. While surveys can certainly be useful tools to help bring people up to speed on an emerging literature, this one in particular faces two principal limitations: the field of neural collapse is still very nascent and so the burden of reading the literature is relatively light, and the paper doesn't provide much significant insight above and beyond that provided by the works it discusses.
> >
> > **Answer:** We respect the reviewer's opinion of our manuscript. We have adjusted our claims to clarify that we are not proposing new principles for unifying NC modelling techniques. Instead, we present a review based on the existing principles of unconstrained features and local elasticity. Also, since the field of neural collapse is rapidly growing, it is of utmost importance for the community to have access to a self-contained review of the key theoretical and empirical results pertaining to NC. Especially, in addition to reviewing existing results, we present key connections and summaries of the approaches as required. For instance:
> >
> > - Based on the work of Taghvaei et al. (2017), we emphasize in section 3.1.4 that the loss landscape of the regularized MSE loss might not be benign as one would expect. Previous works which analysed NC in such settings failed to consider this aspect.
> > - We presented simplified calculations in the Appendix for relating the results of Ergen & Pilanci for ReLU networks + batch normalization to the UFM setting.
> > - Based on the empirical results pertaining to test performance, NC, CDNV and weak/strong collapse metrics, we raise key questions regarding the importance of objective thresholds for determining whether collapse has occurred or not. This is of utmost importance and difficult at the same time as the NC metrics are usually dependent on a variety of factors (ex: data distribution, class imbalance, the network design etc).
> > - We have additionally added a summary section in section 3 which summarizes the key ideas of the reviewed papers and draws relevant conclusions (text highlighted in blue).
> >
> > Finally, we believe that our intuitive and exhaustive presentation of concepts can facilitate new ideas. Especially, since the field is still nascent, our manuscript can help in bringing any new researcher up to speed in a short amount of time.
> >
> > References:
> >
> > - Vardan Papyan, XY Han, and David L Donoho. Prevalence of neural collapse during the terminal phase of
> > deep learning training. Proceedings of the National Academy of Sciences, 117(40):24652–24663, 2020.
> > - Amirhossein Taghvaei, Jin W Kim, and Prashant Mehta. How regularization affects the critical points in
> > linear networks. Advances in neural information processing systems, 30, 2017.

---

> > > ### Comment · Reviewer_cr7x · 2023-03-27
> > > **Response**
> > >
> > > Thanks to the authors for the detailed response and for addressing several of my concerns. I am updating my recommendation accordingly.

---

### Review · Reviewer_64hA · 2023-02-21

**Summary Of Contributions:**

This is a review paper on the "neural collapse" (NC) phenomenon in deep neural networks. The authors summarize, compare, and contrast recent work within a unified framework. Implications of NC on generalization and transfer learning are discussed.

**Audience:**

Yes

**Claims And Evidence:**

No

**Requested Changes:**

Some minor issues:
- In Introduction, what are "complexity based learning techniques"?
- In Section 2.1, I believe there is a typo where ||(r,s)||_E should be ||(r,s)||_F


**Strengths And Weaknesses:**

Strengths:
- Well-organized.
- The manuscript does what it claims, and summarizes recent work on this topic within a unified framework. I am not an expert in this topic but I found the review relatively easy to follow at a high level.
- The unified framework for studying NC was presented in a clear manner.

Weaknesses:
- I am skeptical about the prevalence of these neural collapse phenomena and their purported benefits. While the manuscript cites many recent papers to support these claims, I found myself wishing that there was a clear explanation for what exactly has been shown by this body of work, and why we should take it seriously. This review focuses more on summarizing the methods used for studying the phenomenon.
- The discussion of NC on generalization and transfer learning didn't seem to have much substance. Defining the terminal phase of training (TPT) as beginning when classification error (01-loss) reaches zero appears to be a useful shorthand, but this is not based on anything fundamental since these models are trained to minimize a different objective (i.e. cross-entropy loss). So why is it important to define "weak and strong test-collapse" for generalization and transfer learning scenarios? I couldn't see how this was useful.

---

> ### Author Response · Authors · 2023-02-27
> **Answer to Review of Paper758 by Reviewer 64hA #1**
>
> We are thankful to the reviewer for their constructive feedback regarding our manuscript.
>
>
> **Reviewer:** I am skeptical about the prevalence of these neural collapse phenomena and their purported benefits. While the manuscript cites many recent papers to support these claims, I found myself wishing that there was a clear explanation for what exactly has been shown by this body of work, and why we should take it seriously. This review focuses more on summarizing the methods used for studying the phenomenon.
>
> **Answer:** We have addressed this point by adding the relevant observations and key takeaways in Section 3.3 and Section 5 of the updated manuscript (changes marked in blue). On a brief note, please note that an extensive theoretical analysis of the global minimizers for cross-entropy loss from an NC perspective by Wojtowytsch et al. (2020); Lu & Steinerberger (2020) highlighted the difference in the expressive power of shallow and deep neural networks (since our paper primarily focuses on the modelling techniques and implications of NC on generalization/TL, we omit the in-depth analysis for brevity). Additionally, the work by Han et al. (2021) decomposes the MSE loss based on the gradient flow trajectory of features aligned with NC properties and demonstrated this behaviour for canonical deep classifier networks. Thus, shedding light on their training dynamics. Additionally, Zhu et al. (2021) leverage the simplex ETF property (NC2) and show ~ 20% reduction in the memory footprint of training ResNet18 on MNIST and ~ 52% reduction in the number of trainable parameters of ShuffleNet for ImageNet with 1000 classes. On a related note, Fang et al. (2021); Yang et al. (2022) propose solutions to address class imbalance training by analysing the collapse properties of under-represented classes. In summary, the studies pertaining to neural collapse have presented a unique approach to questioning and understanding the heuristic design choices and training dynamics involved in deep network training.
>
> **Reviewer:** The discussion of NC on generalization and transfer learning didn't seem to have much substance. Defining the terminal phase of training (TPT) as beginning when classification error (01-loss) reaches zero appears to be a useful shorthand, but this is not based on anything fundamental since these models are trained to minimize a different objective (i.e. cross-entropy loss). So why is it important to define "weak and strong test-collapse" for generalization and transfer learning scenarios? I couldn't see how this was useful.
>
> **Answer:** Although the models are trained to minimize objectives such as cross-entropy/MSE loss, the terminal phase of training (TPT) is of particular importance as it represents a regime where the network has interpolated on the training data and thus achieves zero-training error. Due to recent works which highlight the benign effects of interpolating on the training data with over-parameterized networks (for ex: Ma et al., 2018), and the initial work on NC by Papyan et al 2020 which shows NC metrics tending to 0 in this phase, the TPT is of particular interest. The notion of weak and strong test collapse was presented based on the work of Hui et al 2022. The purpose of presenting these metrics is to highlight the potential shortcomings of expecting NC1-4 values on test data to resemble that of the training regime. Additionally, since the generalization and transfer learning bounds depend on CDNV and not on weak/strong test collapse, we have modified our presentation of these two metrics to focus mainly on the idea. Furthermore, we have refined the presentation of Section 4 to convey the key ideas and assumptions behind the generalization and transfer learning bounds. Although the literature on this line of work is limited, we have presented connections with the work of Cohen et al 2018 on the role of random labels/noisy datasets on the effective depth and in section 4.3 briefly discussed the transfer learning bound and it's theoretical limitations in explaining empirical results of Kornblith et al 2021.
>
> **Reviewer:** In the Introduction, what are "complexity based learning techniques"?
>
> **Answer:** This was an error in the choice of words and has now been fixed.
>
>
> **Reviewer:** In Section 2.1, I believe there is a typo where ||(r,s)||_E should be ||(r,s)||_F
>
> **Answer:** We have adopted this notation from Mixon et al 2020, which indicates ||(r,s)||_E^2 = ||r||_F^2 + ||r||_F^2 and aids in simplifying the notation for the (r, s) tuple during gradient flow.

---

> > ### Author Response · Authors · 2023-02-27
> > **Answer to Review of Paper758 by Reviewer 64hA #2**
> >
> > References:
> >
> > - Stephan Wojtowytsch et al. On the emergence of simplex symmetry in the final and penultimate layers of
> > neural network classifiers. arXiv preprint arXiv:2012.05420, 2020.
> > - Jianfeng Lu and Stefan Steinerberger. Neural collapse with cross-entropy loss. arXiv preprint
> > arXiv:2012.08465, 2020
> > - XY Han, Vardan Papyan, and David L Donoho. Neural collapse under mse loss: Proximity to and dynamics
> > on the central path. arXiv preprint arXiv:2106.02073, 2021.
> > - Zhihui Zhu, Tianyu Ding, Jinxin Zhou, Xiao Li, Chong You, Jeremias Sulam, and Qing Qu. A geometric analysis of neural collapse with unconstrained features. Advances in Neural Information Processing Systems, 34, 2021
> > - Cong Fang, Hangfeng He, Qi Long, and Weijie J Su. Exploring deep neural networks via layer-peeled model:
> > Minority collapse in imbalanced training. Proceedings of the National Academy of Sciences, 118(43), 2021.
> > - Yibo Yang, Liang Xie, Shixiang Chen, Xiangtai Li, Zhouchen Lin, and Dacheng Tao. Do we really need a
> > learnable classifier at the end of deep neural network? arXiv preprint arXiv:2203.09081, 2022
> > - Vardan Papyan, XY Han, and David L Donoho. Prevalence of neural collapse during the terminal phase of
> > deep learning training. Proceedings of the National Academy of Sciences, 117(40):24652–24663, 2020.
> > - Gilad Cohen, Guillermo Sapiro, and Raja Giryes. Dnn or k-nn: That is the generalize vs. memorize question.
> > arXiv preprint arXiv:1805.06822, 2018.
> > - Simon Kornblith, Ting Chen, Honglak Lee, and Mohammad Norouzi. Why do better loss functions lead to
> > less transferable features? Advances in Neural Information Processing Systems, 34, 2021.
> > - Siyuan Ma, Raef Bassily, and Mikhail Belkin. The power of interpolation: Understanding the effectiveness
> > of sgd in modern over-parametrized learning. In International Conference on Machine Learning, pp.
> > 3325–3334. PMLR, 2018.

---

> > ### Comment · Reviewer_64hA · 2023-03-11
> > **These responses make sense**
> >
> > I appreciate the authors' thoughtful responses, and I agree with them. I no longer have any concerns about the claims and evidence.
> >
> > Another reviewer questioned the value to the community since this is mostly a review of the recent literature. I am not working in this area, so I cannot provide a strong defense, but I do think this manuscript will be of interest to some in the community.

---

### Decision · Action_Editors · 2023-04-05

**Recommendation:** Accept as is

**Comment:**

This paper presents a summary of recent work on the phenomenon of neural collapse, collecting previous results under a unified framework that is clear and easy to follow. It summarizes the findings of these papers and compares their assumptions and problem settings, yielding a comprehensive view of the research landscape.

Some reviewers noted that it is not always clear when new perspectives or derivations are provided, or when previous work is being reproduced directly. They also mentioned that sometimes the high-level conclusions were not accessible and were only visible after a detailed read of mathematical derivations. These weaknesses have been at least partially addressed, and in any case do not render the manuscript unsuitable for publication. I think this paper will provide value to the community and recommend acceptance.

**Audience:**

This review article will appeal to any reader wishing to learn about neural collapse; as such, the audience intersects substantially with TMLR readership.

**Claims And Evidence:**

The reviewers agree that the manuscript does what it claims, i.e. it surveys recent work on the topic and provides a unifying framework.